# SuperflexPy 1.3.0: An open source Python framework for building, testing and improving conceptual hydrological models

Marco Dal Molin[1,2], Dmitri Kavetski[1,3,4], Fabrizio Fenicia[1]

[1] Eawag, Swiss Federal Institute of Aquatic Science and Technology, Dübendorf, Switzerland
[2] Centre of Hydrogeology and Geothermics (CHYN), University of Neuchâtel, Neuchâtel, Switzerland
[3] School of Civil, Environmental and Mining Engineering, University of Adelaide, SA, Australia
[4] Civil, Surveying and Environmental Engineering, University of Newcastle, NSW, Australia

*Correspondence to*: Fabrizio Fenicia (fabrizio.fenicia@eawag.ch)

## Abstract

Catchment-scale hydrological models are widely used to represent and improve our understanding of hydrological processes and to support operational water resources management. Conceptual models, which approximate catchment dynamics using relatively simple storage and routing elements, offer an attractive compromise in terms of predictive accuracy, computational demands, and amenability to interpretation. This paper introduces SuperflexPy, an open-source Python framework implementing the

SUPERFLEX principles (Fenicia et al., 2011) for building conceptual hydrological models from generic components, with a high degree of control over all aspects of model specification. SuperflexPy can be used to build models of a wide range of spatial complexity, ranging from simple lumped models (e.g. a reservoir) to spatially distributed configurations (e.g. nested sub-catchments), with the ability to customize all individual model components. SuperflexPy is a Python package, enabling modelers to

exploit the full potential of the framework without the need for separate software installations, and making it easier to use and interface with existing Python code for model deployment. This paper presents the general architecture of SuperflexPy, discusses the software design and implementation choices, and illustrates its usage to build conceptual models of varying degrees of complexity. The illustration includes the usage of existing SuperflexPy model elements, as well as their extension to implement new

functionality. Comprehensive documentation is available online and provided as supplementary material to this paper. SuperflexPy is available as open-source code, and can be used by the hydrological community to investigate improved process representations, for model comparison, and for operational work.

# Table of Contents

70

# 1 Introduction

## 1.1 Conceptual hydrological models

Catchment-scale hydrological models are widely used to predict catchment behavior under natural and human-impacted conditions, as well as to represent and improve our understanding of internal catchment functioning (e.g. Beven, 1989). For example, catchment models underlie projections of climate change impact on groundwater recharge and streamflow (e.g., Eckhardt and Ulbrich, 2003), are used as tools for hypothesis testing to identify dominant hydrological processes (e.g., Clark et al., 2011b; Hrachowitz et al., 2014; Wrede et al., 2015), and are used to inform agricultural practices such as irrigation scheduling (e.g., McInerney et al., 2018) and pesticide application (e.g., Moser et al., 2018; Ammann et al., 2020). The typical use of hydrological models is to simulate or forecast the streamflow response (runoff) of a catchment to rainfall forcing; for this reason they are often referred to as rainfall-runoff models (e.g., Moradkhani and Sorooshian, 2009). However, their application extends to the simulation of other environmental variables such as groundwater levels (e.g., Seibert and McDonnell, 2002) and soil moisture (e.g., Matgen et al., 2012), as well as water chemistry (e.g., Bertuzzo et al., 2013; Ammann et al., 2020).

An important class of catchment models are "process based" models, which attempt to explicitly describe the cascade of processes transforming catchment inputs (e.g. precipitation) into outputs (e.g. streamflow). These models are an appealing choice due to their broad physical underpinnings, as well as their ability to represent internal catchment processes and potential for predicting catchment responses under changing environmental conditions. Process based models can be classified according to the nature of their constitutive equations (e.g. conceptual or physically based) and their spatial resolution (e.g. lumped or distributed) (e.g., Refsgaard, 1996).

Conceptual models, where catchment dynamics are approximated using relatively simple storage and routing elements (e.g. Fenicia et al., 2011), are common in practice because they offer an attractive compromise in terms of predictive accuracy, computational demands, and amenability to interpretation. Common conceptual models include TopModel (Beven and Kirkby, 1979), HBV (Lindstrom et al., 1997), GR4J (Perrin et al., 2003), and HyMod (Boyle, 2001).

In terms of spatial resolution, conceptual models can be applied in a lumped configuration (treating the entire catchment as a single unit) if the interest is in modeling integrated catchment outputs (e.g. streamflow at the catchment outlet). Alternatively, distributed configurations can be used if the interest is in modeling hydrological behavior at internal locations (e.g., sub-catchments). In such distributed setups, the catchment is subdivided into spatial elements such as sub-catchments (e.g., Feyen et al., 2008; Lerat et al., 2012), Hydrological Response Units (HRUs) (e.g., Arnold et al., 1998; Fenicia et al., 2016; Dal Molin et al., 2020), or grids (e.g., Samaniego et al., 2010). A common strategy for developing distributed

conceptual models is to represent individual landscape elements using independent (non-interacting) lumped models, and then obtain total catchment outflow by aggregating the outflows from these individual models, potentially incorporating flow routing elements to represent routing delays. This strategy is often referred to as "semi-distributed" modelling (e.g., Boyle et al., 2001), and typically employs discretization based on principles of "hydrological similarity" (e.g., Sivapalan et al., 1987); HRU-based discretization is particularly common (e.g., Leavesley, 1984). In many applications, semi-distributed modelling achieves good predictive ability – while greatly simplifying model representation and reducing computational demands compared to fully-integrated 2D/3D distributed models such as Parflow (Maxwell, 2013) or Mike She (Refsgaard and Storm, 1995), which typically use much smaller landscape elements and explicitly model lateral exchanges. For the purposes of this presentation, we consider semi-distributed modelling to be a special case of distributed modelling.

## 1.2  Hydrological model structure and flexible modeling frameworks

The selection of model structure has preoccupied researchers and practitioners since the early days of hydrological modelling (e.g., Ibbitt and O'Donnell, 1971; Moore and Clarke, 1981; Jakeman and Hornberger, 1993). Although in principle the physical laws governing hydrological processes are the same everywhere, the diversity of catchment conditions in terms of topography, soil, geology, vegetation, and anthropogenic influence results in remarkably different manifestations of these physical laws at the catchment scale. These local differences, also termed "uniqueness of place" (Beven, 2000), considerably limit our ability to develop generalizable hydrological hypotheses (e.g., Wagener et al., 2007).

Model structure selection has motivated multiple research directions, including the search for a single model structure that achieves good prediction across all catchments (the "fixed" model paradigm), and the search for model structures best suited for specific locations and/or environmental conditions (the "flexible" model paradigm). Whether in search of a single model or multiple models, model selection necessarily relies on a process of model development, comparison, and refinement. Approaches to formalize this process include the top-down approach (e.g. Sivapalan et al., 2003), the system identification approach (e.g Young, 1998), and the method of multiple working hypotheses (e.g., Clark et al., 2011a). These approaches are not mutually exclusive, as the notion of comparing multiple model representations is ubiquitous in model development and empirical science in general.

The process of model development, comparison, and refinement can be facilitated using flexible modeling frameworks, which enable hydrologists to hypothesize, implement, and (eventually) test and refine different model structures. Flexible frameworks have themselves developed along multiple directions according to their intended scopes of application. For example, GEOframe-NewAge (Formetta et al., 2014), SUMMA (Clark et al., 2015), and CHM (Marsh et al., 2020) focus on the realm of physically

based models. The CAPTAIN toolbox (Young et al., 2009) is a general toolkit for time series analysis. Machine learning frameworks such as scikit-learn (Pedregosa et al., 2011) and PyTorch (Paszke et al., 2019) can be used to construct data driven models.

In this paper, we focus on flexible frameworks intended for conceptual hydrological modeling. Examples of such frameworks include FUSE (Clark et al., 2008), SUPERFLEX (Fenicia et al., 2011), CMF (Kraft et al., 2011), PERSiST (Futter et al., 2014), ECHSE (Kneis, 2015), MARRMoT (Knoben et al., 2019), and RAVEN (Craig et al., 2020).

When discussing a mathematical model, it is relevant to distinguish its conceptual principles from its software implementation. In the hydrological literature, modelling concepts and their software implementation have been presented both jointly and separately. For example, the original FUSE publication (Clark et al., 2008) introduced the modelling concepts, while subsequent work (Vitolo et al., 2016) provided an R implementation. The original SUPERFLEX publications  presented the modelling principles (Fenicia et al., 2011) and demonstrated its capabilities (Kavetski and Fenicia, 2011); while Fortran and Matlab  implementations were developed as part of research work (e.g., David et al., 2019), these implementations have not been published or made available as standalone products. In contrast, some models, (e.g., MARRMoT) have been presented with a publication describing both the theoretical principles and the software implementation.

A software implementation should fulfill the intended goals of the flexible framework, in particular supporting the envisaged flexibility in terms of processes representation, spatial distribution, numerical solution methods, etc. The software implementation should also be accessible to users in terms of ease of installation, operation, eventual extension, etc. Existing frameworks approach these conceptual and practical requirements with different priorities, e.g., focusing on selected modelling objectives (e.g., model mimicry) and/or limiting the range of applications (e.g., only to lumped setups), in order to simplify the model formulation and operation.

In terms of application scope of a flexible framework for conceptual hydrological modeling, we focus on the following "realms":

1. Lumped models;
2. Distributed setups, including simulation of sub-catchments and flows/processes at internal points;
3. Ability to reproduce existing models, when necessary.
4. Support or extendibility for future applications, e.g. substance transport modelling, including water isotopes, pesticides, etc.;

In terms of software implementation, we consider the following practical criteria:

1. Ease of use, including installation, learning, and operation. Interoperability with external software, for example for model calibration and uncertainty analysis, is of obvious relevance because hydrological models are often used as parts of larger-scale projects and operations.

    2. Ease of modification and extension. Even a comprehensive software implementation will eventually require extension. For example, a modeling framework intended to simulate

streamflow may require extension to simulate water chemistry. Another type of modification might be a switch to a numerical implementation better suited for parallel computing, etc.

    3. Computational efficiency. Hydrological model applications, especially including calibration and uncertainty quantification, may require thousands or even millions of model runs.

    4. Connection to the ecosystem of modern online tools to facilitate model usability by both

researchers and practitioners. This includes online documentation (with examples and demos), and automatic workflows for unit testing, continuous integration and deployment.

These criteria are challenging to meet simultaneously. Hence, implementing a flexible framework entails juggling multiple obvious and less obvious tradeoffs. For example, the intended flexibility of a framework may come at the expense of ease of use, similar to how computer languages have varying degrees of

185 abstraction from the hardware behavior. Implementing a practical flexible framework therefore requires careful code design, experimentation, and inevitably, some compromises.

This work pursues the flexible framework objectives defined above by building upon the concept of SUPERFLEX (Fenicia et al., 2011; Kavetski and Fenicia, 2011; Fenicia et al., 2014; Fenicia et al., 2016). A key attractive feature of SUPERFLEX as a modelling concept is the fine "granularity", i.e., the degree

of flexibility, of model structures it can support, which enables systematic and detailed hypothesis testing (Fenicia et al., 2011). For example, the hydrologist should have the ability to select and combine individual model elements (e.g., reservoirs, lag functions, etc.), as well as to build customized elements.

The development of the proposed framework capitalizes on the authors' collective experience in hydrological model design and application. The original Fortran implementation of SUPERFLEX,

hereafter referred to as SUPERFLEX-F90, has been used in a series of case studies over the last decade, ranging from lumped model implementations (e.g., Kavetski and Fenicia, 2011; Fenicia et al., 2014), to distributed setups (e.g. Fenicia et al., 2016; Dal Molin et al., 2020), interpretation in the context of fieldwork insights (e.g., Wrede et al., 2015), large scale model intercomparisons (e.g., van Esse et al., 2013), and the inclusion of pesticide/substance transport (e.g. Ammann et al., 2020). The earlier Flex

framework was used in studies exploring the use of multivariate data to refine the model structure (e.g., Fenicia et al., 2006, 2008). The modelling framework FUSE was used for a range of experiments in process representation (e.g., Clark et al., 2011b), data analysis (e.g., Henn et al., 2018), and numerical

solution (e.g., Clark and Kavetski, 2010; Kavetski and Clark, 2010). The SUMMA framework represented an application of flexible modelling principles to physically based modelling. These applications have highlighted the versatility of the SUPERFLEX principles, and of flexible modelling approaches in general, to solve increasingly complex modelling problems – but have also highlighted implementation choices that limit the effectiveness and range of application of current software (e.g., the usage of a "master template" from which specific model structures are derived). This work provides a new implementation of SUPERFLEX that addresses many of these limitations.

## 1.3  Aims

This paper introduces SuperflexPy, which is a new open-source Python software implementation of the SUPERFLEX principles for conceptual hydrological model development. Particular attention is given to the challenges of implementing a framework that achieves the flexibility envisaged by SUPERFLEX and flexible frameworks in general. Our objectives are as follows:

1. Present SuperflexPy and its basic building blocks (*components*): *elements*, *units*, *nodes,* and *network*;
2. Illustrate how SuperflexPy can help hydrologists implement a conceptual model structure at the desired level of internal complexity and spatial resolution – including recreating existing models and developing new models;
3. Provide a broad discussion of the hydrological modelling software implementation challenges and of how SuperflexPy contributes to the toolkits available to the hydrological community.

The paper is organized as follows. Section 2 describes the SuperflexPy architecture and building blocks, and provides a short demo (aims 1 and 2). Section 3 illustrates selected applications of the framework, including the setup of SUPERFLEX configurations used in earlier case studies and the use SuperflexPy to create new *elements* (aim 2). Section 4 provides more technical SuperflexPy details, useful for understanding the usage and general potential of the framework (aim 1). Section 5 discusses SUPERFLEX design choices in the context of existing flexible frameworks, including current limitations and future developments (aim 3). Finally, Section 6 provides a brief overall summary and conclusions.

The examples presented in the paper are generally intended to provide the intuition and reasoning behind SuperflexPy. The model documentation provides detailed information and use instructions. The documentation is available and maintained online (refer to "code availability" section); references from the paper to the documentation point to the static PDF version provided as supplementary material to this paper.

## 2  Description of SuperflexPy

**2.1  General organization**

The SuperflexPy framework has a hierarchical organization with four nested levels: "*element*", "*unit*", "*node*", and "*network*", collectively referred as "*components*". These *components* are shown in Figure 1 and described below. Further practical details are provided in Chapter 4 of the supplementary material:

1.  *Element* (Figure 1a). This level represents the basic model building block and is used to create

reservoirs, lag functions, and connections. An *element* can be used to represent an entire catchment, or, more commonly, a specific hydrological process or response mechanism within the catchment.

   The **reservoir** element is used to conceptualize processes involving the storage and release of water and other fluxes. It is described mathematically by ordinary differential equations (ODEs),

$$\frac{d\mathbf{S}(t)}{dt} = \mathbf{g}_S\big(\mathbf{S}(t), \mathbf{X}(t); \boldsymbol{\theta}\big) \tag{1}$$

$$\mathbf{Y}(t) = \mathbf{g}_Y\big(\mathbf{S}(t), \mathbf{X}(t); \boldsymbol{\theta}\big) \tag{2}$$

   where $\mathbf{s}$ are the state variables (e.g., water storages), $\mathbf{X}$ are the inputs (e.g., precipitation), $\mathbf{Y}$ are the outputs (e.g., streamflow), and $\mathbf{g}_s$ and $\mathbf{g}_Y$ are specified constitutive functions (e.g., storage-discharge relationships).

In most conceptual models, reservoir elements have a single state variable (representing water storage); multiple state variables can be accommodated if necessary (e.g., to keep track of snow and liquid water separately). Mathematically, a multistate reservoir can be represented by a system of differential equations of the form of equations (1) and (2).

   The solution of equation (1) is usually obtained numerically using external numerical procedures

referred to as "numerical approximators" (see Section 4.3).

   The **lag function** element is used to represent delays in the transmission of the fluxes (e.g., routing). It is described mathematically by a convolution integral,

$$\mathbf{Y}(t) = \mathbf{X}(t) * \mathbf{g}_H(t; \boldsymbol{\theta}) = \int_0^T \mathbf{X}(t - \tau)\, \mathbf{g}_H(\tau; \boldsymbol{\theta})\, d\tau \tag{3}$$

   where $*$ denotes the convolution operator, $\mathbf{X}$ is the input (e.g., water flux), $\mathbf{g}_H$ is the impulse

response function, and $T$ is the time of influence of $\mathbf{g}_H$ (i.e. the maximum lag).

   There is a general mathematical correspondence between reservoirs and lag functions (e.g., Nash, 1957). SuperflexPy users can select the element specification best suited to their specific

context.

The **connection** element is used to connect two or more *elements* whenever a direct connection is not possible. For example, connection elements are used when a flux needs to be split among multiple *elements* downstream (splitter), or, vice versa, when multiple fluxes need to be aggregated (junction). A particular type of connection is represented by the "transparent" element, which simply outputs the same fluxes it receives as inputs, and is used to facilitate the connection between elements (see description of *unit* below).

All connection elements are stateless and can be represented mathematically as follows,

$$\mathbf{Y}(t) = \mathbf{g}_{\mathrm{C}}\big(\mathbf{X}(t); \boldsymbol{\theta}\big) \tag{4}$$

where $\mathbf{g}_{\mathrm{C}}$ describes the connectivity between input fluxes and output fluxes, and $\boldsymbol{\theta}$ represents connectivity parameters (if any).

2. ***Unit*** (Figure 1b). A *unit* is a collection of multiple connected *elements*, and is generally intended to implement a lumped catchment model or an HRU in a distributed model. Multiple reservoir and lag function elements within a *unit* can be connected to each other, either directly (one-to-one connections), or using connection elements such as splitters and junctions (when a single *element* is connected to multiple *elements*). The multiple *elements* within a *unit* are arranged in *layers,* with the following restrictions: (i) feedback loops *between* the *elements* are not allowed and (ii) *elements* can be connected only if they belong to two consecutive *layers*. Fluxes between elements in nonconsecutive layers are passed using transparent elements. The concept of *layers* will be elaborated and illustrated in Section 5.1.1; see also Section 4.2 of the supplementary material. In technical terms, the structure formed by the *elements* must be a directional acyclic graph (DAG). The motivation and implications of these design choices on model generality and computational efficiency are elaborated in Sections 5.1.1 and 5.2.

3. ***Node*** (Figure 1c). A *node* is a collection of multiple *units* that operate in parallel. In the context of distributed models, the *node* can be used to represent a single catchment and the *units* can be used to represent multiple landscape elements or HRUs within the catchment. Each *unit* within a *node* is characterized by a weight, which typically represents its area fraction or, more generally, its contribution to the total outflow of the *node*. The weights are used to combine the output fluxes from the *units* into the total output flux of the *node*. Another important attribute of a *node* is its "area", which is used when multiple *nodes* are combined into a *network* (see below).

4. ***Network*** (Figure 1d). A *network* connects multiple *nodes* into a tree structure, and is typically intended to develop a distributed model that generates predictions at internal sub-catchment locations (e.g. to reflect a nested catchment setup). The *network* routes the fluxes from upstream

*nodes* (leaves of the tree) to the final downstream *node* (root of the tree). Routing delays in the river network can be simulated by feeding *node* outputs into lag function elements. The area of each *node* is used to determine its contribution to the total outflow of the *network*. Only a single network can be used in a given SuperflexPy model.

The hierarchical organization of SuperflexPy makes the effort required to configure it to a new problem proportional to the problem complexity. In particular, many common model setups can be constructed without necessarily using all levels listed above, thus reducing configuration effort. Some representative examples are given below:

- Level 1 is sufficient to create single-*element* models, e.g., a single-reservoir model or a unit
hydrograph model (e.g. Kirchner, 2009);
- Level 2 is sufficient to create a lumped model structure, such as GR4J (Perrin et al., 2003) or Hymod (Boyle, 2001);
- Level 3 is sufficient create a distributed model that represents spatial heterogeneity but generates predictions only at the catchment outlet (e.g. Beven and Kirkby, 1979; Gao et al., 2014; Nijzink
et al., 2016);
- Level 4 is needed only in models that generate predictions at interior points, such as SWAT (Arnold et al., 2012), GEOframe-NewAge (Formetta et al., 2014), and distributed SUPERFLEX applications (e.g. Fenicia et al., 2016; Dal Molin et al., 2020).

Examples of SuperflexPy models implemented at Levels 2 and 4 are given later in Section 3. Note that
the association of specific SuperflexPy components to specific hydrological entities, e.g., the use of *units* for HRUs and *nodes* for sub-catchments, is not intended as a rigid prescription. Other association choices may be favored by the modeler depending on the required model structure and spatial connectivity.

The clarity of visual model representation is particularly important in flexible frameworks because they can generate many subtly different configurations (e.g., Bancheri et al., 2019). The model schematics in
this paper indicate explicitly every element, including reservoirs, lag functions, and junctions (e.g., Figure 1).

From a software design prospective, SuperflexPy embraces the object-oriented paradigm (e.g., Meyer, 1988). All framework *components* are represented by objects that can operate either alone or together, interacting with each other and with external libraries (e.g. for calibration) through defined interfaces.
More details are provided in Section 4.2.

All SuperflexPy *components* have states and/or parameters, which are controlled programmatically using dedicated methods (refer to Section 4.1).

## 2.2 A simple illustration of SuperflexPy: creating a new model from existing components

This section illustrates the key steps needed to configure and run a hydrological model using the SuperflexPy framework. The illustration presents a distributed model intended to represent a catchment with 2 HRUs and 3 sub-catchments. The model structure is shown in Figure 1d. The catchment is represented using a *network*, the sub-catchments are represented using *nodes*, and the HRUs are represented using *units*. Two distinct HRU-specific model structures are specified, and are implemented using *elements*. The corresponding SuperflexPy code is shown in Figure 2. An extended version of this demo is provided in Section 6.5 of the supplementary material.

In this example, an implementation of the necessary *elements* with SuperflexPy already exists; therefore, the *elements* only need to be imported. The case where the model structure requires *elements* for which an implementation is not yet available is considered in Section 2.3. More complex setups are described in Section 3 and in the supplementary material.

We start by importing the model *components* required by the model structure, namely the *elements* (`LinearReservoir` and `HalfTriangularLag`), *unit*, *node*, and *network*. The numerical approximator `ImplicitEulerPython` and root finder `PegasusPython` needed to solve the ODEs associated with the reservoir elements are also imported (see Section 4.3 for details). The import operation is shown in Lines 1-7.

The imported *components* are then initialized, which entails specifying the model structure (connectivity between model *components*) and the initial values of parameters and states. The initialization sequence starts with the numerical procedures (Lines 10-11) and proceeds from the lowest-level *components* (*elements*) to the highest-level *component* (*network*).

Specifically:

L1. An *element* is initialized by specifying its parameters, states, and, where relevant, the numerical solver (Lines 14-16). Each element is given an identifier (`id`) for subsequent use, as shown on Line 23.

L2. A *unit* is initialized by specifying the *elements* that compose it and the identifier (Lines 19-20). As noted earlier in Section 2.1, the connectivity between *elements* is defined by conceptualizing the *unit* as a succession of *layers* that contain the *elements*. More complex examples are given in Section 3. The parameters and states of *elements* can be changed after initialization using the methods `set_parameters` and `set_states` of the containing units. This operation is shown on Line 23 for the `LinearReservoir` element.

L3. A *node* is initialized by specifying the *units* that compose it, their contribution (weight) to the *node* output, the influence area of the *node* (here, the area of the sub-catchment), and the identifier (Lines 26-28).

L4. The *network* is initialized by specifying the *nodes* that compose it and their connectivity, called `topology` (Line 31). The connectivity is defined indicating, for each *node*, the *node* downstream of it. A network identifier is not specified (as only a single network can be used).

The next step is to set the model inputs and time step. Lines 34-36 show how the inputs are assigned directly to the *nodes*, enabling the model to receive spatially varying rainfall and PET. The time step is set on Line 39 (variable time steps are also supported, see Section 4.5.1 of the supplementary material).

The model can now be run by calling the `get_output` method of the highest-level *component*, as shown on Line 42.

Note that all input quantities provided to SuperflexPy, including fluxes, time step length, parameters, states, areas, etc., must have consistent units. To reduce model code complexity and execution overhead, we take the perspective that unit checks represent pre-processing and are best handled by the user according to their own preferences and standards. Output fluxes have the same (assumed) units as input fluxes, e.g., if precipitation is in mm/h, then streamflow is also in mm/h, etc.

## 2.3  Creating new model *components* with SuperflexPy

We now consider the case where the intended model structure has *components* beyond those already available in SuperflexPy.

New model *components* can be created by extending existing SuperflexPy *components*. To this end, SuperflexPy provides a library of built-in high-level *components* that can be extended to achieve the desired functionality. We anticipate that the SuperflexPy *components* most likely to require extension are the *elements*, where new constitutive functions may be required in reservoir elements and new weight functions may be required in lag function elements. In contrast, it is less likely that *unit*, *node,* and *network* functionalities would require extension.

The extension of existing SuperflexPy *elements* takes advantage of the object-oriented paradigm underlying the SuperflexPy software design. The inheritance principle, one of the core concepts of the object-oriented paradigm, allows the user to construct new *components* by "inheriting" most of the functionalities (methods) from existing classes. Separate implementation is then required only for methods where the new model differences are to be introduced. This approach reduces substantially the amount of coding required to implement a new model *component*.

A detailed example of this procedure is given in Section 3.2, which shows how to implement a reservoir with a new storage-discharge relationship. More examples are provided in Chapters 8 and 9 of the supplementary material.

## 3    Examples of building hydrological models using SuperflexPy

This section provides more detailed examples of using SuperflexPy to implement hydrological models, including the use of built-in *elements* and the creation of new *elements*. We follow a progression from simple to complex. Section 3.1 shows the implementation of model M4, a lumped model built solely from reservoir elements and used in the original SUPERFLEX case study (Kavetski and Fenicia, 2011). Section 3.2 shows how to define a new *element* with a different storage-discharge relationship for one of the reservoirs of M4. Section 3.3 shows the implementation of a distributed model from a recent application of SUPERFLEX in the Thur catchment (Dal Molin et al., 2020).

Compared to the demo in Section 2.2, which was intended to give a general sense of model building with SuperflexPy, the examples in this section represent "realistic" applications of SuperflexPy, including setting up a spatially distributed model with multiple HRUs and more complex model structure. Further technical details and additional examples, including the implementation of popular conceptual models (e.g., GR4J, HYMOD), are provided in the supplementary material (chapters 8-11).

### 3.1    Implementing SUPERFLEX configuration M4

M4 is a simple lumped model presented in Kavetski and Fenicia (2011). As shown in Figure 3, M4 comprises two reservoirs connected in series: an "unsaturated" reservoir (UR) intended to represent the partitioning of precipitation between evaporation and runoff, and a "fast" reservoir (FR) intended to represent subsequent streamflow generation mechanisms.

UR partitions precipitation $P^{(\text{UR})}$ into a portion that enters the UR storage and eventually evaporates through flux $E_{\text{A}}^{(\text{UR})}$, and a portion $Q^{(\text{UR})}$ that is directed to the downstream FR reservoir:

$$\frac{\mathrm{d}S^{(\text{UR})}}{\mathrm{d}t} = P^{(\text{UR})} - E_{\text{A}}^{(\text{UR})} - Q^{(\text{UR})} \tag{5}$$

where

$$\overline{S}^{(\text{UR})} = \frac{S^{(\text{UR})}}{S_{\text{max}}^{(\text{UR})}} \tag{6}$$

$$Q^{(\text{UR})} = P^{(\text{UR})} \times \left(\overline{S}^{(\text{UR})}\right)^{\beta^{(\text{UR})}} \tag{7}$$

$$E_{\mathrm{A}}^{(\mathrm{UR})} = E_{\mathrm{P}}^{(\mathrm{UR})} \times \frac{\overline{S}^{(\mathrm{UR})}\left(1 + m^{(\mathrm{UR})}\right)}{\overline{S}^{(\mathrm{UR})} + m^{(\mathrm{UR})}} \tag{8}$$

In equations (6)-(8), $S_{\max}^{(\mathrm{UR})}$ and $\beta^{(\mathrm{UR})}$ are model parameters. The quantity $m^{(\mathrm{UR})}$ is used to approximate a "smooth" threshold behavior; we typically fix $m^{(\mathrm{UR})} = 0.01$.

FR is a power-law reservoir,

$$\frac{\mathrm{d}S^{(\mathrm{FR})}}{\mathrm{d}t} = P^{(\mathrm{FR})} - Q^{(\mathrm{FR})} \tag{9}$$

with the storage-discharge relationship given by

$$Q^{(\mathrm{FR})} = k^{(\mathrm{FR})}\left(S^{(\mathrm{FR})}\right)^{\alpha^{(\mathrm{FR})}} \tag{10}$$

where $k^{(\mathrm{FR})}$ and $\alpha^{(\mathrm{FR})}$ are model parameters.

The inflow $P^{(\mathrm{FR})}$ is given by the outflow from UR, i.e., $P^{(\mathrm{FR})} = Q^{(\mathrm{UR})}$.

M4 is a lumped model with multiple *elements*, and hence can be implemented using SuperflexPy levels L1 and L2 (*element* and *unit*, see Section 2.1). Figure 4 shows the code needed to implement M4. The numerical procedures are imported and initialized on Lines 1-2 and 7-8 respectively. Similar to the model described in Section 2.2, the two model *elements* (UR and FR) are already implemented. Hence, the user

only needs to import the elements (Lines 1-3) and initialize their parameters (Lines 7-13). Next, the *unit* is imported (Line 4) and initialized to contain the two reservoirs (Line 15). The model configuration is then complete.

The loading of input data from text file(s), databases, etc. is separate from the configuration of SuperflexPy, and can be carried out using any suitable Python library or function. In this example, we use

Numpy to read time series of precipitation and PET from a text file, as shown in Lines 17-18. The corresponding SuperflexPy inputs are set using these Numpy arrays, as shown on Line 20. Further practical details on input-output are provided in Section 4.5.5 of the supplementary material.

The model can now be run with the given input data to produce the model outputs, as shown on Line 23. The outputs contain streamflow time series in the form of Numpy arrays.

**3.2   Changing the equations of the fast reservoir in M4**

Suppose the modeler wishes to modify model M4 by changing the storage-discharge equation of the fast reservoir given in equation (10) to a new relationship

$$Q^{(\mathrm{FR})} = \frac{k^{(\mathrm{FR})} \left( S^{(\mathrm{FR})} \right)^{\alpha^{(\mathrm{FR})}}}{S^{(\mathrm{FR})} + b^{(\mathrm{FR})}} \tag{11}$$

where $k^{(\mathrm{FR})}$, $\alpha^{(\mathrm{FR})}$, and $b^{(\mathrm{FR})}$ are model parameters.

An *element* with this storage-discharge relationship has not been implemented in SuperflexPy yet (as of version 1.3.0). The following sections give two approaches for creating such an *element*.

### 3.2.1   General approach for creating a new reservoir with SuperflexPy

The general approach for creating a new reservoir in SuperflexPy is to define a new class that inherits most of its functionality (methods) from the class `ODEsElement`. This operation is illustrated in the
code snippet in Figure 5 (see Section 8.1 of the supplementary material for full details). The new class must override the following methods:

- `__init__`: constructor of the class. Its main purpose is to invoke the constructor of the parent class (Lines 5-6) and to point to the method used to calculate the fluxes, here, `_fluxes_function_python` (see also Section 4.3, which illustrates the efficiency
benefits of using Numba-optimized methods for calculating the fluxes);
- `set_input`: takes the input fluxes in a predefined order (here, just precipitation) and assigns them a key (Line 15) that is then used when setting up and solving the model equations;
- `get_output`: invokes the functionalities implemented by the `ODEsElement` to solve the *element* equation over the entire simulation (all time steps). Lines 20-22 get the current state of
the reservoir, invoke the ODE solver, and set the state to its final value. Lines 24-28 get the output flux arrays from the numerical approximator (see Section 4.3). Line 30 returns a list with the output of the *element* (here, the streamflow);
- `_fluxes_function_python`: calculates the fluxes and (optionally) their derivatives with respect to the state for a given state, inputs, and parameters. Line 36 implements the vector version
while Lines 38-41 implements the scalar version. Both versions are needed by the numerical approximator (see Section 4.3; further practical details are provided in Section 8.1 of the supplementary material).

The new *element* `NewFastReservoir` is now defined and can be used in the "new" version of M4, in lieu of the previous *element* `PowerReservoir`. The Object-Oriented features of Python are very
useful here to enable the new class `NewFastReservoir` to inherit most of the methods from the base class `ODEsElement`. Otherwise, in addition to the methods listed above, we would have needed to

implement many other methods, e.g., for interfacing with numerical solvers, for setting *element* parameters and states, etc.

### 3.2.2 Simplified approach for creating a new reservoir element (from an existing element)

The same new reservoir element can be implemented in a simpler way by noting that `NewFastReservoir` differs from `PowerReservoir` solely in the definition of the outflow equation. This difference affects only one of the four methods implemented in Figure 5, namely `_fluxes_function_python`. A simpler implementation of `NewFastReservoir` can be therefore achieved by inheriting this class directly from class `PowerReservoir` rather than from class

`ODEsElement`. The code in Figure 6 illustrates this approach and implements only the method `_fluxes_function_python`. All other methods are inherited from class `PowerReservoir`.

Note that this simplified implementation is a consequence of the required modification being relatively minor, i.e., a change solely in the constitutive function equation. More complex modifications, such as the inclusion/exclusion of input/output fluxes (e.g. inclusion of evapotranspiration into the

485 `PowerReservoir`), would require the general implementation approach described in Section 3.2.1.

### 3.3 Implementing a distributed model

This section illustrates the implementation of an HRU-based, distributed hydrological model, intended to simulate streamflow in a nested catchment. This implementation requires the entire workflow illustrated in Section 2.2. The example is provided by model M02, developed in Dal Molin et al. (2020) to provide

streamflow predictions at 10 sub-catchments of the Thur catchment in Switzerland (Figure 7a).

Each sub-catchment receives its own forcing, namely precipitation, potential evapotranspiration, and temperature. Two HRU types are defined based on geology: consolidated and unconsolidated formations (Figure 7b). Both HRU types are characterized by the same model structure, which is shown in Figure 8. This HRU model structure differs from model structure M4 (section 3.1) in the following additional

495 *elements*: (i) a "snow" reservoir, WR, which controls the partition of incoming precipitation between rainfall and snowfall based on temperature, (ii) a lag function between UR and FR, and (iii) a "slow" reservoir, SR, which acts in parallel to FR and is controlled by the same equations as FR but with different parameter values.

Similar to the simpler previous example in Section 3.1, this "lumped" model structure is implemented as

a *unit*. However, a key difference is that in the previous example the *unit* represented the entire system, whereas here it is part of a more complex system.

Given the spatial organization of the model, *nodes* are used to represent sub-catchments and *units* are used to implement HRU types. Note that the sub-catchments may share (one or more) HRU types, which in SuperflexPy translates into the *nodes* sharing (one or more) *units*. The *network* level is used to connect multiple *nodes*, and enables predictions at internal catchment locations. Figure 10 shows the SuperflexPy representation of the spatial organization shown in Figure 7.

We start by implementing the *units*. As seen in Figure 8, the HRU model structure has *elements* operating in parallel and, therefore, requires the use of connections. Figure 9 shows how the HRU model structure is "translated" into a SuperflexPy *unit*. Recall, from Section 2.1, that *elements* can be connected only if they belong to two consecutive *layers*, which implies that "gaps" in the structure must be filled using *transparent* elements, which output the same fluxes they receive as inputs. Splitters and junctions are used to divide and merge the fluxes to implement the parallel flow paths.

Comparing Figure 8 with Figure 9, we see how the HRUs structure has been implemented within SuperflexPy. The following implementation aspects are noted:

1. The incoming precipitation is partitioned into rainfall and snowfall. This partitioning is done internally in the WR *element*. The SuperflexPy implementation of WR takes care of two processes: (i) partitioning of precipitation into rainfall and snowfall; and (ii) simulation of snow processes (accumulation and melting). The output of WR is, logically, the sum of rainfall and snowmelt. Alternatively, a (new) splitter element could have been defined to partition the fluxes between UR (rainfall) and WR (snowfall) based on temperature.

2. WR, as currently implemented, does not receive as input the potential evapotranspiration (PET), which is needed by the downstream *element* UR. Therefore, the transfer of PET values to the UR element is implemented using a separate path composed by three *elements*, labelled "upper splitter", "upper transparent", and "upper junction" (Figure 9). This choice simplifies the interface of element WR at the expense of a somewhat more complicated model structure with additional elements.

3. The parallel part of the structure is composed by two *elements* on one branch (lag and FR) and only one *element* on the other branch (SR). To satisfy the requirement of not having "gaps" in the *unit* structure, a transparent element ("lower transparent") is added after the SR.

The code to setup this model is detailed in Figure 11. Similar to the earlier example in Section 2.2, the user initializes and connects all model *components*, proceeding sequentially from the lowest level (*elements*) to the highest level (*network*). The procedure can be summarized as follows:

1. Lines 10-29: Initialize the *elements* needed for the lumped model structures used in the HRUs;
2. Lines 32-39: Initialize the units used to represent the HRUs, linking all the *elements*;

3. Lines 42-51: Initialize the nodes used to represent the sub-catchments. Both *units* are assigned to 9 *nodes*; the Mosnang sub-catchment contains a single HRU and hence only a single *unit* is assigned to the corresponding *node* (Line 49).

4. Lines 54-60: Connect the *nodes* using a *network*. The topology of the *network* is defined by indicating, for each node, the downstream one.

The *network* runs the *nodes* from upstream to downstream, collects their outputs, and routes them to the outlet. Customized routing functions can be implemented, as shown in Section 9.1 of the supplementary material. The output of the *network* is a Python dictionary, with keys given by the node identifiers and values given by the list of Numpy arrays representing the time series of output fluxes over the simulation period.

# 4 Implementation details of SuperflexPy

This section presents additional technical details of SuperflexPy needed to understand better some aspects of the functioning of the framework. A more detailed and practical description is provided in the supplementary material.

## 4.1 Parameters and states

All SuperflexPy *components* can have parameters and states. Parameters specify *component* characteristics, whereas states keep track of the *component* history. States and parameters are set as part of initializing the model *components*, and can be manipulated using `get` and `set` methods provided by the framework at all levels of its hierarchy (see the example in Section 2.2).

The parameters can be either constant or variable in time. Constant parameters represent the most common set up of hydrological models. In conceptual hydrological modelling, time-varying parameters have been proposed to represent "deterministic" system variability (e.g. seasonality, Westra et al., 2014) and/or "stochastic" system variability (e.g., Kuczera et al., 2006; Reichert and Mieleitner, 2009; Renard et al., 2011); see also earlier work in data-based mechanistic modelling (e.g., Young, 2000).

## 4.2 Modular design following the Object-Oriented paradigm

As noted in Section 2.1, SuperflexPy embraces the object-oriented paradigm (e.g. Meyer, 1988), which is widely used in general software and is increasingly adopted in scientific software.

Figure 12 shows the unified modeling language (UML) class diagram of SuperflexPy. The schematic illustrates the classes underlying the core framework (i.e., the base classes that define SuperflexPy architecture), but excludes, for simplicity, the specific implementations of *components* and numerical

routines. All the classes in the diagram can be extended to implement customized *components*; for example, a reservoir can be implemented by extending the class `ODEsElement`, a splitter can be implemented by extending the class `ParameterizedElement`, a node with a particular routing mechanism can be implemented by extending the class `Node`, etc.

The object-oriented design provides several advantages in the context of SuperflexPy:

- The inheritance principle enables the creation of new classes by extending existing ones. Inheritance reduces drastically the amount of new code that needs to be generated to implement a new model *component* (see example in Section 3.2);
- Changes to a class (e.g. a *component*) and the creation of new classes can be carried out in isolation from the rest of the code, as long as the interfaces between classes are respected;

- When creating a model, only the necessary objects need to be initialized and used. This principle makes the model configuration effort roughly proportional to required model complexity, i.e., simple model structures can be constructed from the minimal set of required components;
- Objects retain their history (states), which can be accessed post-run to undertake model analysis and/or subsequent computation;

- The modular nature of objects facilitates the development and testing of new code.

These benefits make it easier to achieve clean and maintainable code, which is essential for any practical modelling framework.

### 4.3    Numerical solution of ODEs

The mass balance of reservoir elements is described using ordinary differential equations (ODEs), which
are typically solved (approximately) using numerical time-stepping algorithms. Many such algorithms have been described in the numerical methods literature, e.g. Euler methods, Runge-Kutta methods, etc. (e.g., Butcher and Goodwin, 2008).

SuperflexPy separates the formulation of model equations from the solution of these equations. Specifically, flux equations are defined internally as methods of the *elements* (as shown in Section 3.2),
while the numerical algorithm to solve the ODEs is specified externally to the *element*, creating a class specific to this task. The separation of equations and solvers in the model specification enables the modeler, within some restrictions, to select the numerical method without making any changes to the governing model equations (see section 5.2 of the supplementary material). That said, given SuperflexPy primary emphasis on enabling hydrologists to experiment with flexible conceptual model structures,
numerical flexibility is given a relatively lower level of priority and the choice of numerical architecture of the framework is largely driven by findings of previous studies (see below).

SuperflexPy conceptualizes the solution of its mass balance ODEs as a two-step process: (1) construct a discrete-time numerical approximation of the ODEs (e.g., using Euler time stepping schemes), and (2) when an implicit time stepping scheme is used, solve the associated nonlinear algebraic equation(s). The procedures used for these tasks are referred to as the "numerical approximator" and the "root finder", respectively. This distinction helps achieve better software modularization, disentangling the choice of the numerical approximator and of the root finder.

Currently, SuperflexPy provides three built-in numerical approximators, namely the fixed-step implicit and explicit Euler time stepping schemes (e.g., Clark and Kavetski, 2010) and Runge Kutta 4. Two built-in root finders are provided, namely the Pegasus algorithm (Dowell and Jarratt, 1972) and a hybrid Newton-bisection algorithm (Press et al., 1992). Additional numerical routines are currently being developed. To avoid mass balance discontinuities, as well as to ensure better numerical stability and faster convergence, we recommend using smooth flux functions (e.g., Kavetski and Kuczera, 2007).

An additional approximation is employed within SuperflexPy, namely that all model fluxes are constant within the model time step. This approximation is consistent with the typical format of hydrological data, such as rainfall, PET, etc, which are tabulated in discrete steps (e.g., daily, hourly, etc), but is applied not only to the forcing data but also to all internal fluxes. As such, this pragmatic approximation enables a further simplification of the solution procedure, because the output flux from each element becomes a scalar value (per time step). Note that first order time stepping schemes, which we recommend for SuperflexPy, themselves make exactly the same assumption and are hence not impacted. However, higher order time stepping schemes and adaptive substepping schemes would be impacted by additional first-order discretization error, because the variation of internal fluxes within the model time step is ignored. Further details about this pragmatic approximation are provided in section 5.2 of the supplementary material.

The user can implement additional numerical algorithms, either by coding them directly or by interfacing with external code (e.g. ODE solvers from SciPy). Detailed instructions are provided in section 5.1 of the supplementary material, which also includes a description of how to implement a numerical solver "from scratch", bypassing of the current numerical approximator / root finder architecture.

As detailed next in Section 4.4, the choice of numerical implementation, and its compatibility with optimizing compilers, may have a strong impact on the overall computational speed of the model.

## 4.4   Computational efficiency and language choice

Computational efficiency is a key requirement of a practical modelling framework. Model calibration via parameter optimization is a common computationally demanding task required by most hydrological

models, typically requiring hundreds or thousands of model runs. Moreover, conceptual hydrological models are often used in Monte Carlo uncertainty quantification, with comparable or even larger computational cost (up to millions of model runs in some cases).

The choice of programming language inevitably requires trade-offs between computational efficiency and ease of use. The choice of Python for SuperflexPy was motivated by the attraction of a flexible and widely used scripting language in conjunction with two efficient numerical libraries: Numpy (Walt et al., 2011) and Numba (Lam et al., 2015). Numpy provides highly efficient arrays for vectorized operations (i.e. elementwise operations between arrays). Numba provides a "just-in-time compiler" that compiles (at runtime) a Python method into machine code that interacts efficiently with Numpy arrays.

The combined use of Numpy and Numba is particularly effective when solving ODEs, where the numerical algorithm performs element-wise sequential operations. The built-in SuperflexPy approaches for solving ODEs are compatible with such numerical infrastructure, and therefore enable fast computation times. Note that switching to ODEs solvers that do not take advantage of such libraries might dramatically increase the model runtime.

Numba offers drastic computational speed ups compared to native Python; our experimentation suggests runtime reductions by factors of up to 30. However, a drawback of Numba is the requirement to compile the code each time it is executed (run). For a lumped model composed of a few reservoirs, the Numba compilation time is of the order of a few seconds. Therefore, Numba will outperform Python when the simulation is long (e.g. multiple years of hourly data) and/or when the model needs to be run a large number of times. For example, as a broad illustration of runtimes on a standard laptop, calibration of a HYMOD-like SuperflexPy model to observed daily data, requiring 1000's of model runs each with 1000 time steps, takes a few seconds with the Numba implementation compared to a couple of minutes with native Python execution. Note that here we refer to the runtime of the SuperflexPy model itself, and exclude the runtime of the calibration tool procedures; more details on benchmarking are given in section 5.3 of the supplementary material. Examples of interoperability of SuperflexPy with external libraries for model calibration (e.g., SPOTPY, Houska et al., 2015) are given in chapter 14 of the supplementary material.

## 4.5 Ability to represent multiple fluxes and states

SuperflexPy can operate with multiple fluxes and state variables. In particular, connection elements, *units*, *nodes*, and the *network* can accommodate an arbitrarily large number of fluxes. The use of multiple fluxes has been already shown in the model structure described in section 3.3, where the `upper_splitter`

handles three different variables (precipitation, temperature, and PET). Additional examples are provided in the supplementary material (e.g., chapters 10, 11).

The capability to simulate multiple fluxes and states is intended to support the future extension of SuperflexPy to new modelling scenarios. Several such scenarios may be of interest, including the transport of chemical substances (e.g., Fenicia et al., 2010; Ammann et al., 2020), the interaction between 665 frozen and liquid water in a snow *element* (e.g., Jansen et al., 2021), interactions in the saturated/unsaturated soil zones (e.g., Seibert et al., 2003), and so forth.

While the current examples in SuperflexPy do not include all the cases listed above, the framework architecture anticipates the need for more general simulation functionality, and has been designed to support extension to accommodate such multi-state processes.

## 670 5   Discussion

### 5.1   Balancing functionality, scope, and usability in a flexible model implementation

A software implementation that maximizes flexibility and usability is challenging to achieve, because flexible modelling functionality may increase configuration effort and computational cost. Existing flexible frameworks have approached this tradeoff with different priorities, based on their respective 675 modelling objectives and paradigms.

The following sections offer a brief discussion of the design choices made by SuperflexPy in the context of selected existing frameworks with a similar scope. The discussion makes use of Table 1 and Table 2, which summarize key design choices related to usability and simulation capabilities respectively.

### 5.1.1   Structural flexibility

Structural flexibility refers to the flexibility in how *elements* can be connected to compose the structure of the model (i.e., of the *unit*, following SuperflexPy terminology). This consideration applies both to lumped and distributed models; the flexibility in specifying the spatial organization of the model is considered separately in Section 5.1.2.

Some flexible frameworks are implemented using a master structure that incorporates all supported model 685 configurations. In these implementations, the user can choose the flux equation(s) (e.g., FUSE, SUPERFLEX-F90) and/or activate/deactivate specific elements (e.g., SUPERFLEX-F90), but cannot change the overall connectivity of model elements. To the extent that the master structure is sufficiently general, it may not unduly restrict the practical usage of the framework.

Other frameworks (e.g., MARRMoT) propose a collection of existing conceptual model structures ready to use, which have been implemented following the same design rules in order to allow for a fair comparison. Such frameworks are typically intended for model intercomparison studies.

The most general frameworks allow connecting the elements freely without constraints. A distinction can be made between frameworks that allow for mutual interactions between the elements (e.g., CMF) and frameworks that do not allow such interactions (e.g., ECHSE).

SuperflexPy adopts the latter philosophy, allowing to connect the *elements* freely within the *unit* but restricting mutual interactions, i.e., constraining the structure to be a DAG (see Section 5.2). Moreover, we have chosen to define the DAG as a succession of *layers*, listing the *elements* in order from upstream to downstream and allowing for parallel flow paths (e.g., see the model structure in Figure 9). This "list" formulation has been selected in preference to other methods for defining a graph, e.g., connectivity matrix, adjacency list, etc., for the following reasons: (i) simplicity/scalability, as the list dimension scales linearly with the number of *elements*, in contrast to the connectivity matrix approach where this scaling is quadratic; (ii) arguably better readability, as the *elements* are listed in the order they appear in the DAG; and (iii) it guarantees a graph topology without loops. Note that other popular modelling tools (e.g., neural networks) adopt this type of formulation.

### 5.1.2  Spatial flexibility

Most frameworks (e.g., CMF, ECHSE, SUPERFLEX-F90, etc.) support multiple types of spatial discretization (e.g., lumped, HRUs, sub-catchments, grids, etc.). Some frameworks (e.g., FUSE, MARRMoT) support solely lumped models.

SuperflexPy uses 4 hierarchical levels of *components,* intended to facilitate the formulation of models that range in spatial complexity from a simple lumped model, to a composition of lumped models intended for prediction at a single location (e.g. a catchment with several HRUs), and ultimately to a distributed model capable of making predictions at multiple internal locations. The use of a hierarchical set of components could be contrasted to a framework based solely on the lowest level components, here, *elements*. The use of higher level components enables the modeler to capture explicitly the natural groupings in the catchment of interest, e.g., sub-catchments, HRUs, etc.

### 5.1.3  Usability

The usability of a framework can be judged according to several aspects.

The first aspect is how a framework is operated. Some frameworks are standalone and operated through a graphical interface (e.g., PERSiST) or the command line interface (e.g., SUPERFLEX-F90). Other

frameworks are designed as libraries that can be called from the user code in a specific programming language to initialize, configure and run the model (e.g., CMF, MARRMoT; SUPERFLEX-F90 also allows this option when using the source code from Fortran). SuperflexPy is implemented as a Python package. Models can be created using a Python script and interfaced easily with external libraries (examples are provided in chapter 14 of the supplementary material).

The second aspect is the scope of the framework. Most frameworks (e.g., SUPERFLEX-F90, ECHSE) adopt, by design, the philosophy of "one tool per problem" and limit their functionality to the simulation of hydrological processes. Other frameworks integrate tools for parameter calibration and sensitivity analysis, uncertainty quantification, pre- and post-processing tasks such as input unit checks/conversions, etc. (e.g., RAVEN, PERSiST). SuperflexPy adopts the first philosophy: it limits its functionality to hydrological simulation.

Finally, documentation is another key aspect in the usability of a framework. Virtually all considered frameworks provide such documentation to a varying degree of detail. SuperflexPy documentation is available online and explains in detail how to use and further develop the framework.

Figure 13 illustrates the online software management tools that are used to develop and deploy SuperflexPy. The framework itself, including source code, documentation, examples, etc., is hosted on GitHub. Automated workflows (dashed lines in the figure) are then used to create new releases (PyPI), get DOIs for the software releases (Zenodo), host the documentation (ReadTheDocs), and create runnable examples (hosted on Binder as Jupyter notebooks). From a general user perspective, this setup improves model accessibility and reproducibility. From a developer and contributor perspective, it reduces the effort needed to maintain and extend the framework.

### 5.1.4 Possibility of extension and customization

Most frameworks have open source code and permissive licenses, making it possible to modify and extend their codebase. Within this category, some frameworks are specifically intended to be customized (e.g., implementing new functionalities) as part of their regular usage without an expectation of "developer-level" skills (e.g., ECHSE). Other frameworks do not envisage customization in their primary scope, but can still be modified by modelers with appropriate programming expertise in consultation with available developer guides (e.g., RAVEN).

Some frameworks have not been released as open source, and the only way to access their codebase for customization and extension is by contacting their developers (e.g., SUPERFLEX-F90, PERSiST).

SuperflexPy is designed to facilitate extension and customization as part of its regular usage. New *components* can be created by extending or modifying existing components, as demonstrated in Section 3.2.

### 5.1.5 Computational efficiency

The computational efficiency of a model code, i.e., the time required to run a simulation, depends
primarily on two aspects, namely the programming language and the numerical algorithms.

In terms of programming languages, most frameworks have been implemented in C/C++ and Fortran, which enable very fast computation. These implementations can be either purely single-language (e.g., FUSE, RAVEN), or wrapped within a scripting language to provide a more suitable interface (e.g., CMF). Amongst the considered existing frameworks, only MARRMoT is implemented entirely in an interpreted
language (Matlab/Octave).

In terms of numerical algorithms, a wide range of options are available for solving differential equations. Broadly speaking, time stepping algorithms can be classified as implicit or explicit, and may employ fixed or adaptive step size. The choice of algorithm and its settings brings tradeoffs between solution accuracy, algorithm complexity and computational cost. In the context of model development and comparison, it is
765 important to separate the specification of model equations from the choice of numerical solution and to use robust numerical methods to avoid spurious artefacts (e.g., Kavetski and Clark, 2010). The majority of frameworks implement this separation and provide a choice of built-in numerical algorithms.

SuperflexPy, while written entirely in Python (a nominally "slow" language), makes several implementation choices to reduce computational costs. These choices include the use of efficient
numerical libraries (section 4.4) and the solution of the *elements* in succession (DAG, section 5.2). This solution of the elements "one-at-a-time" enables the usage of robust solvers that operate on a single ODE at a time; in such cases, also the root finder operates on a single algebraic equation at a time, reducing the computational effort. The choice of numerical algorithm for individual elements is left to the user (section 4.3). The (recommended) built-in approximators include the implicit Euler scheme with fixed step size,
which offers stability and smoothness benefits valuable in parameter estimation contexts.

### 5.2    Current restrictions in model structure specification

As part of balancing the flexibility, ease of use, and computational performance of SuperflexPy, some restrictions have been imposed on the connectivity between model *components*.

The first restriction is that *elements* within a *unit* must form a directional acyclic graph (DAG), with no
feedback loops from downstream to upstream *elements* (Section 2.1). This restriction enables the

numerical solvers to proceed, at each time step, in a single pass from upstream to downstream *elements* and improves the computational performance of the framework. The restriction on internal model feedbacks is not expected to be overly limiting when developing conceptual hydrological models, where the fluxes from a given *element* typically depend only on the state in that *element* and not on downstream *elements*. In such systems, flows occur only in one direction, e.g. in model M4 the water flows from UR to FR but not vice versa. A counter-example where internal model feedbacks are required is given by the bidirectional interaction between surface water and groundwater in the hyporheic zone, where the exchange flux (or fluxes) depends on both states. Such interactions can still be modelled in SuperflexPy by introducing *elements* that embed feedbacks internally. For example, the hyporheic zone can be represented using a two-state reservoir with interacting states (e.g., Seibert et al., 2003). In other words, the SuperflexPy restriction on model feedbacks applies to interactions *between* elements, but not to interactions *within* an element.

The second restriction, which also applies at the *unit* level, derives from the decision to define the DAG as a succession of *layers* (section 5.1.1). This choice simplifies the model definition in typical use cases, when there are many *elements* with relatively few connections (i.e., the DAG is "sparse" rather than "dense"). However, the definition of a DAG as a succession of *layers* requires the *elements* to be connected directly one to the other, without skipping *layers*. Hence the need for transparent elements, which output the inputs they receive and are used to fill the gaps that arise when two or more parallel flow paths have a different number of *elements*. An example of such model configuration is given in Figure 9, where a transparent *element* (labeled "lower_transparent") is used to fill the gap in layer 7.

The third restriction is that the topology of a *network* must represent a tree where any given *node* can connect and transfer fluxes only to a single downstream *node* (Section 2.1). This restriction has a similar motivation to the restriction of a *unit* structure to a DAG, and allows for a simple and efficient computational implementation, which starts from the headwater *nodes* and proceeds downstream one *node* at a time. Typical distributed conceptual models meet this restriction, for example as illustrated in Section 3.3. However, fully integrated distributed models, such as Parflow and Mike-She, do include mutual dependencies between spatial elements, e.g., leading to 2D or 3D groundwater flows. Such configurations are considered beyond the scope of conceptual distributed models, and therefore are not currently supported in SuperflexPy.

## 5.3   Current usage and future developments

SuperflexPy is easy to install and run; it is written in pure Python and its dependencies are limited to the packages Numpy and Numba (Section 4.4). Installation can be done directly using the package installer for Python (pip) and does not require (additional) external libraries. We stress that SuperflexPy is not a

wrapper of earlier SUPERFLEX-F90 code but offers a completely new implementation that is not constrained by choices taken in the earlier code versions.

SuperflexPy has already been used for research applications. Jansen et al. (2021) performed a "model mimicry" study where similarities and differences within the HBV family of models were investigated. SuperflexPy was used to construct a set of HBV-like models and compare them in terms of the behavior of individual model components, the impact of numerical implementation, and so forth. A list of publications using SuperflexPy is maintained on the documentation website.

In terms of future developments, we hope that SuperflexPy offers the broader hydrological community a versatile new tool for research work and practical applications. Further SuperflexPy developments are likely to follow from such work and collaborations, including: (i) expansion of the library of model *components* beyond the ones here presented (as shown in the example in Section 3.2), and (ii) more fundamental developments in response to future model applications. It is important to highlight that SuperflexPy can be used to create and combine new model *components*, thereby enabling experimentation with new model structures and general conceptualizations. The framework, therefore, is not limited to *components* and structures taken from existing models – though such collections could be also produced. The SuperflexPy model library may grow as new users share their implementations with the community. In order to facilitate the use of SuperflexPy, its code is accessible on GitHub with license LGPL-3.0 and distributed using the Python package installer PyPI (see the code availability section at the end of this paper). The online documentation provides a guide for colleagues interested in contributing to the framework (section 2.1 of the supplementary material).

## 6    Summary and conclusions

SuperflexPy is a new Python flexible modelling framework for building conceptual catchment-scale hydrological models ranging from lumped to distributed configurations. SuperflexPy offers detailed control over each aspect of model configuration, and caters to a wide range of typical conceptual model applications. In order to facilitate the model building process, the framework defines its *components* (building blocks) at four hierarchical levels, namely *element*, *unit*, *node,* and *network*. These *components* support conceptual model setups of increasing levels of complexity, including but not limited to: a single element model (e.g. a reservoir), a typical lumped model (e.g. a collection of interconnected reservoirs), a semi-distributed model designed to provide prediction at a single outlet, and a semi-distributed model designed to provide predictions at internal sub-catchments. The construction of a model from components up to a given hierarchical level does not require specifying components at higher levels, which makes the model  configuration  effort  proportional  to  the  complexity  of  the  application  and  reduces

configuration/computational overheads. The framework supports multiple states and fluxes in each *component*, which facilitates future extension to applications where such functionality is needed.

SuperflexPy offers an open source implementation of the SUPERFLEX principles (Fenicia et al., 2011) that builds on the collective experience of the authors and their colleagues in hydrological model design and application. The paper discusses the key design choices made in SuperflexPy, with emphasis on the ease of use and interfacing, availability, amenability of extensions, and computational efficiency.

The use of the SuperflexPy framework is illustrated using two examples that represent typical tasks in conceptual hydrological modelling: the implementation of a lumped model to simulate an entire catchment, and the implementation of a distributed model to simulate a system of multiple sub-catchments with spatially varying landscape characteristics. We hope the framework will contribute to ongoing efforts in the hydrological modelling community to develop more robust and representative models. The framework is open source, available with license LGPL-3.0 on GitHub.

**Code availability**

The source code of SuperflexPy, together with documentation and examples, is hosted in the public GitHub repository https://github.com/dalmo1991/superflexPy. Github is used for issue-tracking. Package releases are distributed using the Python package index https://pypi.org/project/superflexpy. Releases are identified using a version number based on Semantic Versioning 2.0.0 and assigned a DOI through Zenodo. The release associated with this paper represents version 1.3.0 and has DOI https://doi.org/10.5281/zenodo.5235158. Detailed documentation is available through Read the Docs at https://superflexpy.readthedocs.io. The supplementary material to this paper represents a snapshot of the documentation at the time reported on the front page.

SuperflexPy is implemented using Python 3.7 and depends on Numpy (version 1.19) and Numba (version 0.50).

SuperflexPy is available under the license LGPL-3.0. Users of the framework are invited to share their modelling solutions with the community by contributing to the GitHub repository.

**Author contributions**

All authors contributed to writing the paper. MDM designed, implemented, and documented the Python package, with input from FF and DK.

## Competing interests

The authors declare that they have no conflict of interest.

## Acknowledgements

We thank Associate Editor Andrew Wickert, Philip Kraft, Riccardo Rigon, and two anonymous reviewers for their thoughtful and constructive feedback on our manuscript. We are grateful to James Craig, Martyn Futter, David Kneis, Wouter Knoben, and Philip Kraft for providing fast and informative responses that

helped us construct Tables 1 and 2.

## Financial support

This research has been supported by the Schweizerischer Nationalfonds zur Förderung der Wissenschaftlichen Forschung (grant no. 200021_169003).

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

**Figures**

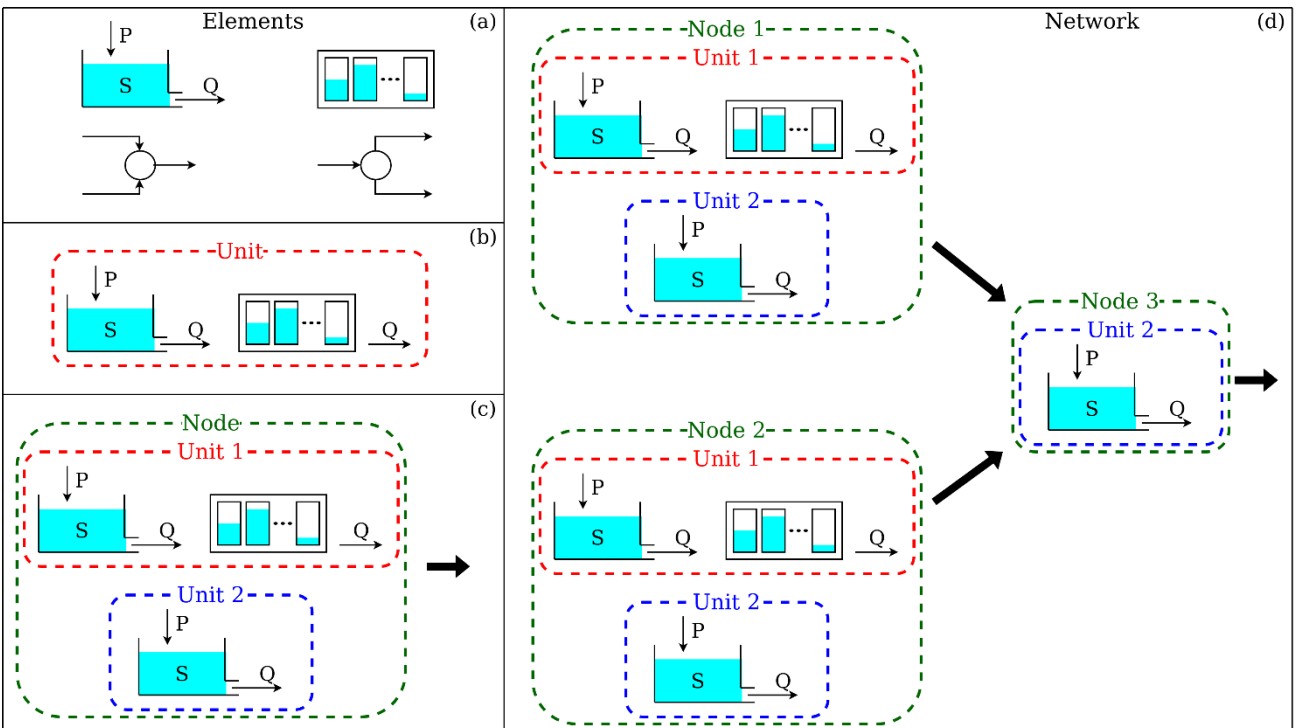

**Figure 1.** The four hierarchical levels of SuperflexPy and their respective *components*. (a) *Elements* (e.g. reservoirs, lags, connections) are used to represent individual hydrological processes/catchment response mechanisms; (b) *Units* connect multiple elements and are intended to implement lumped catchment models; (c) *Nodes* collect multiple units that operate in parallel representing different landscape elements within a catchment; (d) *Network* connects multiple nodes and is used to represent distributed setups.

```
 from superflexpy.implementation.elements.hymod import LinearReservoir
 from superflexpy.implementation.elements.thur_model_hess import HalfTriangularLag
 from superflexpy.framework.unit import Unit
 from superflexpy.framework.node import Node
 from superflexpy.framework.network import Network
 from superflexpy.implementation.root_finders.pegasus import PegasusPython
 from superflexpy.implementation.numerical_approximators.implicit_euler import ImplicitEulerPython
 # Initialize computational tools
root_finder = PegasusPython()
numerical_approximator = ImplicitEulerPython(root_finder=root_finder)
# Initialize the elements
linear_reservoir = LinearReservoir(parameters={'k': 0.1}, states={'S0': 10.0},
approximation=numerical_approximator, id='LR')
lag = HalfTriangularLag(parameters={'lag-time': 3.5}, states={'lag': None}, id='LAG')
# Initialize the units
unit1 = Unit(layers=[[linear_reservoir], [lag]], id='U1')
unit2 = Unit(layers=[[linear_reservoir]], id='U2')
# Change parameters
unit2.set_parameters({'U2_LR_k': 0.2})
# Initialize the nodes
node1 = Node(units=[unit1, unit2], weights=[0.7, 0.3], area=5.0, id='N1')
node2 = Node(units=[unit1, unit2], weights=[0.9, 0.1], area=2.0, id='N2')
node3 = Node(units=[unit2], weights=[1.0], area=1.0, id='N3')
# Initialize the network
net = Network(nodes=[node1, node2, node3], topology={'N1': 'N3', 'N2': 'N3', 'N3': None})
# Assign the inputs to the nodes (assume P1, P2, P3 have been read)
node1.set_input([P1])
node2.set_input([P2])
node3.set_input([P3])
# Set the timestep
net.set_timestep(1.0)
# Run the model
net.get_output()
```

**Figure 2.** SuperflexPy code implementing the simple illustrative model in Figure 1d.

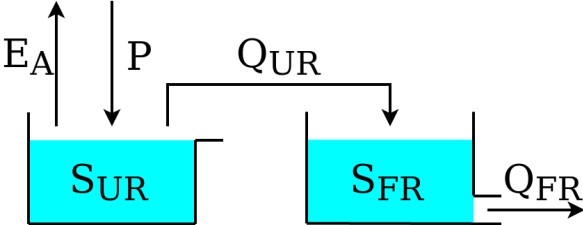

**Figure 3.** Schematic of model M4 used in the original SUPERFLEX case studies of Kavetski and Fenicia (2011).

```
from superflexpy.implementation.root_finders.pegasus import PegasusPython
from superflexpy.implementation.numerical_approximators.implicit_euler import ImplicitEulerPython
from superflexpy.implementation.elements.hbv import UnsaturatedReservoir, PowerReservoir
from superflexpy.framework.unit import Unit
import numpy as np
root_finder = PegasusPython()
numeric_approximator = ImplicitEulerPython(root_finder=root_finder)
ur = UnsaturatedReservoir(parameters={'Smax': 50.0, 'Ce': 1.0, 'm': 0.01, 'beta': 2.0},
states={'S0': 25.0}, approximation=numeric_approximator, id='UR')
fr = PowerReservoir(parameters={'k': 0.1, 'alpha': 1.0}, states={'S0': 10.0},
approximation=numeric_approximator, id='FR')
model = Unit(layers=[[ur], [fr]], id='M4')
P = np.loadtxt('precipitation.txt')
EP = np.loadtxt('evap_pot.txt')
model.set_input([P, EP])
model.set_timestep(1.0)
output = model.get_output()
```

**Figure 4.** SuperflexPy code implementing model M4 in Figure 3.

```python
class NewFastReservoir(ODEsElement):

    def __init__(self, parameters, states, approximation, id):

        ODEsElement.__init__(self, parameters=parameters, states=states,
                             approximation=approximation, id=id)

        # _fluxes_python is used to calculate the fluxes doing vector operations
        self._fluxes_python = [self._fluxes_function_python]
        # _fluxes is used to solve the ODE and it is specific to the architecture of the
numerical_approximator
        self._fluxes = [self._fluxes_function_python]

    def set_input(self, input):

        self.input = {'P': input[0]}

    def get_output(self, solve=True):

        if solve:
            self._solver_states = [self._states[self._prefix_states + 'S0']]
            self._solve_differential_equation()
            self.set_states({self._prefix_states + 'S0': self.state_array[-1, 0]})

        fluxes = self._num_app.get_fluxes(fluxes=self._fluxes_python,
                                          S=self.state_array,
                                          S0=self._solver_states,
                                          **self.input,
                                          **{k[len(self._prefix_parameters):]: self._parameters[k]
for k in self._parameters})

        return [- fluxes[0][1]]

    @staticmethod
    def _fluxes_function_python(S, S0, ind, P, k, alpha, b):

        if ind is None:
            return ([P, -(k * S**alpha)/(S + b)], 0.0, S0 + P)
        else:
            return ([P[ind], -(k[ind] * S**alpha[ind])/(S + b[ind])],
                    0.0,
                    S0 + P[ind],
                    [0.0, (k[ind] * S**alpha[ind])/((S + b[ind])**2) - (alpha[ind] * k[ind] *
S**(alpha[ind] -1))/(S + b[ind])S0 + P[ind] * dt[ind]])
```

**Figure 5.** General approach for implementing a new reservoir element `NewFastReservoir` by extending the class `ODEsElement` (Section 3.2.1).

```
 class NewFastReservoir(PowerReservoir):
 @staticmethod
 def _fluxes_function_python(S, S0, ind, P, k, alpha, b):
 if ind is None:
 return ([P, -(k * S**alpha)/(S + b)], 0.0, S0 + P)
 else:
 return ([P[ind], -(k[ind] * S**alpha[ind])/(S + b[ind])],
0.0,
S0 + P[ind],
[0.0, (k[ind] * S**alpha[ind])/((S + b[ind])**2) - (alpha[ind] * k[ind] *
    S**(alpha[ind] -1))/(S + b[ind])S0 + P[ind] * dt[ind]])
```

**Figure 6.** Simplified approach for implementing the `NewFastReservoir` by inheriting directly from class `PowerReservoir` (Section 3.2.2).

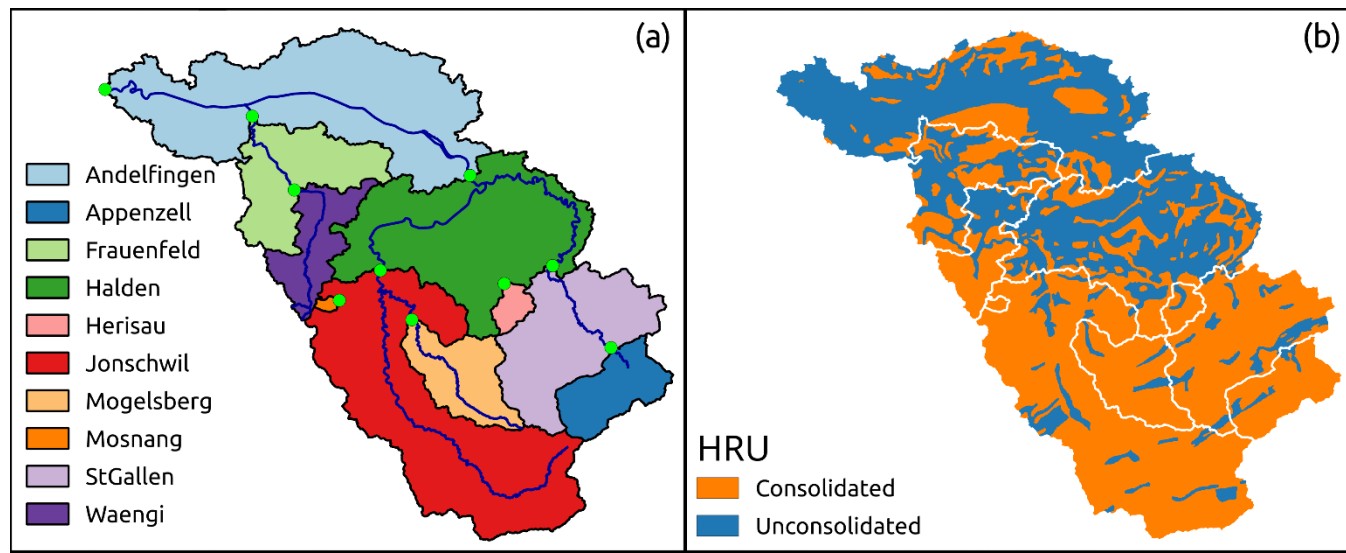

**Figure 7.** Illustration of catchment discretization used for a distributed application of SuperflexPy in the Thur catchment: (a) discretization into sub-catchments and (b) discretization into hydrological response units (HRUs) as presented in model M02 in Dal Molin et al. (2020). The panels of this figure were
1115 originally published in figures 1a and 6 of Dal Molin et al. (2020). The HRU model structure is shown in **Figure 8**.

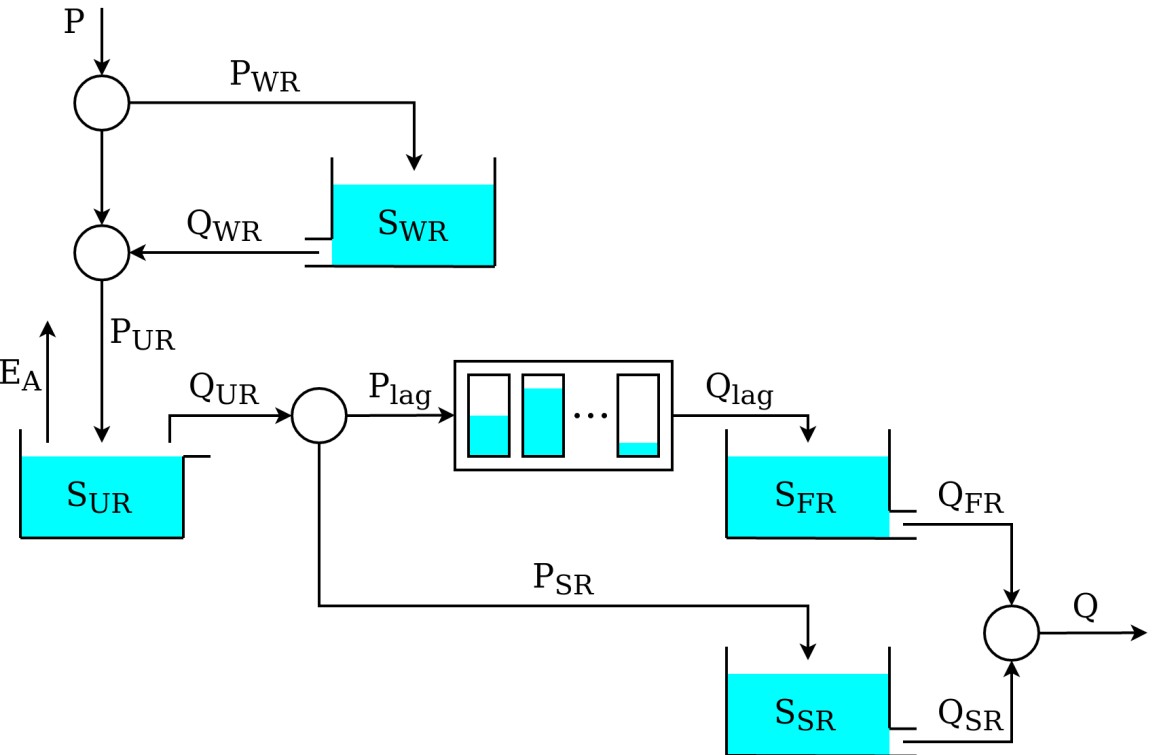

**Figure 8.** Model structure used to represent the HRUs in model M02 in Dal Molin et al. (2020). Refer to **Figure 7** for the corresponding HRU discretization of the Thur catchment.

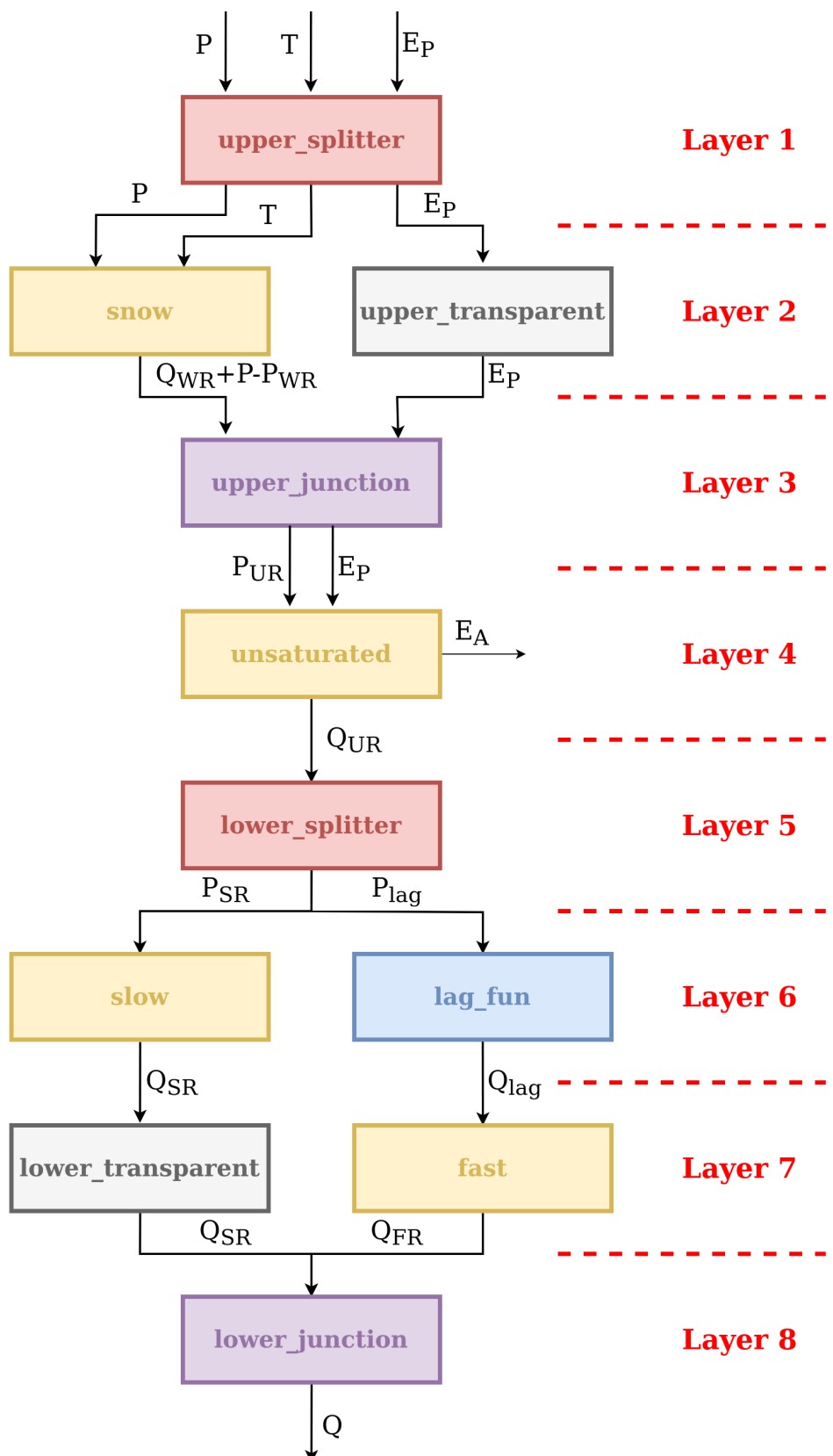

**Figure 9.** SuperflexPy representation of the model structure M02 in Figure 8.

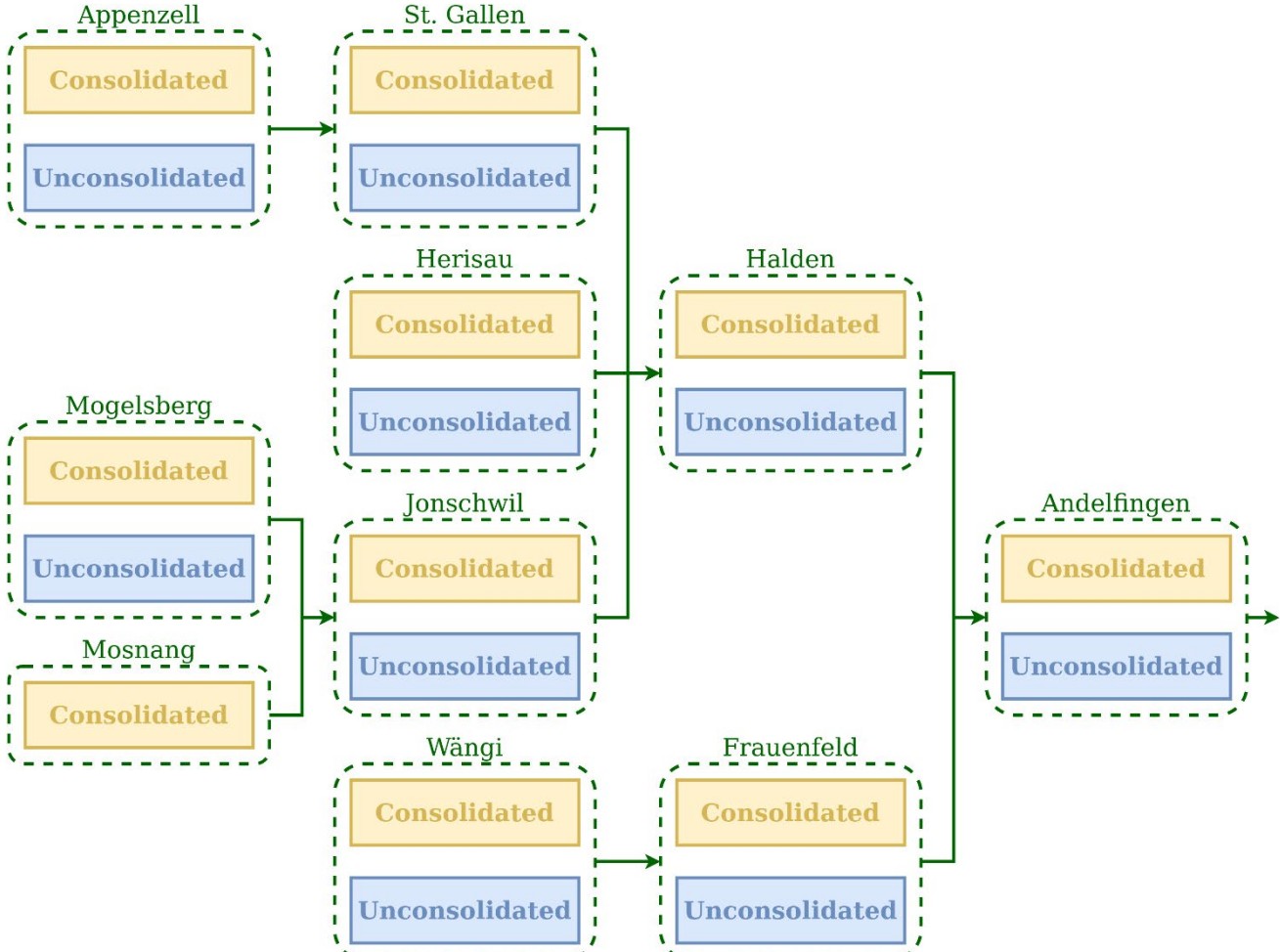

**Figure 10.** Spatial organization of the SuperflexPy model configuration used to simulate water fluxes in the Thur catchment (M02 inDal Molin et al., 2020). The *units*, used to represent the HRUs, are shown using the blue and yellow boxes. The nodes, used to represent the sub-catchments, are shown using the green dashed boxes. The group of nodes connected together (green arrows) creates a *network*.

```
 from superflexpy.implementation.root_finders.pegasus import PegasusPython
 from superflexpy.implementation.numerical_approximators.implicit_euler import ImplicitEulerPython
 from superflexpy.implementation.elements.thur_model_hess import SnowReservoir, UnsaturatedReservoir,
    HalfTriangularLag, PowerReservoir
 from superflexpy.implementation.elements.structure_elements import Transparent, Junction, Splitter
 from superflexpy.framework.unit import Unit
 from superflexpy.framework.node import Node
 from superflexpy.framework.network import Network
 # Initialize the elements
solver = PegasusPython()
approximator = ImplicitEulerPython(root_finder=solver)
upper_splitter = Splitter(direction=[[0, 1, None], [2, None, None]],
weight=[[1.0, 1.0, 0.0], [0.0, 0.0, 1.0]],
id='upper-splitter')
snow = SnowReservoir(parameters={'t0': 0.0, 'k': 0.01, 'm': 2.0}, states={'S0': 0.0},
approximation=approximator, id='snow')
upper_transparent = Transparent(id='upper-transparent')
upper_junction = Junction(direction=[[0, None], [None, 0]], id='upper-junction')
unsaturated = UnsaturatedReservoir(parameters={'Smax': 50.0, 'Ce': 1.0, 'm': 0.01, 'beta': 2.0},
states={'S0': 10.0}, approximation=approximator, id='unsaturated')
lower_splitter = Splitter(direction=[[0], [0]], weight=[[0.3],  [0.7]], id='lower-splitter')
lag_fun = HalfTriangularLag(parameters={'lag-time': 2.0}, states={'lag': None}, id='lag-fun')
fast = PowerReservoir (parameters={'k': 0.01, 'alpha': 3.0}, states={'S0': 0.0},
approximation=approximator, id='fast')
slow = PowerReservoir (parameters={'k': 1e-4, 'alpha': 1.0}, states={'S0': 0.0},
approximation=approximator, id='slow')
lower_transparent = Transparent(id='lower-transparent')
lower_junction = Junction(direction=[[0, 0]], id='lower-junction')
# Initialize the HRUs
consolidated = Unit(layers=[[upper_splitter], [snow, upper_transparent], [upper_junction],
[unsaturated],  [lower_splitter],  [slow, lag_fun],
[lower_transparent, fast], [lower_junction]],
id='consolidated')
unconsolidated = Unit(layers=[[upper_splitter], [snow, upper_transparent], [upper_junction],
[unsaturated],  [lower_splitter],  [slow, lag_fun],
[lower_transparent, fast], [lower_junction]],
id='unconsolidated')
# Create the catchments
andelfingen = Node(units=[consolidated, unsaturated], weights=[0.24, 0.76], area=403.3,
    id='andelfingen')
appenzell = Node(units=[consolidated, unsaturated], weights=[0.92, 0.08], area=74.4, id='appenzell')
frauenfeld = Node(units=[consolidated, unsaturated], weights=[0.49, 0.51], area=134.4,
    id='frauenfeld')
halden = Node(units=[consolidated, unsaturated], weights=[0.34, 0.66], area=314.3, id='halden')
herisau = Node(units=[consolidated, unsaturated], weights=[0.88, 0.12], area=16.7, id='herisau')
jonschwil = Node(units=[consolidated, unsaturated], weights=[0.9, 0.1], area=401.6, id='jonschwil')
mogelsberg = Node(units=[consolidated, unsaturated], weights=[0.92, 0.08], area=88.1,
    id='mogelsberg')
mosnang = Node(units=[consolidated], weights=[1.0], area=3.1, id='mosnang')
stgallen = Node(units=[consolidated, unsaturated], weights=[0.87, 0.13], area=186.6, id='stgallen')
waengi = Node(units=[consolidated, unsaturated], weights=[0.63, 0.37], area=78.9, id='waengi')
# Create the network
thur_catchment = Network(nodes=[andelfingen, appenzell, frauenfeld, halden, herisau,
jonschwil, mogelsberg, mosnang, stgallen, waengi],
topology={'andelfingen': None, 'appenzell': 'stgallen',
'frauenfeld': 'andelfingen', 'halden': 'andelfingen',
'herisau': 'halden', 'jonschwil': 'halden',
'mogelsberg': 'jonschwil', 'mosnang': 'jonschwil',
'stgallen': 'halden',  'waengi': 'frauenfeld'})
```

**Figure 11.** SuperflexPy code implementing the distributed model in Figure 9 and Figure 10.

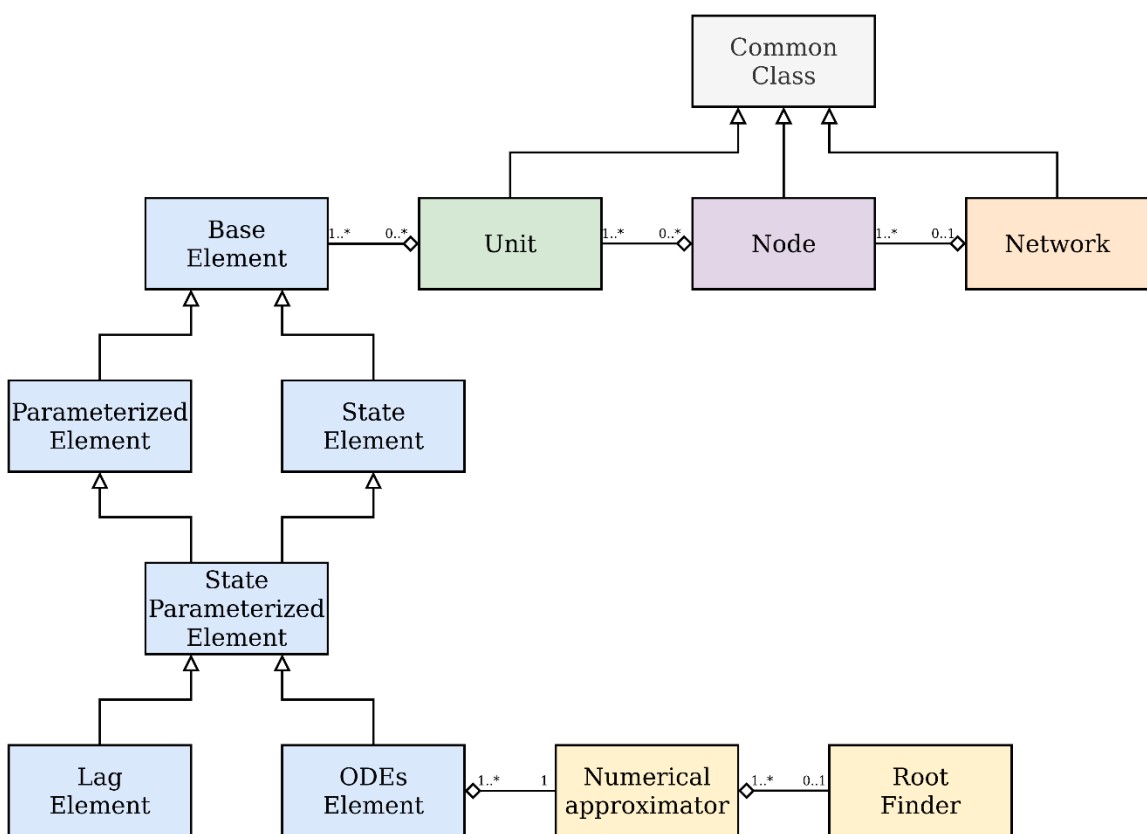

**Figure 12.** UML class diagram showing the organization of the classes used to represent SuperflexPy components. The core framework is presented, excluding the specific implementations of *components* and numerical routines.

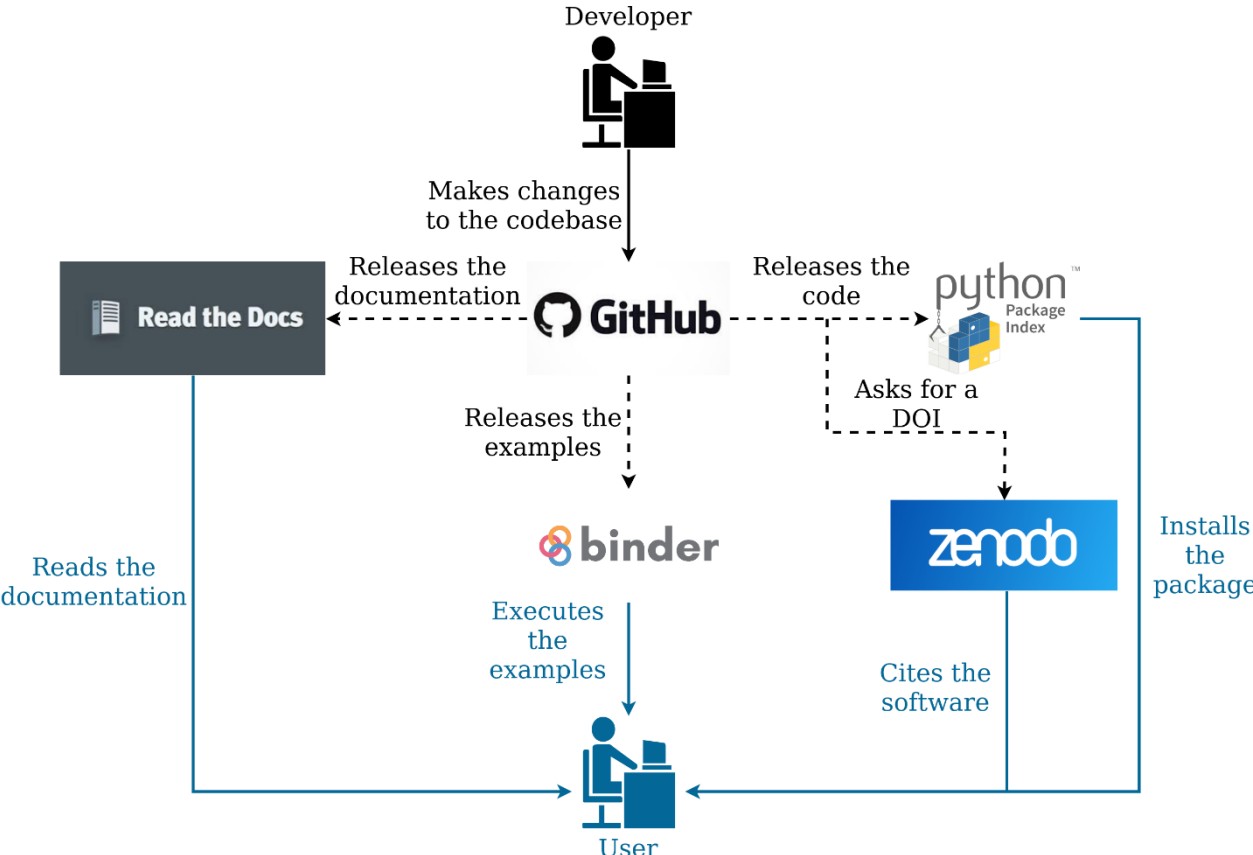

**Figure 13.** Organization of the SuperflexPy project, indicating the online software management tools used to develop the source code and documentation, release product versions with associated DOIs, and provide general open access to all project components. Typical workflow paths for users and developers are shown, respectively, in the blue and black lines and font. Dashed lines represent automated workflows.

**Table 1.** Summary of usability characteristics of SuperflexPy in the context of selected flexible frameworks for conceptual hydrological modeling.*

| | Availability | Distribution and installation | Documentation | Interface and setup | I/O format for settings and data | Possibility of customization | Built-in calibration and uncertainty analysis |
|---|---|---|---|---|---|---|---|
| **SuperflexPy** | Open source | Python package | Available | Python package. Python script to setup | Direct I/O with Python. No binding to particular formats | Possible with moderate programming expertise | Not present |
| **FUSE (Fortran) (2008)** | Exe or code, by request from authors | Standalone exe/code | Comments in code (limited) | Executable with/without GUI, or Fortran subs. Setup files | Structured text files | Possible but not supported systematically | Some versions are coupled with optimization and MCMC sampling tools |
| **SUPERFLEX-F90 (2011)** | Exe or code, by request from authors | Standalone exe | Comments in code (limited) | CLI or DLL or Fortran subs. Setup files | Structured text files | Possible but not supported systematically | Not present |
| **CMF (2011)** | Open source | Python package. Code compilation for enhancements | Available | Python package. Python script to setup. GUI only for lumped models | Direct I/O with Python. No binding to particular formats | Customization using C++. Possibility with Python under development | No. Developers recommend to use the SPOTpy package from the same group |
| **PERSiST (2014)** | Exe/webapp after registration | Standalone executable or webapp | Exists. Not public at the moment | Desktop app or webapp. Setup files or GUI | Structured text files and XMLs | Possible but not supported systematically | Incorporates MCMC toolkit |
| **ECHSE (2015)** | Open source | R package to generate C code that has to be compiled | Available | CLI. Setup through text file or CLI | Delimited text files | Possible with moderate programming expertise | Not present |
| **MARRMoT (2019)** | Open source | Matlab/Octave package | Available | Collection of scripts and functions. Setup with script. | Direct I/O with Matlab/Octave. No binding to particular formats | Possible with moderate programming expertise | Not present |

| RAVEN (2020) | Open source | Standalone executable. May require NetCDF | Available | Executable without GUI. Setup files | Structured text files | Possible but requires developer-level expertise. Instructions in the documentation | DDS optimization. Reports model performance metrics usable by external software |
|---|---|---|---|---|---|---|---|

*This information was collated based on published information. A brief informal review was provided by the framework developers.

Abbreviations: exe = binary executable, subs = subroutines, GUI = graphical user interface, CLI = command line interface, DLL = dynamic link library, MCMC = Monte Carlo Markov Chain, DDS = Dynamic Dimensioned Search

**Table 2.** Summary of simulation capabilities of SuperflexPy in the context of selected flexible frameworks for conceptual hydrological modeling.*

| | Structural flexibility | Spatial flexibility | Hydrological processes | Numerical solution options | Pre and post processing | Programming language |
|---|---|---|---|---|---|---|
| **SuperflexPy** | Components can be connected freely | Lumped; semi-distributed | Water fluxes; Designed to handle multiple fluxes | Fixed step implicit and explicit Euler. Possibility to use custom solvers | Not available | Python |
| **FUSE (Fortran) (2008)** | Master structure; components selected for each model decision | Lumped | Water fluxes | Implicit, semi-implicit, explicit schemes; fixed and adaptive step solvers | Not available | Fortran |
| **SUPERFLEX-F90 (2011)** | Master structure; components can be turned on/off | Lumped; semi-distributed | Water fluxes; transport processes | Fixed step implicit and explicit Euler | Not available | Fortran |
| **CMF (2011)** | Components can be connected freely | Lumped; semi-distributed; fully-distributed | Water fluxes; transport processes | Implicit and explicit schemes; single or multistep solvers | Calculation methods for PET | Python wrapping of C++ code |
| **PERSiST (2014)** | Components can be connected freely | Semi-distributed | Water fluxes; designed to be coupled with transport models (INCA) | Implemented as a series of first order difference equations | PET calculated internally | C++ |
| **ECHSE (2015)** | Components can be connected freely | Lumped; semi-distributed; grids | Water fluxes; transport processes | To be implemented by user when defining the components | Not available | C++; R package to generate C++ code |
| **MARRMoT (2019)** | Library of model structures. Possibility to combine different components | Lumped | Water fluxes | Fixed step implicit and explicit Euler. Possibility to use custom solvers | Not available | Matlab/Octave |
| **RAVEN (2020)** | Components can be connected freely | Grids; subbasin/HRUs; triangulated irregular network | Water fluxes; transport processes | Ordered series, Euler, and predictor/corrector global methods; local methods at process level | Calculation and interpolation (spatial and temporal) of derived fluxes and other variables | C++ |

1155

*This information was collated based on published information. A brief informal review was provided by the framework developers.