# Peer review of "SuperflexPy 1.3.0: An open source Python framework for building, testing and improving conceptual hydrological models"

_Geoscientific Model Development, 2020_

## Referee Comment (RC1) · Philipp Kraft (Referee) · 18 Jan 2021

The manuscript with the title "SuperflexPy 1.2.0: an open source Python framework for building, testing and improving conceptual hydrological models" by Marco Dal Molin et al, describes the structure and use cases of a toolbox to build conceptual hydrological models and test hypotheses about catchment behavior. The toolbox is based on an earlier approach SuperFLEX, but presents its translation into the programming language Python together with some new features.

The manuscript is well written , and I would like to leave language review to native speakers. The structure is mostly sound, but I have these major issues with the

manuscript and would like to have them addressed in a revised version:

1. The manuscript misses a discussion, where the reader can understand what kind of problems this new system can solve, that are not possible to tackle with the tools already available. What is the improvement of the status quo?

2. The mathematical framework, especially the integration of elements is not sufficiently explained

3. The user interface (in this case the API) of the model system does not comply with the norms and standards of the Python programming language.

Minor issues are some misunderstandings in the introduction and shortcomings in the explanation of the model element description.

**1   General Comments**

**1.1   Missing discussion**

The introduction explains that some of the existing frameworks are only suitable for lumped models, like MARRMOT, FUSE and PERSiST and stresses the importance of distributed conceptual models. However, model frameworks with that ability exist and are cited in the introduction: CMF, ECHSE, SUMMA and RAVEN. What problems exist with these frameworks, and how does SuperFlexPy solve these problems? The authors state in l. 210 as an aim for the ms: "Provide a broad discussion of how the SuperflexPy contributes to the toolkits available to the hydrological community, including existing flexible frameworks, in terms of intended scope of application, advantages, and limitations", but mention the capbilities only very briefly in the introduction. This should

be expanded and moved to the discussion to include SuperFlexPy in the comparison. The current sections 5.1 and 5.2 seems to be a good place (see specific comments). As author and maintainer of CMF I can give my view on the differences between CMF and SuperflexPy (references for information only, not intended as suggestions for the reference list of the manuscript) and would be very interested about differences and similarities with other frameworks.

Similarities:

- Model is composed as a Python script

- Well defined coupling with optimization / rejection framework (in both cases SpotPy)

- Object oriented design

- Interactive exploration of model behavior (via Python prompt)

- Unrestricted possibilities for spatial granularity

- Fine process granularity

- Open source, open access

- Available via PyPI

- User defined time stepping scheme

Features that SuperFlexPy has, but not CMF:

- Lag functions

- Toolbox extendible by users without compiling (only experimental and unpublished feature in CMF)

Features of CMF not present in SuperFlexPy:

- Calculation methods for ETpot (eg. Jehn et al., 2017)

- Implicit and explicit single and multistep solvers (eg. Kraft et al., 2011)

- Multiprocessor support for large model systems (eg. Wlotzka et al., 2017)

- Complex topology (cyclic bidirectional graph) (eg. Maier et al., 2017)

- Energy potential based flow descriptions (eg. Richards- and St. Venant equation etc.) (Maier et al., 2017; Windhorst et al., 2014)

- Tested solute and isotope transport (eg. Haas et al., 2013; Windhorst et al., 2014)

- Explicit run time loop for simple model coupling (eg. Kellner et al., 2017; Kraft et al., 2010)

**1.2 Missing mathematical explanations**

- The term numerical approximator is not well defined.

- What happens if the root finding procedure does not converge? Flexible time stepping or does the implementation stop with an exception? Typically happens with fast snowmelt or power law equations with a large exponent.

- The standard implicit euler method implementation with the Pegasus root finding algorithm should be explained briefly

- How do the solvers deal with discontinuous or not continuously differentiable flux equations? The problem is described by Knoben et al 2019's MARRMoT Paper, Ch. 2.4( https://doi.org/10.5194/gmd-12-2463-2019) - it is the reason why I gave up mimicking exisiting models with CMF.

- The system can use for the solution of single elements implicit solvers - the need for that was very well explained by the co-authors in their "ancient daemons paper". How the solutions of the elements are combined to a the response of the entire model is not explained. I guess, that some kind of operator split is employed, but how do they deal with non linear behavior between timesteps? Is the numerical error of the operator split somehow controlled? Is there some lag of fluxes that are routed through lower nodes in a network?

1.3   Programming interface

The programming interface has a number of quirks and behaviour outside the norms of the Python language. I guess it is unusual to request changes to the programing interface of the software presented, but both manuscript and software would be improved. Not knowing and following the Python standards was one of the biggest mistakes I did with CMF, and it is very difficult to change the programming interface later.

Leading underscore:

In all code examples, where behavior of superflexpy components is extended / changed (polymorphism for object oriented programmers) the authors of the framework indicate with a leading underscore "something". A leading underscore of a class member has a clear and well defined meaning in the Python community:

"_single_leading_underscore:   weak "internal use" indicator.   E.g.     from M

import * does not import objects whose names start with an underscore."
(https://www.python.org/dev/peps/pep-0008/#public-and-internal-interfaces)

Internal use means, class members with a leading underscore should nearly never be
assessed by a user of the framework, neither for reading, nor for writing. The authors
seem to understand the leading underscore to indicate the concept of a "protected"
member in java, C# or C++, a concept that does not exist in Python.

Use of literals and implicit relations

Usage of literals to access parameter and state names instead of keyword arguments
or class properties makes usage and composition of the model components harder
to write. If "magic" literals are avoided, modern Python IDE's (integrated development
environments) can help with code completion. If the framework can allow one of the fol-
lowing two alternatives to create a component, the IDE can help with code completion,
instead of all purpose dictionaries. For network creation see comment l.469

```
linear_reservoir = LinearReservoir(
    k=0.1, S0=10.0,
    approximation=num_app,id='LR'
)

linear_reservoir = LinearReservoir(approximation=num_app,id='LR')\
    .parameters(k=0.1)\
    .states(S0=10.0)
```

The Splitter interface relates to the position of certain input datasets in a list, that does
not exist explicitly. This is quite hard to follow and to read. The definition of the topology
(falsely called "topography") of the Network class is redundant and uses the id-string

literals instead of the node objects directly. A very easy to read variant for the creation of the Network is explicit setting of the downstream node:

```
stgallen.downstream = appenzell
```

Another option would be to define the tree as a nested list structure. Each list contains the node and the left and right upsteam nodes, if present.

```
thur_catchment = Network([
    andelfingen,
    [halden,
     [stgallen,
      [appenzell]
      ],
     [jonschwil,
      [mogelsberg, mosnang]],
     [herisau],
     [frauenfeld, [waengi]]
     ]
])
```

Most modern programming environments (IDE) can mark typos in the node names with both versions presented here, and even employ code completion to help the user with typing, but this does not work for meaningful literals.

**2  Specific Comments**

132: RAVEN is clearly a framework for conceptual models (no energy potential / head based flow equations) for internal transport but CMF was originally developed for phys-

ically based models.

171 - 180: Here is a description of what RAVEN and SUMMA do missing.

180: CMF can model substance / isotope advective transport with adsorption without additional software, and reactive fluxes in coupled model approaches

187: Same fine granularity as CMF, RAVEN and MARRMoT.

210: See general comments

213 - 218: Link sections with the numbered aims of the paper

225: "An element can represent an entire catchment...": From my understanding this might be technically true, if the catchment can be represented by a single process. However, the meaning of "element" in SuperFlexPy is simpler to understand if 225 is changed to: "An element represents a specific process within the catchment." If needed this sentence can be extended by "In special cases, this specific process covers the entire catchment behavior and a single element is sufficient for the model."

Eq1: To ensure mass conservancy, how can g_s ever be different to $\frac{dS}{dt} = X(t) - g_y(S(t), X(t); ?)$?

230: This is a bit unclear: does SuperFLEXpy support substance transport? If yes, only theoretically, or has it been tested already?

234: How would a multistate reservoir look like (mathematically)?

242: Please give an example or more concrete description, what a connection is in terms of hydrological processes. As they are needed later, please explain the splitter, junction and transparent elements here.

260: Here only weight is mentioned (as the area fraction) while in 310 weight and area are different things. Please explain more consistent

274: The Level 3 concept predates the given references by a long time, please use

more classical example (eg. TOPMODEL, HBV (not light), etc.)

276: Same as above, please give a reference to classic "Level 4" model, eg. SWAT, SHETRAN, etc.

295: This is a structure problem: You need the numerical solution of the ODE for the construction of the model, but the numerical solution is not yet explained. Please consider moving Section 4.3 up as a new section 2.2. However, section 4.3 does not explain the terms "numerical approximator" and "root finder" and how these work together. Secondly, the "numerical approximator" is given as a parameter to each reservoir element. How many instances of numerical approximators exist? One global object to solve the entire ODE-system or is the numerical approximator copied and is a non-shared object of the reservoir? Or is the numerical approximator more a kind of function without any notion of specific state? If each element is solved on its own, how is the whole system integrated, i.e. how does the operator split work? As the "standard" solver is the own implementation of ImplicitEuler with Pegasus root finder a short explanation of the math behind it is missing. How does the composition of integration solution work?

305: The definition of the routing is quite obscure. Somehow the routing is derived from a List[List[Element]]. Since "explicit is better than implicit" (Zen of Python) an alternative (obvious) way to define routing would be preferable. Otherwise some more explanation, how the nested list translates into a tree structure is needed.

310: What unit is used for area?

315: Topography is not the right term here, do you mean topology? The dictionary with id's to define the topology is not helpful when using IDE's with code completion

392: "Input fluxes" is ambiguous here: Input can either mean the direction of the flux (input flux adds water to the element) or in the information sense: input is an externally defined time series that may add or take water from the element. This ambiguity needs

to be addressed in the whole manuscript and the documentation.

399: User facing methods should not start with an underscore. There is no concept of "protected" in Python (see general issue 2)

411: ditto

427: Is there a better symbol for the snow reservoir than WR? What is W?

443: Here is an explanation of splitters and junction missing, because the "gaps" can only exists together with the multipath structure (or better: explain them in section 2).

448: What is the role of the upper_splitter here?

465: When a unit is composed from elements (why the term "layers"?), are those elements copied or referenced (must be copied, but is not mentioned in the manuscript)? If I count correctly, each unit has 4 states and 11 parameters. So each node has 8 states and 22 parameters?

466: Same question here: does assigning a unit to a node make a copy of the unit? If yes, I would understand "consolidated" a kind of template. As the code is given, the parameters for consolidated and unconsolidated are the same, how do you change their parameters, if the units are copied? If copied, the model has 80 states and technically 220 parameters, that could be tuned independently, correct? Of course, the parameters for each of the 10 consolidated units might be coupled, to be the same, by convention.

469: cf. general issue 2

490: An UML-class diagram of the main components would be helpful

506 - 519: See comments to line 295

540: In l. 163, "Interoperability with external software, for example for model calibration and uncertainty analysis" is claimed as one of the desired features of modeling code.

This is a good place to mention how SuperFlexPy can be interfaced with calibration / uncertainty packages like Ostrich, SpotPy, PEST etc.

546: This section would be much improved, if there would be a single working prototype in the documentation for it, otherwise it should be shortened substantially and its content moved into section 5.3 (future development).

559 - 584: The missing manuscript aim 3 should go here. Section 5.1 can be enhanced by citing articles dealing or struggling with the "right complexity" across framework boundaries

585 - 605: How do the limitations of SuperFLEXPy compare to the other hydrological frameworks, especially the similar ones: MARRMoT, RAVEN, CMF, ECHSE? (See general comment)

654: Missing link to the Github repository. Figure 12 should be deleted, as it conveys only little information. The role of binder for the framework is unclear.

658: Please write the DOI out here. BTW, the subtitle of the code release "**The** flexible language of hydrological modelling" (emphasis mine) is a bit bold, given that similar frameworks / domain languages are available since a decade.

Figures:

Fig 2: See general comments about literals

Fig. 5: Users should not access class fields starting with an underscore (see general comment 3). Please use comments to explain l.8 and l.9. Is self._fluxes and self._fluxes_python callable or are these values?

Fig 6: see general comment 3.

Fig 7: Catchment boundaries and HRU boundaries can be presented in a single map

Fig 8: Why does "W" denote snow?

Fig 11: Parametrization of the Splitters is unclear. Missing different parametrization of the two HRU templates (if they are templates in fact)

Fig 12: Not necessary, can be removed

**3   References**

(no need to include into manuscript)

Haas, E., Klatt, S., Fröhlich, A., Kraft, P., Werner, C., Kiese, R., Grote, R., Breuer, L. and Butterbach-Bahl, K.: LandscapeDNDC: a process model for simulation of biosphere-atmosphere-hydrosphere exchange processes at site and regional scale, Landscape ecology, 28(4), 615-636, 2013.

Jehn, F. U., Breuer, L., Houska, T., Bestian, K. and Kraft, P.: Incremental model breakdown to assess the multi-hypotheses problem, Hydrology and Earth System Sciences Discussions, 1-22, https://doi.org/10.5194/hess-2017-691, 2017.

Kellner, J., Multsch, S., Houska, T., Kraft, P., MĂźller, C. and Breuer, L.: A coupled hydrological-plant growth model for simulating the effect of elevated CO2 on a temperate grassland, Agricultural and Forest Meteorology, 246, 42-50, 2017.

Kraft, P., Multsch, S., Vaché, K., Frede, H. and Breuer, L.: Using Python as a coupling platform for integrated catchment models, Adv. Geosci, 27, 51-56, https://doi.org/10.5194/adgeo-27-51-2010, 2010.

Kraft, P., Vache, K. B., Frede, H.-G. and Breuer, L.: A hydrological programming language extension for integrated catchment models, Environmental Modelling and Software, 26, 828-830, https://doi.org/10.1016/j.envsoft.2010.12.009, 2011.

Maier, N., Breuer, L. and Kraft, P.: Prediction and uncertainty analysis of a parsimonious floodplain surface water-groundwater interaction model, Water Resources Research, 53(9), 7678-7695, 2017.

Windhorst, D., Kraft, P., Timbe, E., Frede, H.-G. and Breuer, L.: Stable water isotope tracing through hydrological models for disentangling runoff generation processes at the hillslope scale, Hydrology and Earth System Sciences, 18(10), 4113-4127, 2014.

Wlotzka, M., Heuveline, V., Klatt, S., Kraus, D., Haas, E., Kiese, R., Butterbach-Bahl, K., Kraft, P. and Breuer, L.: Parallel Multiphysics Simulations Using OpenPALM with Application to Hydro-Biogeochemistry Coupling, in Modeling, Simulation and Optimization of Complex Processes HPSC 2015, pp. 277-291, Springer, 2017.

---

## Referee Comment (RC2) · Anonymous Referee #2 · 18 Jan 2021

The paper titled "SuperflexPy 1.2.0: an open source Python framework for building, testing and improving conceptual hydrological models" by M. Dal Molin et al. details the development and implementation details of a new flexible hydrological modelling framework. The framework is based upon an earlier code SUPERFLEX developed by the second and third authors (Fenicia et al., 2011), but re-built using object-oriented Python programming approaches.

The paper is generally well-written and appropriately structured, and the software is clearly the fruit of much labor and potentially worthy of its own publication. However, I am not sure that the authors have presented a sufficient argument as to the unique

value of this contribution. I have a number of addressable concerns mostly with respect to evaluation and assessment of the framework:

1) Most importantly, I think the authors miss out on an opportunity (and expectation) to distinguish this effort from other modelling frameworks cited herein. What makes SuperflexPy unique? What types of problems may it be applied to that other flexible frameworks cannot readily tackle? It would be useful to illustrate any perceived advantages via one or two case studies. In particular, the authors need to make a very strong case as to why this implementation is particularly advantageous relative to the original SUPERFLEX code, since the conceptualization seems very similar. Is it merely the object-oriented Python wrapper? If so, is this alone a sufficiently unique contribution for this journal? As part of this, they will necessarily have to discuss some of the strengths and weaknesses of existing modular hydrology tools and what role SuperflexPy takes in addressing perceived gaps. This is the most critical comment for the authors to address.

2) The authors refer a number of things that might someday be done using the framework but have not yet been implemented, which I found problematic. Specifically, they discuss transport of contaminants and isotopes and use of a more complex numerical solver. However, none of these advances have actually been implemented in this model. This content needs to be removed, as it is not a current advantage of the software tool, it is a hypothetical future advantage.

3) The evaluation of the computational efficiency lacks rigour. There are insufficient model details to evaluate the computational benefits of SuperflexPy or the specific Numba implementation, and the speedup is quantified as improving from "a couple of minutes" to "a few seconds" of runtime. A quantitative assessment is warranted here if the authors wish to make a defensible argument regarding computational efficiency.

4) The authors have made a unique choice of coupling all of the constitutive laws for fluxes into a single element, i.e., use of the UnsaturatedReservoir() element implies use

of the relationships in Eqns 6-8. This is quite different from what is seen in models such as SUMMA, MARMMoT, or RAVEN where the swappable "element" is the constitutive law rather than the collection of constitutive laws applied to one storage element. Can you justify this selection and/or discuss the implications of this approach as compared to the flux-based components? It seems like just swapping one of the constitutive laws for a storage element will often necessitate creating a new component. Likewise, I would like to see a defense of the use of a fixed number of layers which necessitates the use of "transparent elements" and a clarification of the role of these layers – why are they even necessary? What problem do they solve?

5) I recommend including a UML depicting the inheritance structure and currently implemented elements in the SuperflexPy code (or a subset of these in the UML with a list/table of elements elsewhere). It is very unclear how the breadth and quantity of options compares to other modular frameworks including the original SUPERFLEX.

I have included most of my minor comments in the attached PDF file.

Please also note the supplement to this comment:
https://gmd.copernicus.org/preprints/gmd-2020-409/gmd-2020-409-RC2-supplement.pdf

—————————————————

[Figure]

**Supplement:**

[revised manuscript text omitted]

$$Q^{(UR)} = P^{(UR)} \times \left(\overline{S^{(UR)}}\right)^{\beta^{(UR)}} \tag{7}$$

$$E_A^{(UR)} = E_P^{(UR)} \times \frac{\overline{S^{(UR)}}\left(1 + m^{(UR)}\right)}{\overline{S^{(UR)}} + m^{(UR)}} \tag{8}$$

In equations (6)-(8), $S_{max}^{(UR)}$ and $\beta^{(UR)}$ are model parameters. The quantity $m^{(UR)}$ is used to approximate a "smooth" threshold behavior; we typically fix $m^{(UR)} = 0.01$.

FR is a power-law reservoir,

$$\frac{dS^{(FR)}}{dt} = P^{(FR)} - Q^{(FR)} \tag{9}$$

with the storage-discharge relationship given by

$$Q^{(FR)} = k^{(FR)}\left(S^{(FR)}\right)^{\alpha^{(FR)}} \tag{10}$$

where $k^{(FR)}$ and $\alpha^{(FR)}$ are model parameters.

The inflow $P^{(FR)}$ is given by the outflow from UR, i.e., $P^{(FR)} = Q^{(UR)}$.

M4 is a lumped model with multiple *elements*, and hence can be implemented using SuperflexPy levels L1 and L2 (*element* and *unit*, see Section 2.1). Figure 4 shows the code needed to implement M4. Similar to the model described in Sect. 2.2, the two model *elements* (UR and FR) are already implemented. Hence, the user only needs to import (lines 1-3) and initialize (lines 7-13) the *elements* together with the numerical

routines. Next, the *unit* that comprises the two reservoirs is imported (line 4) and initialized (line 15). The input data, namely precipitation and PET time series, are set on line 20. Input data is provided using Numpy arrays. The reading of input data (from text file(s), databases, etc.) is done separately from SuperflexPy, using any suitable Python library or function. In this case, we use Numpy to read from a text file, as shown in lines 17-18.

The model configuration is then complete – line 23 runs the model with given input data to produce the model outputs. The outputs contain streamflow time series in the form of Numpy arrays.

**3.2 Changing the equations of the fast reservoir in M4**

Suppose the modeler wishes to modify model M4 by changing the storage-discharge equation of the fast reservoir given in equation (10) to a new relationship

$$Q^{(\text{FR})} = \frac{k^{(\text{FR})} \left( S^{(\text{FR})} \right)^{\alpha^{(\text{FR})}}}{S^{(\text{FR})} + b^{(\text{FR})}} \qquad (11)$$

[revised manuscript text omitted]

---

## Referee Comment (RC3) · Anonymous Referee #3 · 20 Jan 2021

The paper "SuperflexPy 1.2.0: An open source Python framework for building, testing and improving conceptual hydrological models" by Dal Molin et al. describes the development and implementation of a flexible hydrological modeling platform. SuperflexPy is based on the earlier SUPERFLEX model, but uses the Python language and adds new features.

The paper is well-written and generally well-structured, and I believe that SuperflexPy can provide a valuable and flexible tool for hydrologists. But I have some concerns with how the authors frame and present the model.

1. In section 1.3, the authors state that their 3rd aim is to provide a broad discussion

of how SuperflexPy contributes to the hydrological community and how it compares to existing modeling tools. The included discussion does very little to situate SuperflexPy with respect to existing models. The authors should include a discussion about the similarities and differences, as well as perceived advantages and disadvantages, of SuperflexPy with respect to existing model frameworks. Why should someone use this model rather than previously available tools?

2. On line 160, the authors state that a flexible model framework should cover substance transport modeling, but the version of SuperflexPy presented does not have modules to handle substance transport. It would be possible to add substance transport, but that's presented as a hypothetical rather than an existing part of the model framework.

3. Beginning on line 162, the authors outline three computational criteria that a model should meet: ease of use, including interoperability with external software, ease of modification, and computational efficiency. Of these three, I think that only ease of modification is adequately addressed in the manuscript. Installation and operation are discussed, as the authors outline the strengths of Python as an object-oriented and commonly used programming language. But they do not discuss operability with external software, model calibration, or uncertainty analysis. There are references to recently published manuscripts that seem to cover these topics in more detail, but I believe that more discussion is warranted.

Section 4.4 covers computational efficiency, but does not provide much quantification or comparison with other models. The authors state that Numpy and Numba provide a speed up compared to native Python of "factors up to 30" and state a difference between "a few seconds with the Numba implementation compared to a couple of minutes with native Python execution" but do not provide sufficient details about the structure or complexity of the model used to calculate these runtimes. A comparison to other modeling frameworks would be especially valuable.

[Figure]

4. I think that Sections 4.1, 4.2, 4.3, and potentially 4.5 should come before Section 3. These detail the computational structure of the model and would follow the initial description in Section 2 quite well. An expanded version of section 4.4 could then be incorporated into an expanded discussion comparing SuperflexPy to other model frameworks.

Specific line comments are included below:

125-128: This paper doesn't explore model comparison, and it's not clear to me how the latter half of this paragraph informs the manuscript.

178: The term granularity is unclear

224: The terms element and unit are somewhat unintuitive. It's clear how nodes connect in a network, but it's not clear why a unit should be made up of elements.

250. The acyclic directional graph seems like a significant drawback – backwater effects, capillary action, hyporheic flow, etc. are common hydrologic occurrences. How do other modeling platforms handle these?

270: The definition of Levels should be introduced prior to this

315: Topography should be topology

351: Are specific units required? A small discussion of how the model handles units would be helpful.

371-373: Unclear whether importing data as a Numpy array is required

436: If nodes share units, does that mean that the unit is duplicated in each node?

443-445: It's not clear to me why connectivity gaps would occur and why a transparent element is the only way to solve this issue.

500: Are there memory concerns with retaining history, particularly for complex systems or long runtimes?

528: Numba was mentioned before but is detailed for the first time here.

615:SuperflexPy used to construct and compare

---

## Referee Comment (RC4) · Riccardo Rigon (Referee) · 24 Jan 2021

Review of Dal Molin, Marco, Dmitri Kavetski, and Fabrizio Fenicia. 2020. "SuperflexPy 1.2.0: An Open Source Python Framework for Building, Testing and Improving Conceptual Hydrological Models." *Geosci. Model Dev.* https://doi.org/10.5194/gmd-2020-409.

By Riccardo Rigon

The paper illustrates the new software superflexPy, a system for doing hydrological modelling at catchment scale, which originates from the previous Superflex adding to it a more appealing implementation and offering a easier access and improved usability within the Jupyter/Python interface.

I think it is built on solid scientific premises and well deployed, even if I believe that overall its engineering has not the quality of a System Product (Frederick and Frederick, 1995). However it will be useful for many researchers in Hydrology. Its relatively easiness of use, its being based on Open Source tools, its effort to be object oriented, and its use of Jupyter infrastructures for the documentation and dissemination will encounter the favor of many users and researchers. It comes as one Python infrastructures, in which Landlab (Hobley et al., 2017, and https://landlab.github.io/#/) is a mature example. From its pros is the freshness of the approach and its usability and the very flat learning curve (if we do not include the learning of all the tools for developing illustrated in Figure 12 of the paper).

The paper is well written, well organised and requires very minor modifications. The main concerns I have regard the Figures, in which many symbols are incorrect, as I list below. Another concern has to do with the traditional representation that is given in the paper of the model M2 and M04. As proponent of a different way to represent the hydrological models I believe that some further effort can be made as I mention below, even if this is a side issue. Essentially I think that the representation used in the paper does not actually shows the mathematics behind the model. One part that does not work as it is, for instance is the "lag" item used in the paper. As shown in Bancheri et al., 2019, and Rigon and Bancheri (2020) the lag functions imply the existence of a reservoir which remains, in this case hidden because its functioning is assigned through a travel time distribution. Therefore, willing to preserve the same type of representation used in the paper, the lag item should be promoted at the same graphical level than the reservoirs of which is just a different expression. In our, Extended Petri Net representation, (Bancheri and Rigon, 2020) the M02 and M04 would be represented as shown in the attached pdf.

Having awareness of this is indeed important in this modelling because the strategy underneath SuperflexPy is the comparison of model structures for a better representation of catchment processes (Clark et al., 2011) and getting, for what is possible in lumped models, the "right answer for the right reasons" (Kirchner, 2006).

Though my point is marginal to the economy of this paper I want the Authors and the readers to consider that having a proper visualization of the models is deemed necessary in view of the selection tasks and of implementing those extensions to treat contaminants that the Authors envision in the final parts of the manuscript.

In synthesis, I believe that the paper can be published with very minor modifications, even if I would prefer, but I cannot require that much, that they change their model representation.

**Detailed comments**

Line 235 - As I mentioned in the main text, the lag function is nothing different from a reservoir from the point of view of the model structure. It is just that the reservoir dynamics is given a different way. E.g. Rigon and Bancheri, 2020

Line 3 - Is the node here what elsewhere is called "Hydrologic Response Unit" ?

Line 275 - Among the models that generate Level 4 predictions, I would cite Formetta et al., 2014

Line 440 - acyclic directional graph or directed acyclic graph ?

Figure 3 - The right reservoir should be renamed S_{FR}

Figure 4 - The Figure suggests me to ask if there is any method to have, inside SuperflexPy, the list of the available reservoirs.

Figure 8 - The bottom reservoir should be renamed S_{SR}

Figure 9 - Why transparent layers are necessary ? Are not they a weakness in the software design ?

Figure 10 - A curiosity here: how can parameters of the consolidated and unconsolidated can be distinguished. Is there any problem with identifiability ?

**Further comments on lumped models representation**

As side issue, I show here below the representation of the Model M2 with the Extended Petri Net

[Figure]

This Figure is actually intended to clarify the fact that the delay introduced by the lag function assume the existence of a reservoir. It is the yellow circle in Figure and I stress that is important to account for properly when discussing of the models structure. The black frame on the flux is used to indicate that the budget of this reservoir is assigned through a travel time distribution (see also Rigon and Bancheri, 2020).

[Figure]

M4 model is much simpler though and possibly the EPN representation does not have any particular added value with respect to representation used in the reviewed paper. For a short introduction to the EPN, please see http://abouthydrology.blogspot.com/2020/10/introducing-extended-petri-net-by.html

**References**

Bancheri, Marialaura, Francesco Serafin, and Riccardo Rigon. 2019. "The Representation of Hydrological Dynamical Systems Using Extended Petri Nets (EPN)." *Water Resources Research* 55 (11): 8895–8921.

Brooks, Frederick Phillips, and Frederick P. Brooks Junior. 1995. *The Mythical Man-Month: Essays on Software Engineering*. Addison-Wesley.

Clark, Martyn P., Dmitri Kavetski, and Fabrizio Fenicia. 2011. "Pursuing the Method of Multiple Working Hypotheses for Hydrological Modeling: HYPOTHESIS TESTING IN HYDROLOGY." *Water Resources Research* 47 (9). https://doi.org/10.1029/2010wr009827.

Formetta, G., A. Antonello, S. Franceschi, and O. David. 2014. "Hydrological Modelling with Components: A GIS-Based Open-Source Framework." *Modelling & Software*. https://www.sciencedirect.com/science/article/pii/S1364815214000292.

Hobley, Daniel E. J., Jordan M. Adams, Sai Siddhartha Nudurupati, Eric W. H. Hutton, Nicole M. Gasparini, Erkan Istanbulluoglu, and Gregory E. Tucker. 2017. "Creative Computing with Landlab: An Open-Source Toolkit for Building, Coupling, and Exploring Two-Dimensional Numerical Models of Earth-Surface Dynamics." *Earth Surface Dynamics* 5 (1): 21–46.

Kirchner, James W. 2006. "Getting the Right Answers for the Right Reasons: Linking Measurements, Analyses, and Models to Advance the Science of Hydrology: GETTING THE RIGHT ANSWERS FOR THE RIGHT REASONS." *Water Resources Research* 42 (3): 1–5.

Rigon, Riccardo, and Marialaura Bancheri. 2020. "On the Relations between the Hydrological Dynamical Systems of Water Budget, Travel Time, Response Time and Tracer Concentrations." *Hydrological Processes*, no. hyp.14007 (December). https://doi.org/10.1002/hyp.14007.

---

## Editor Comment (EC1) · Andrew Wickert (Editor) · 6 Mar 2021

This paper received an astounding four reviews, which occurred because I was not fast enough at the trigger to close down the invitations after three referees accepted. I think that this, and the very constructive nature of these reviews, indicates a combination of enthusiasm for your work and feedback on how to improve both it and its presentation. Indeed, these reviews are varied and present a range of different areas of focus and expertise. I would like to invite the manuscript authors to respond thoughtfully to these broad-ranging reviews, and to use these to create an improved and revised submission. I look forward to seeing your responses to the feedback.

---

## Editor Comment (EC2) · Andrew Wickert (Editor) · 19 Mar 2021

Many thanks to the authors for their thoughtful responses to the review comments. I invite a revised submission, and include a few notes on the general high-level responses below. Where I do not include notes, you may assume that I approve of your plan for moving forward. I have read but not commented on your responses to each individual referee.

- **MP2 - Detailed comparison with existing frameworks**: I understand the hesitancy with regard to a large comparison that would, I agree, not be the focus

of a model description paper. A table of different existing packages and a short discussion, or something like this, should suffice.

- **1.3 MP3 - Degree of details: balance of content between the paper and the documentation**: I agree that referencing the documentation would be helpful to the reader. However, if you expect the code and documentation to evolve over the next 5-10 years, I would caution against detailed cross-referencing (e.g., of sections), unless you are explicit about the links with the version of record. This is in order to match the static paper with the potential evolution of a useful modeling tool.

- **1.5 MP5 - The representation of substance transport**: Agreed, and focus on the capabilities more than the aspirations (though you can include appropriate motivation).

---

## Author Comment (AC1) · 19 Mar 2021

**Response to the reviews of "SuperflexPy 1.2.0: an open source Python framework for building, testing and improving conceptual hydrological models."**

We thank the 4 reviewers and the Editor for their careful reading of the manuscript. Their insightful feedback and suggestions will be very helpful in improving the manuscript and addressing its current limitations.

This response is organized in two pieces.

First, we summarize and respond in a consolidated way to the following major points ("MP's") raised by the reviewers:

MP1.    Scope of the paper and novelty of SuperflexPy

MP2.    Detailed comparison with existing frameworks

MP3.    Degree of details; balance between the paper and the documentation

MP4.    Structure of the paper

MP5.    Representation substance transport

Second, we provide individual responses to the specific comments of the reviewers.

In the remainder of this document, the original comments by the reviewers are in typeset in *blue and italics font* and our replies are typeset in black font. The 4 reviewers are referred to as RC1-RC4, as per the GMD Open Discussion webpage.

**Consolidated responses to the major points raised by the reviewers**

**1.1 MP1 – Scope of the paper and novelty of SuperflexPy**

Some reviewers requested a clarification and justification of the scope and contribution of the paper.

For example: "*What makes SuperflexPy unique?*" (RC2.2), "*The authors should include a discussion about [..] perceived advantages and disadvantages, of SuperflexPy [..]. Why should someone use this model rather than previously available tools?*" (RC3.1)

The main intention of our paper is to document the development of new hydrological modelling software in order to allow other researchers/practitioners to use it and to acknowledge its use in a scientifically appropriate and reproducible way. Another important contribution is to share and exchange ideas regarding the software implementation of a flexible hydrological model. We believe these goals are well aligned with the Geoscientific Model Development (GMD) paper category "Model description papers".

As noted in the paper, SuperflexPy represents the first open source implementation of the Superflex modelling framework introduced in Fenicia et al. (2011) and applied in a series of subsequent publications (e.g., Fenicia et al., 2014; Fenicia et al., 2016; Fenicia et al., 2010). In the hydrological literature, modeling concepts and their software implementations have traditionally been introduced both concurrently or separately from each other. In this case, Superflex has already been previously introduced and is being actively used in several applications and therefore, the objective of this work, is not to introduce a new hydrological modelling theory or concept and to motivate its existence. Rather, this paper is intended to document the development of a Superflex open source software implementation, SuperflexPy.

The paper goes well beyond previous implementations of Superflex and contributes a completely new software design, including a detailed documentation and examples. Previous Superflex implementations were not specifically designed for broad distribution. The interest in a new Superflex implementation is indicated by the growing number of Superflex users and applications, in scientific research (more than 30 publications using Superflex), education, and training (e.g., university courses and regular summer schools). With this paper, we also facilitate the adoption of Superflex concepts and broader flexible modelling concepts, providing the hydrological community with open source software.

This paper is therefore submitted to the GMD paper category "Model description papers", which is dedicated, among others, to the description of "model components and modules, as well as frameworks and utility tools used to build practical modelling systems, such as coupling frameworks or other software toolboxes with a geoscientific application." (ref: https://www.geoscientific-model-development.net/about/manuscript_types.html#item1)

**PROPOSED ACTION**

1. Clarify better the scope of the paper; in particular emphasize the contribution of broader insights into software implementation of flexible models.
2. Revise the introductions to increase the prominence of the role of software in order to better connect to the main contribution of the paper (i.e. software implementation), and to avoid the potential confusion that we are proposing a new theoretical development.

**1.2   MP2 - Detailed comparison with existing frameworks**

Reviewers RC1, RC2, and RC3 suggest the paper should include a detailed comparison with existing frameworks.

For example: *"I think the authors miss out on an opportunity (and expectation) to distinguish this effort from other modelling frameworks cited herein"* (RC2.1), *"The authors should include a discussion about the similarities and differences, as well as perceived advantages and disadvantages, of SuperflexPy with respect to existing model frameworks"* (RC3.1).

We agree with the reviewers on the utility of comparison between SuperflexPy and other flexible modelling frameworks.

The aim of the paper is to present SuperflexPy, and in doing this we clearly recognize existing flexible frameworks for hydrological modelling and their software implementation. In particular, the introduction lists existing frameworks and software that operate in a similar context (lines 138-146). We then illustrate briefly, how the proposed software (i.e., SuperflexPy) should complement their characteristics (lines 172-180).

The presentation of these alternative frameworks helps contextualize our work, and acknowledges similar efforts by other researchers. As listed in the introduction, Aim 3 of the paper is to "Provide a broad discussion of how the SuperflexPy contributes to the toolkits available to the hydrological community, including existing flexible frameworks, in terms of intended scope of application, advantages, and limitations". This aim is reflected in several sections of the current paper. In particular, the scope is described in the introduction (lines 156-171), advantages are illustrated through the proposed demos and examples, and the limitations are detailed in Section 5.2.The main aim of this paper is to describe the new model software and how it achieves the objectives of the Superflex concepts. As we note in Main Point 1, this aim is consistent with the intended aim of the "Model description papers" category of GMD papers. Achieving this aim required a detailed exposition of the new software, representative case studies, and as required by GMD, a comprehensive online documentation.

As part of the revision process, we will make a more direct analysis of the software implementation choices made during the development of SuperflexPy, also in relation to alternative solutions (e.g. refer to the reply to RC1.33 later in this document). We believe these insights will be of interest to hydrological modelers.

For this reason, we suggest that a fuller, more detailed, comparison between SuperflexPy and concurrent flexible modelling frameworks would require a separate study and presentation. In particular, such comparisons would require a careful analysis of the numerical code of the different frameworks considered (as many details for such comparisons are not present in the publications), and to perform detailed numerical assessments on specific case studies.

Our understanding is that such dedicated model comparison studies are accommodated by GMD using the category of "Model evaluation papers" ([https://www.geoscientific-model-development.net/about/manuscript_types.html#item5](https://www.geoscientific-model-development.net/about/manuscript_types.html#item5)), which can be contrasted to the category "Model description papers" to which our paper is actually submitted. Alternatively, such studies might belong in other journals with different scope (e.g., HESS, as in Astagneau et al., 2020).

**PROPOSED ACTION**

1. In order to avoid misunderstandings, we will rephrase aim 3 as "Provide a broad discussion of hydrological modelling software implementation challenges and how SuperflexPy contributes to the toolkits available to the hydrological community".
2. Revise the introduction to enforce the relationship between SuperflexPy and alternative frameworks.
3. Provide additional insight on how the software organization of SuperflexPy tries to fulfill the ideal requirements of a flexible framework listed in lines 157-171, how it differs from other existing modelling toolkits, and what are the respective advantages and limitations.

**1.3 MP3 - Degree of details: balance of content between the paper and the documentation**

The reviewers RC1, RC2, and RC3 suggest including considerable additional technical detail to the paper.

For example: *"What happens if the root finding procedure does not converge? Flexible time stepping or does the implementation stop with an exception?"* (RC1.5), *"Here is an explanation of splitters and junction missing, because the "gaps" can only exists together with the multipath structure (or better: explain them in section 2)"* (RC1.40), *"There are insufficient model details to evaluate the computational benefits of SuperflexPy or the specific Numba implementation"* (RC2.9), *"not much discussion of routing at this point"* (RC2.38), *"Unclear whether importing data as a Numpy array is required"* (RC3.14), and others.

We appreciate this concern. Many of these questions are indeed of key interest to potential users of any modelling tool. We will ensure the paper and documentation provide sufficiently detailed responses to these issues.

Note that, according to the GMD guidelines (https://www.geoscientific-model-development.net/about/manuscript_types.html#item1), the publication is composed by both the paper and the documentation. The documentation is mentioned in the abstract and referenced many times throughout the paper (e.g., lines 319, 345, 476, etc.). The documentation (roughly 110 pages) is hosted online (http://superflexpy.readthedocs.org) and was provided as supplementary material as part of the review process.

The paper and the documentation serve distinct purposes:

- The paper is intended to present SuperflexPy to a general audience, highlight its key features, provide examples of usage, and discuss its scope of application.
- The documentation is intended to enable users to adopt the framework. Therefore, it contains detailed instruction on how to install, operate, interface, and (if desired) extend the framework.

Logically, we would like to manage the overlap between paper and documentation, to avoid repetition and potential inconsistencies/confusion.

Similar arguments apply to the comments on model "benchmarking". RC2 and RC3 have criticized the generality of the assessment done in section 4.4 on the performance of SuperflexPy. We argue that the purpose of that presentation is to stress that a flexible hydrological model written in Python can be relatively fast when efficient libraries are used. In order to give an impression of the time requirement of model computations, we provide only an order of magnitude (e.g. "a few seconds") for the runtime instead of a precise value. Instead, a full set of details is provided in the documentation (section 5.3), which contains all the specifications

requested by the reviewers (e.g., model configuration used, quantification of the runtime, dependence on the length of the simulation, etc.).

**PROPOSED ACTION**

1. Enhance paper to respond to key requests for detail by the reviewers, where it is clear these details are needed by readers to understand key aspects of SuperflexPy.
2. Enhance the paper with more cross-references to specific chapters of the documentation for the aspects that cannot be described in full depth in the paper.
3. Ensure the documentation provides detailed descriptions addressing the reviewer queries.
4. Explicitly explain the role of the paper and the documentation in contributing together to the publication.

**1.4 MP4 - Structure of the paper**

Reviewers RC1, RC2, and RC3 raise concerns regarding the current structure of the paper. In particular, the paper presents basic demos (Section 2) and examples (Section 3) before providing a detailed description of the framework (Section 4).

Relevant reviewer comments include: *"I think that Sections 4.1, 4.2, 4.3, and potentially 4.5 should come before Section 3. These detail the computational structure of the model and would follow the initial description in Section 2 quite well."* (RC3.5), *"This is a structure problem: You need the numerical solution of the ODE for the construction of the model, but the numerical solution is not yet explained. Please consider moving Section 4.3 up as a new section 2.2."* (RC1.28).

We agree that some details present in sections 2 and 3 are explained later in section 4, and that this may have created some confusion in understanding the sequence of the presentation.

The choice to present the material in this particular way was motivated by the intention to first present the more general (high level) aspects of SuperflexPy to "set the scene" on general appearance and scope of applicability, and subsequently walk the reader through more detailed demos and examples, to provide additional context on customization and interoperability.

Following this rationale, we provide basic demos and examples (sections 2 and 3) prior to detailed explanations (section 4). Demos and examples are intended for a general presentation of SuperflexPy that will help a broader audience get a quick sense of how SuperflexPy operates. Section 4, on the other hand, provides specific details of SuperflexPy that are targeted to a reader that wants to understand more about the framework.

**PROPOSED ACTION:**

1. Revise the paper so that this organization is clearly explained
2. Enhance Section 2.1 with some information from Sections 4.1 and 4.3 that may help the understanding of the rest of section 2 and section 3. Sections 4.2 (object-oriented design), 4.4 (computational efficiency), and 4.5 (possibility of having multiple fluxes and states) are not strictly necessary for the understanding of what follows.
3. Include more forward references to Section 4 from Sections 2 and 3.

**1.5 MP5 - The representation of substance transport**

Reviewers criticize the references to the simulation of transport processes because *"none of these advances have actually been implemented in this model"* (RC2.8). Particular critique is made of section 4.5 "Ability to represent multiple fluxes and states". The reviewers suggest removing such content *"as it is not a current advantage of the software tool, it is a hypothetical future advantage"* (RC2.8).

We agree that a model software paper should not present unsupported features as being "ready to use". We disagree that our paper made such claims, though as part of the revisions we will carefully check the wording to avoid any such confusion.

First, we note that the "possibility to use SuperflexPy for transport processes" is not a major aspect of the paper. We do list the possibility to represent substance transport as a desirable feature of flexible frameworks (e.g. line 197) and then clearly state that "The available examples in SuperflexPy do not include transport processes" (line 553).

Second, model extensibility is one of the key aims of our implementation (as stated in the introduction). SuperflexPy is intended to provide a platform (framework) for creating customizable hydrological models and rather than provide a "fixed" collection of components (line 622). Therefore, we believe it is appropriate to give examples of model aspects to which such extensibility could be applied (while clearly indicating such extensions have not yet been implemented). Since transport processes are becoming a common application of hydrological models, we believe it is appropriate and relevant to users to note this possibility in the paper (as a desirable feature of a flexible framework) and in the design of SuperflexPy.

In terms of specific text dedicated to this topic, it is very brief. In particular, it is limited to Section 4.5, which states that SuperflexPy already supports the creation of elements that have multiple fluxes and states. Such feature can become useful when simulating transport processes, for example. These extensions are more than an optimistic hope, as we are already using such setup in our own work on substance transport modeling (e.g., Ammann et al., 2020). We believe that this section is useful to the paper because, without it, a developer would not consider the possibility of extending SuperflexPy for substance transport.

**PROPOSED ACTION**

1. Improve the content of section 4.5 to make clearer that the section is about the possibility to have multiple fluxes and states.
2. Add more examples in 4.5 where the possibility to have elements with multiple states is an advantage (e.g. multiple state reservoirs, snow reservoirs with one states representing frozen and liquid water, etc.).
3. Ensure the paper does not give an impression that the current version of SuperflexPy can simulate transport processes "out of the box".

**2 Response to comments by RC1 (Dr. Philip Kraft)**

*The manuscript with the title "SuperflexPy 1.2.0: an open source Python framework for building, testing and improving conceptual hydrological models" by Marco Dal Molin et al, describes the structure and use cases of a toolbox to build conceptual hydrological models and test hypotheses about catchment behavior. The toolbox is based on an earlier approach SuperFLEX, but presents its translation into the programming language Python together with some new features.*

*The manuscript is well written, and I would like to leave language review to native speakers. The structure is mostly sound, but I have these major issues with the manuscript and would like to have them addressed in a revised version:*

- *The manuscript misses a discussion, where the reader can understand what kind of problems this new system can solve, that are not possible to tackle with the tools already available. What is the improvement of the status quo?*

- *The mathematical framework, especially the integration of elements is not sufficiently explained*

- *The user interface (in this case the API) of the model system does not comply with the norms and standards of the Python programming language.*

*Minor issues are some misunderstandings in the introduction and shortcomings in the explanation of the model element description.*

We thank the reviewer for his careful reading of the manuscript and his insightful suggestions.

*1 General Comments*

*1.1 Missing discussion*

*RC1.1: The introduction explains that some of the existing frameworks are only suitable for lumped models, like MARRMOT, FUSE and PERSiST and stresses the importance of distributed conceptual models. However, model frameworks with that ability exist and are cited in the introduction: CMF, ECHSE, SUMMA and RAVEN. What problems exist with these frameworks, and how does SuperFlexPy solve these problems?*

As noted in MP1, SuperflexPy is the first open source implementation of Superflex, which already has a series of case studies that demonstrate its utility. For this reason, our focus in this paper is on presenting SuperflexPy, focusing on its implementation and usability.

We note that from a conceptual perspective, Superflex has its unique features compared to other frameworks, and notably its fine granularity in terms of model components, which enables to encompass a wide variety of conceptual model applications. The merit of the proposed software

is to realize such broad coverage of conceptual model applications, while maintaining ease of configuration and operability. We believe that these characteristics set SuperflexPy apart from existing software.

In our presentation (lines 157-171), we review existing models based on several desirable requirements of flexible frameworks, including:

- spatial flexibility
- ease of use
- possibility of extension and modification
- computational efficiency
- potential of go beyond just water fluxes

SuperflexPy is designed to target these requirements collectively, while other frameworks may be based on different philosophies. In lines 172-180, we describe how the considered existing frameworks achieve (partially) the desired features.

As noted in MP2 earlier, it is not feasible to provide a detailed comparison of existing frameworks, which would require several additional analyses. This aspect will be clarified in our revised paper, where we will stress that our exposition of existing frameworks is mainly intended to discuss their complementarity to SuperflexPy in relatively broad terms.

*RC1.2: The authors state in l. 210 as an aim for the ms: "Provide a broad discussion of how the SuperflexPy contributes to the toolkits available to the hydrological community, including existing flexible frameworks, in terms of intended scope of application, advantages, and limitations", but mention the capbilities only very briefly in the introduction. This should be expanded and moved to the discussion to include SuperFlexPy in the comparison. The current sections 5.1 and 5.2 seems to be a good place (see specific comments).*

We agree this is a pertinent point. Please see the detailed response to MP2.

*RC1.3: As author and maintainer of CMF I can give my view on the differences between CMF and SuperflexPy (references for information only, not intended as suggestions for the reference list of the manuscript) and would be very interested about differences and similarities with other frameworks.*

We thank the reviewer for these detailed comments and insights, which will help improve our paper. Specific replies are detailed below.

*Similarities*

- *Model is composed as a Python script*

    Agreed.

- *Well defined coupling with optimization / rejection framework (in both cases SpotPy)*

Agree. In our case, we used SpotPy to illustrate an example of interfacing with a calibration framework. Many other inference frameworks can be coupled, depending on the user needs (but we guess this is the case also for CMF).

- *Object oriented design*

Agreed

- *Interactive exploration of model behavior (via Python prompt)*

Agreed

- *Unrestricted possibilities for spatial granularity*

Agreed

- *Fine process granularity*

Agreed

- *Open source, open access*

Agreed

- *Available via PyPI*

Agreed

- *User defined time stepping scheme*

Agreed

*Features that SuperFlexPy has, but not CMF:*

- *Lag functions*

Agreed

- *Toolbox extendible by users without compiling (only experimental and unpublished feature in CMF)*

Agreed

Moreover, we see these features

- 100% Python with no wrapping of other languages (e.g. C++)
- Explicitly designed with a future community development in mind, with users using SuperflexPy to create new model components for their needs.

*Features of CMF not present in SuperFlexPy*

- *Calculation methods for ETpot (eg. Jehn et al., 2017)*

  We agree this could be useful. However our philosophy is that computation of data represents pre-processing (relative the hydrological model), and is best done offline or through dedicated libraries.

- *Implicit and explicit single and multistep solvers (eg. Kraft et al., 2011)*

  Currently, SuperflexPy provides the implicit and explicit Euler solvers. The numerical solution of the differential equations is decoupled from the specification of the equations themselves, which makes it easy to add new solvers without changing the rest of the SuperflexPy framework.

- *Multiprocessor support for large model systems (eg. Wlotzka et al., 2017)*

  We agree computational efficiency is very important. Given the focus of SuperflexPy on conceptual models, which run in seconds / fractions of seconds, we find "inner" model parallelization less effective than "outer" parallelization. In particular, parallelization can be used at the calibration/inference level: libraries like SpotPy can potentially run multiple independent instances of SuperflexPy in parallel when inferring the parameters (e.g. multiple parallel Markov chains). This will be clarified in the revised Paper and Documentation.

- *Complex topology (cyclic bidirectional graph) (eg. Maier et al., 2017)*

  Agree. This is a limitation that we expressed and discussed in Section 5.2 together with possible ways to overcome it. As detailed as part of responding to point RC1.33 below, the existing discussion and documentation will be enhanced with additional material on this topic.

- *Energy potential based flow descriptions (eg. Richards- and St. Venant equation etc.) (Maier et al., 2017; Windhorst et al., 2014)*

  Agree. SuperflexPy is specifically targeted to conceptual hydrological models, therefore to models that can be represented by reservoirs and lag functions. We therefore consider such equations as out of scope.

- *Tested solute and isotope transport (eg. Haas et al., 2013; Windhorst et al., 2014)*

Agree. However, please refer to MP5 for further details on this topic.

- *Explicit run time loop for simple model coupling (eg. Kellner et al., 2017; Kraft et al., 2010)*

Agreed.

**1.2 Missing mathematical explanations**

Please refer to MP3 as general reply to this comment

- *RC1.4: The term numerical approximator is not well defined.*

Agreed. We will add a brief definition in line 513 and refer to the documentation.

- *RC1.5: What happens if the root finding procedure does not converge? Flexible time stepping or does the implementation stop with an exception? Typically happens with fast snowmelt or power law equations with a large exponent.*

We agree this is an important point. In the Python implementation, we raise an exception; when using Numba, we return `None` because Numba does not support exceptions. In both cases user notices the problem (either the simulation crashes or the result is plenty of `None` values).

- *RC1.6: The standard implicit euler method implementation with the Pegasus root finding algorithm should be explained briefly*

Good point. We will provide citations to publications that provide detailed descriptions of these methods. A description of these methods will also be added to the documentation.

- *RC1.7: How do the solvers deal with discontinuous or not continuously differentiable flux equations? The problem is described by Knoben et al 2019's MARRMoT Paper, Ch. 2.4( https://doi.org/10.5194/gmd-12-2463-2019) - it is the reason why I gave up mimicking exisiting models with CMF.*

This is a pertinent point. Generally speaking the SuperflexPy philosophy is to use smooth flux functions. This may include applying smoothing to otherwise discontinuous formulations – please see previous publications such as Kavetski and Kuczera (2007).

That said, if a user wanted to perform modelling experiments with discontinuous flux functions, the framework enables to do so. The EE solver can work with non-smooth RHS of the differential equations, whereas the IE solver requires smooth equations. Users could also integrate in SuperflexPy their own solvers with more specialized techniques for non-smooth problems.

These points will be noted briefly in the revised paper and documentation.

- ***RC1.8:*** *The system can use for the solution of single elements implicit solvers - the need for that was very well explained by the co-authors in their "ancient daemons paper". How the solutions of the elements are combined to a the response of the entire model is not explained. I guess, that some kind of operator split is employed, but how do they deal with non linear behavior between timesteps? Is the numerical error of the operator split somehow controlled? Is there some lag of fluxes that are routed through lower nodes in a network?*

The SuperflexPy framework is built on a model representation that maps to a directional acyclic graph. Model elements are solved sequentially from upstream to downstream, with the output from each element being used as input to its downstream elements.

When fixed-step solvers are used, this "one-element-at-a-time" strategy is equivalent to applying the same (fixed-step) solver to the entire ODE system simultaneously (i.e., no additional approximation error is introduced). This is one of the pragmatic reasons we favor the fixed-step implicit Euler scheme.

When the solvers use internal substepping, then the "one-element-at-a-time" strategy does introduces additional approximation error. This additional approximation error is due to treating the fluxes as constant over the time step, whereas the exact solution would have varying fluxes within the time step. However, in most practical applications, this "uniform flux" approximation is already applied to the meteorological inputs (rainfall and PET), hence applying it to internal fluxes does not represent a large additional approximation.

The option to solve the system of equation jointly would avoid the "constant flux" approximation for the internal fluxes (but not for the meteorological one). However, the gain in accuracy is expected to be small and come at the expense of a considerable computational effort and additional code complexity.

We agree these details are pertinent – they will be explained in the Documentation and a cross-reference will be added to the Paper.

If individual elements have multiple outgoing fluxes (e.g., streamflow and evapotranspiration), these are calculated simultaneously by solvers such as IE, and there is no need to specify an order for how such outgoing fluxes are calculated (it is however necessary if EE is used).

**1.3 Programming interface**

*The programming interface has a number of quirks and behaviour outside the norms of the Python language. I guess it is unusual to request changes to the programing interface of the software presented, but both manuscript and software would be improved. Not knowing and*

*following the Python standards was one of the biggest mistakes I did with CMF, and it is very difficult to change the programming interface later.*

We appreciate this feedback by the reviewer. We certainly agree that adherence to programming conventions (here, of Python) is very important to facilitate code understanding and further development.

*RC1.9: Leading underscore:*

*In all code examples, where behavior of superflexpy components is extended / changed (polymorphism for object oriented programmers) the authors of the framework indicate with a leading underscore "something". A leading underscore of a class member has a clear and well defined meaning in the Python community:*

*"_single_leading_underscore: weak "internal use" indicator. E.g. from M import \* does not import objects whose names start with an underscore." (https://www.python.org/dev/peps/pep-0008/#public-and-internal-interfaces)*

*Internal use means, class members with a leading underscore should nearly never be assessed by a user of the framework, neither for reading, nor for writing. The authors seem to understand the leading underscore to indicate the concept of a "protected" member in java, C# or C++, a concept that does not exist in Python.*

This is a subtle point.

We believe that it is important to have a convention to signal to the users/developers which methods are designed as "public interface" and which ones are for internal usage/inheritance.

We indeed use the leading underscore to indicate a "protected" method. The concepts of "public", "protected", and "private" are quite fuzzy in Python.

Although the language does not enforce these concepts, it is a weak convention to use a single leading underscores to indicate protected and private methods. The cited PEP-8 (Python style guide) proposes to "Use one leading underscore only for non-public methods and instance variables." (https://www.python.org/dev/peps/pep-0008/#id47). Several blog-posts (https://stackoverflow.com/questions/36967787/how-to-represent-protected-methods-in-python-classes; https://radek.io/2011/07/21/private-protected-and-public-in-python/) indicate this as a convention (we are aware that blog-posts do not represent an official Python guideline but, given the absence of such concepts in the official guideline, they still signal the orientation of the community on this topic).

For this reason, we believe the current usage of leading underscores is helpful and does not contradict Python conventions.

*Usage of literals to access parameter and state names instead of keyword arguments or class properties makes usage and composition of the model components harder to write. If "magic" literals are avoided, modern Python IDE's (integrated development environments) can help with code completion. If the framework can allow one of the following two alternatives to create a component, the IDE can help with code completion, instead of all purpose dictionaries. For network creation see comment l.469*

```
linear_reservoir = LinearReservoir(
k=0.1, S0=10.0,
approximation=num_app,id='LR'
)

linear_reservoir=LinearReservoir(approximation=num_app,id='LR')\
.parameters(k=0.1)\
.states(S0=10.0)
```

Interesting alternative - we agree that it would facilitate the usage of IDEs.

However a downside is that it would also require the definition of specific methods to set and get states and parameters when creating new elements. In this example, the "parameters" method of a LinearReservoir would have "k" as argument while the one of an "UnsaturatedReservoir" would have 4 arguments.

The definitions would therefore be
```
class LinearReservoir(...):
    ...
    def parameters(k):
        ...
    ...

class UnsaturatedReservoir(...):
    ...
    def parameters(Smax, Ce, m, beta):
        ...
    ...
```

In the current configuration, the usage of dictionaries allows the presence of generic methods for setting and getting states and parameters that are inherited by the new elements, which makes the implementation of new elements easier.

An alternative would be the usage of `**kwargs` in those methods but it would not solve the problem of the IDE not helping with code completion and checks.

*RC1.11:* *The Splitter interface relates to the position of certain input datasets in a list, that does not exist explicitly. This is quite hard to follow and to read.*

The definition of the connectivity of the splitter relies on the relative position of the downstream elements. This choice is a consequence of the definition of the structure of the unit as a succession of layers where the order of the elements already matters. The definition of the splitter is made even more complex by its design for handling multiple fluxes.

*RC1.12: The definition of the topology (falsely called "topography") of the Network class is redundant and uses the id-string literals instead of the node objects directly. A very easy to read variant for the creation of the Network is explicit setting of the downstream node:*
```
stgallen.downstream = Appenzell
```

In principle, the Network class could also not exist and the framework could incorporate the connectivity in the nodes directly, as suggested by the reviewer. However, we have decided to keep the Network class in order to facilitate model definition and usage. In particular:

- When calling the methods get/set parameters/states of the network they propagate the call to all the nodes
- When running the network it has already incorporated the right order of execution of the elements
- Commands on the network are agnostic of the connectivity of the nodes

*RC1.13: Another option would be to define the tree as a nested list structure. Each list contains the node and the left and right upsteam nodes, if present.*
```
thur_catchment = Network([
    andelfingen,
    [halden,
        [stgallen,
            [appenzell]
        ],
        [jonschwil,
            [mogelsberg, mosnang]
        ],
        [herisau],
        [frauenfeld,
            [waengi]
        ]
    ]
])
```

*Most modern programming environments (IDE) can mark typos in the node names with both versions presented here, and even employ code completion to help the user with typing, but this does not work for meaningful literals.*

This is a valid alternative to the representation of the network topology. We personally do not see a clear advantage in terms of readability compared to the usage of dictionaries indicating, for each node, the downstream one. An example of the current solution is here reported.

```
{
    'andelfingen': None,
    'appenzell': 'stgallen',
    'frauenfeld': 'andelfingen',
    'halden': 'andelfingen',
    'herisau': 'halden',
    'jonschwil': 'halden',
    'mogelsberg': 'jonschwil',
    'mosnang': 'jonschwil',
    'stgallen': 'halden',
    'waengi': 'frauenfeld'
}
```

**2 Specific Comments**

*RC1.14: 132: RAVEN is clearly a framework for conceptual models (no energy potential / head based flow equations) for internal transport but CMF was originally developed for physically based models.*

Thank you for the clarification. We will clarify this point in the paper.

*RC1.15: 171 - 180: Here is a description of what RAVEN and SUMMA do missing.*

We describe only the frameworks that belong to the scope of SuperflexPy. According to our classification, SUMMA is more towards physically based/fully distributed models. Regarding RAVEN, we will change its classification and add a brief description as done with other frameworks in the scope of SuperflexPy.

*RC1.16: 180: CMF can model substance / isotope advective transport with adsorption without additional software, and reactive fluxes in coupled model approaches*

Thank you for the clarification. We will clarify this point in the paper.

*RC1.17: 187: Same fine granularity as CMF, RAVEN and MARRMoT.*

The paragraph that contains line 187 is only about SuperflexPy being built on principles defined by Superflex. There is no intention in this paragraph to include/exclude alternative frameworks.

*RC1.18: 210: See general comments*

Please see MP2.

***RC1.19:*** *213 - 218: Link sections with the numbered aims of the paper*

We will add this information in the paper. Note that the three aims are addressed by multiple sections. Aim 1, for example is in sections 2 and 4, aim 2 is in sections 2 and 3, aim 3 is in sections 1, 5, and marginally in all the others..

***RC1.20:*** *225: "An element can represent an entire catchment. . . ": From my understanding this might be technically true, if the catchment can be represented by a single process. However, the meaning of "element" in SuperFlexPy is simpler to understand if 225 is changed to: "An element represents a specific process within the catchment." If needed this sentence can be extended by "In special cases, this specific process covers the entire catchment behavior and a single element is sufficient for the model."*

Thanks – we agree and will replace "process" with "response mechanism".

***RC1.21:*** *Eq1: To ensure mass conservancy, how can g_s ever be different to dS/dt = X(t)-gy(S(t);X(t); ?)?*

This notation follows the original notation used in the Superflex papers. Sometimes it may be necessary to modify the input fluxes (e.g. using a multiplicative constant to the precipitation to adjust for measurement errors) and therefore the current formulation is more appropriate.

***RC1.22:*** *230: This is a bit unclear: does SuperFLEXpy support substance transport? If yes, only theoretically, or has it been tested already?*

Please see MP 5 - we will remove "substance concentration" from the parenthesis to avoid confusion.

***RC1.23:*** *234: How would a multistate reservoir look like (mathematically)?*

Reservoirs with multiple states are represented by a system of differential equations like equation 1 in the paper. For brevity, they can be written in vector form, where S is a vector of states instead of a scalar.

***RC1.24:*** *242: Please give an example or more concrete description, what a connection is in terms of hydrological processes. As they are needed later, please explain the splitter, junction and transparent elements here.*

We agree such details will help the reader. We will add a brief explanation and cross-reference to the documentation, which has 3 pages explaining the connections.

***RC1.25:*** *260: Here only weight is mentioned (as the area fraction) while in 310 weight and area are different things. Please explain more consistent*

The `weight` of a given unit represents the of the node outflow that is contributed by that unit. The weight can be thought as the areal fraction occupied by the unit, but other interpretations are possible (e.g. to give a higher importance to a certain mechanism).

The `area` of the node is a property of the node that is used by the network when calculating the contribution of different nodes.

We will clarify these points in the paper.

*RC1.26: 274: The Level 3 concept predates the given references by a long time, please use more classical example (eg. TOPMODEL, HBV (not light), etc.)*

We agree, and will clarify this point in the paper.

*RC1.27: 276: Same as above, please give a reference to classic "Level 4" model, eg. SWAT, SHETRAN, etc.*

We agree, and will clarify this point in the paper.

*RC1.28: 295: This is a structure problem: You need the numerical solution of the ODE for the construction of the model, but the numerical solution is not yet explained. Please consider moving Section 4.3 up as a new section 2.2.*

Please see MP4.

*RC1.29: However, section 4.3 does not explain the terms "numerical approximator" and "root finder" and how these work together.*

We will explain these terms in the revised paper.

*RC1.30: Secondly, the "numerical approximator" is given as a parameter to each reservoir element. How many instances of numerical approximators exist?*

Only a single instance of the numerical approximator is needed.

It is assigned to each element because potentially reservoir 1 and 2 can use two different approximators. We will reserve this detail to the documentation.

*RC1.31: One global object to solve the entire ODE-system or is the numerical approximator copied and is a non-shared object of the reservoir? Or is the numerical approximator more a kind of function without any notion of specific state?*

The numerical approximator object is not copied. It is defined as an object (needed because we use more than one method internally). Although it could be thought of as a function, this would not be formally correct.

*RC1.32: If each element is solved on its own, how is the whole system integrated, i.e. how does the operator split work? As the "standard" solver is the own implementation of ImplicitEuler with Pegasus root finder a short explanation of the math behind it is missing. How does the composition of integration solution work?*

Please see point RC1.8.

*RC1.33: 305: The definition of the routing is quite obscure. Somehow the routing is derived from a List[List[Element]]. Since "explicit is better than implicit" (Zen of Python) an alternative (obvious) way to define routing would be preferable. Otherwise some more explanation, how the nested list translates into a tree structure is needed.*

This is an important point. We agree that it is very important to have clear definitions of model components.

As detailed on line 250 in the paper, the structure of the unit is a directional acyclic graph (DAG). There are several ways for defining a DAG (e.g. connectivity matrix, adjacency list, incorporating the connectivity information in the elements, or as a list of layers).

For comparison, here is how the structure of the Unit of model M02 would be represented with the different methods.

- Connectivity matrix (here as a table): columns and rows represent elements. The cell (i, j) (i=row, j=column) is 1 if the element i is connected to (i.e., flows into) the element j (direction matters)

|  | UpSpl | WR | UpJun | UR | LowSpl | Lag | SR | FR | UpJun |
|---|---|---|---|---|---|---|---|---|---|
| UpSpl | 0 | 1 | 1 | 0 | 0 | 0 | 0 | 0 | 0 |
| WR | 0 | 0 | 1 | 0 | 0 | 0 | 0 | 0 | 0 |
| UpJun | 0 | 0 | 0 | 1 | 0 | 0 | 0 | 0 | 0 |
| UR | 0 | 0 | 0 | 0 | 1 | 0 | 0 | 0 | 0 |
| LowSpl | 0 | 0 | 0 | 0 | 0 | 1 | 1 | 0 | 0 |
| Lag | 0 | 0 | 0 | 0 | 0 | 0 | 0 | 1 | 0 |
| SR | 0 | 0 | 0 | 0 | 0 | 0 | 0 | 0 | 1 |
| FR | 0 | 0 | 0 | 0 | 0 | 0 | 0 | 0 | 1 |
| UpJun | 0 | 0 | 0 | 0 | 0 | 0 | 0 | 0 | 0 |

Please keep in mind that M02 is a relatively simple structure; representations like the connectivity matrix become increasingly "bulky" and cumbersome for model structures with more elements (e.g., a structure with 20 elements will yield a 20x20 matrix with mostly "0" entries).

In addition, note that this definition may lead to structures that are not DAGs

- Adjacency list: for each element specify the downstream(s). It can be defined as a Python dictionary

```
{
        'UpSpl': ['UpJun'],
        'WR': ['UpJun'],
        'UpJun': ['UR'],
        'UR': ['LowSpl'],
        'LowSpl': ['Lag', 'SR'],
        'Lag': ['FR'],
        'SR': ['LowJun'],
        'FR': ['LowJun'],
        'LowJun': [],
}
```

This definition may also lead to structures that are not DAGs.

- Incorporating the connectivity information in the elements would require adding to each element an attribute that records its downstream elements. The concept is quite similar to the use of the adjacency list. Therefore, after the initialization of an element (or at initialization), we would have to define its downstream element(s). For the WR, this would be:

```
wr.downstream = ['UpJun']
```

And so on for the other elements.

- A succession of layers, which is the approach used by SuperflexPy:

```
[
        [upper_splitter],
        [snow, upper_transparent], [upper_junction],
        [unsaturated],
        [lower_splitter],
        [slow, lag_fun],
        [lower_transparent, fast],
        [lower_junction]

],
```

We believe that the definition of a DAG as a succession of layers, while bringing some drawbacks (e.g. the necessity of transparent elements), has also advantages: it stays relatively simple with a growing number of elements (especially compared with the connectivity matrix), it gives an immediate idea of the connectivity, it is easy to insert/delete internal layers, etc. Note also that such representation is common in other applications like, for example, neural networks.

The use of `List[List[Element]]` is motivated by the internal lists representing the single layers and the external collecting all the layers.

We will incorporate part of the discussion above in the revised contribution (probably in 5.2 or in the discussion and in the documentation) to make design choices more explicit.

*RC1.34: 310: What unit is used for area?*

SuperflexPy is agnostic towards the units that are used: it does not have functionalities that deal with unit checking and conversion. It is up to the user to keep the units consistent (e.g. all the fluxes in mm/h, all areas in km2, time step consistent with the fluxes, etc.). Internal unit checks/conversions can be computationally costly, and in our opinion they are best implemented as pre-/post-processing operations taking advantage of specialized libraries (e.g. Pint - pint.readthedocs.io).

*RC1.35: 315: Topography is not the right term here, do you mean topology? The dictionary with id's to define the topology is not helpful when using IDE's with code completion*

We agree - "topology" is the correct word here. Please see points RC1.12 and RC1.13.

*RC1.36: 392: "Input fluxes" is ambiguous here: Input can either mean the direction of the flux (input flux adds water to the element) or in the information sense: input is an externally defined time series that may add or take water from the element. This ambiguity needs to be addressed in the whole manuscript and the documentation.*

An input is intended as the term X in eq 1 of the paper. As noted by the reviewer, depending on the element, an input can be an actual flux (e.g. precipitation) or a variable that conditions the behavior of the element (e.g. temperature for the snow reservoir). Please keep in mind that X in eq. 1 accommodates both cases.

*RC1.37: 399: User facing methods should not start with an underscore. There is no concept of "protected" in Python (see general issue 2)*

A general reply to this issue is given to RC1.9.

Specific to this comment, given the distinction between a "developer" (i.e., a person who wishes to modify SuperflexPy) and a "user" (i.e., a person who applies the model without modifying it), those methods are "developer facing". For developers, we believe that the usage of the leading underscore is appropriate in this context to indicate protected methods.

*RC1.38: 411: ditto*

Please see point RC1.37.

*RC1.39: 427: Is there a better symbol for the snow reservoir than WR? What is W?*

"W" stands for "Winter" and "snoW". We agree that it might not be "ideal", ultimately we are limited by the alphabet. In the specific example, we follow the notation of Dal Molin et al., 2020 and the notation in numerous Superflex publications where the subscript "S" is already used to denote the slow reservoir "SR".

*RC1.40: 443: Here is an explanation of splitters and junction missing, because the "gaps" can only exists together with the multipath structure (or better: explain them in section 2).*

Good point. We will improve section 2 and add reference to the documentation where these details are explained.

*RC1.41: 448: What is the role of the upper_splitter here?*

As explained in point 2 (lines 454-457) the upper splitter is used to transmit PET to the UR.

*RC1.42: 465: When a unit is composed from elements (why the term "layers"?), are those elements copied or referenced (must be copied, but is not mentioned in the manuscript)? If I count correctly, each unit has 4 states and 11 parameters. So each node has 8 states and 22 parameters?*

The concept of "layers" will become clearer after the revision of section 2.

The default behavior is to copy the elements entirely when they are put in the unit. When units are put together to create a node, states are copied and parameters are shared (all units point to the same dictionary). We have achieved this behavior by manipulating the copy behavior of the elements and of the units (`__copy__` and `__deepcopy__` methods). Details on this are given in the documentation.

The default behavior can be deactivated – in which case both parameters and states are copied when adding the unit to the node.

It is correct to say that a node with two units has 8 states and 22 parameters. But, due to the "sharing parameters" behavior, when 10 nodes share the same 2 units, the model has 80 states and still 22 (shared) parameters.

*RC1.43: 466: Same question here: does assigning a unit to a node make a copy of the unit?*

Please see point RC1.42.

*RC1.44: If yes, I would understand "consolidated" a kind of template. As the code is given, the parameters for consolidated and unconsolidated are the same, how do you change their parameters, if the units are copied? If copied, the model has 80 states and technically 220 parameters, that could be tuned independently, correct? Of course, the parameters for each of the 10 consolidated units might be coupled, to be the same, by convention.*

Please see point RC1.42. Given that parameters are shared, the method `set_parameters` can be from any node, e.g.

```
andelfingen.set_parameters({'unitID_eleID_parName': val})
```

and the change will propagate to all the nodes.

These details are presented in the documentation.

**RC1.45:** *469: cf. general issue 2*

Probably the reviewer means "issue 3". See reply to general comment 1.3 - Programming interface.

**RC1.46:** *490: An UML-class diagram of the main components would be helpful*

Good idea. We will add a UML-class diagram to the documentation.

**RC1.47:** *506 - 519: See comments to line 295*

We will clarify the concept of "numerical approximator" and cross-reference to documentation.

**RC1.48:** *540: In l. 163, "Interoperability with external software, for example for model calibration and uncertainty analysis" is claimed as one of the desired features of modeling code. This is a good place to mention how SuperFlexPy can be interfaced with calibration / uncertainty packages like Ostrich, SpotPy, PEST etc.*

Good idea. We will provide a brief statement in the paper and link to a detailed technical description in the documentation.

**RC1.49:** *546: This section would be much improved, if there would be a single working prototype in the documentation for it, otherwise it should be shortened substantially and its content moved into section 5.3 (future development).*

Please see MP5.

**RC1.50:** *559 - 584: The missing manuscript aim 3 should go here. Section 5.1 can be enhanced by citing articles dealing or struggling with the "right complexity" across framework boundaries*

See reply to general comment 1.1 - Missing discussion.

**RC1.51:** *585 - 605: How do the limitations of SuperFLEXPy compare to the other hydrological frameworks, especially the similar ones: MARRMoT, RAVEN, CMF, ECHSE? (See general comment)*

See reply to general comment 1.1 - Missing discussion.

*RC1.52: 654: Missing link to the Github repository. Figure 12 should be deleted, as it conveys only little information. The role of binder for the framework is unclear.*

We will add the missing Github link. We will remove figure 12.

*RC1.53: 658: Please write the DOI out here. BTW, the subtitle of the code release "The flexible language of hydrological modelling" (emphasis mine) is a bit bold, given that similar frameworks / domain languages are available since a decade.*

We will remove the reference and put the DOI of the code release.

*Figures:*

*RC1.54: Fig 2: See general comments about literals*

Ok

*RC1.55: Fig. 5: Users should not access class fields starting with an underscore (see general comment 3). Please use comments to explain l.8 and l.9. Is self._fluxes and self._fluxes_python callable or are these values?*

We will do that. Yes, they are callable.

*RC1.56: Fig 6: see general comment 3.*

Please see point RC1.9.

*RC1.57: Fig 7: Catchment boundaries and HRU boundaries can be presented in a single map*

We believe that the proposed change would make the figure more difficult to understand.

*RC1.58: Fig 8: Why does "W" denote snow?*

Please see point RC1.39.

*RC1.59: Fig 11: Parametrization of the Splitters is unclear. Missing different parametrization of the two HRU templates (if they are templates in fact)*

Connections in SuperflexPy are designed to handle multiple fluxes for each element. Details are provided in the documentation.

*RC1.60: Fig 12: Not necessary, can be removed*

Ok

*3 References*

*(no need to include into manuscript)*

*Haas, E., Klatt, S., Fröhlich, A., Kraft, P.,Werner, C., Kiese, R., Grote, R., Breuer, L. and Butterbach-Bahl, K.: LandscapeDNDC: a process model for simulation of biosphere-atmosphere-hydrosphere exchange processes at site and regional scale, Landscape ecology, 28(4), 615-636, 2013.*

*Jehn, F. U., Breuer, L., Houska, T., Bestian, K. and Kraft, P.: Incremental model breakdown to assess the multi-hypotheses problem, Hydrology and Earth System Sciences Discussions, 1-22, https://doi.org/10.5194/hess-2017-691, 2017.*

*Kellner, J., Multsch, S., Houska, T., Kraft, P., MˇAˊzller, C. and Breuer, L.: A coupled hydrological-plant growth model for simulating the effect of elevated CO2 on a temperate grassland, Agricultural and Forest Meteorology, 246, 42-50, 2017.*

*Kraft, P., Multsch, S., Vaché, K., Frede, H. and Breuer, L.: Using Python as a coupling platform for integrated catchment models, Adv. Geosci, 27, 51-56, https://doi.org/10.5194/adgeo-27-51-2010, 2010.*

*Kraft, P., Vache, K. B., Frede, H.-G. and Breuer, L.: A hydrological programming language extension for integrated catchment models, Environmental Modelling and Software, 26, 828-830, https://doi.org/10.1016/j.envsoft.2010.12.009, 2011.*

Thank you for providing these references. We will integrate the 2010 and 2011 references as they are quite pertinent to the discussions.

**3  Response to comments by RC2**

*The paper titled "SuperflexPy 1.2.0: an open source Python framework for building, testing and improving conceptual hydrological models" by M. Dal Molin et al. details the development and implementation details of a new flexible hydrological modelling framework. The framework is based upon an earlier code SUPERFLEX developed by the second and third authors (Fenicia et al., 2011), but re-built using object-oriented Python programming approaches. The paper is generally well-written and appropriately structured, and the software is clearly the fruit of much labor and potentially worthy of its own publication.*

We thank the reviewer for their careful reading of the manuscript and their insightful suggestions.

*However, I am not sure that the authors have presented a sufficient argument as to the unique value of this contribution. I have a number of addressable concerns mostly with respect to evaluation and assessment of the framework:*

1. **RC2.1:** *Most importantly, I think the authors miss out on an opportunity (and expectation) to distinguish this effort from other modelling frameworks cited herein.*

   Please see MP2.

   **RC2.2:** *What makes SuperflexPy unique?*

   The desirable features of a flexible framework, which SuperflexPy aims to achieve, are listed in lines 157-171. Out of them, we believe that the focus on ease of extension and customization distinguishes SuperflexPy from alternative frameworks. See also point RC1.1.

   **RC2.3:** *What types of problems may it be applied to that other flexible frameworks cannot readily tackle?*

   In terms of applications, existing frameworks do partially cover the scope of SuperflexPy. SuperflexPy is built with the intention to cover the features listed in lines 157-171, while other frameworks may be built with different philosophies. Moreover, as discussed in MP1 and MP2, our intention is not to provide a detailed comparison, but rather to provide Superflex, which is already used in a variety of application, with an open source implementation. An overview of the status of existing frameworks is given in lines 172-180.

   **RC2.4:** *It would be useful to illustrate any perceived advantages via one or two case studies.*

Sections 2.2, 2.3, 3.1, 3.2, 3.3 present examples of usage of SuperflexPy with the objective of showing potential applications and the possibilities offered in terms of model customization. See also MP1 and MP2 where we describe the intended contributions of this work.

*RC2.5: In particular, the authors need to make a very strong case as to why this implementation is particularly advantageous relative to the original SUPERFLEX code, since the conceptualization seems very similar.*

As stated in lines 186-201, SuperflexPy builds on the concepts proposed by Superflex and proposes a new software implementation. SuperflexPy is the first open source implementation of Superflex. Therefore, there is no alternative Superflex software available to the community.

In our opinion, there is little value in comparing SuperflexPy with other Superflex implementations that are unpublished and not openly available.

Indeed, one of the reasons for software papers such as the GMD "Model description papers" category is to have "published" versions of software, which can then be compared in a systematic and reproducible way.

*RC2.6: Is it merely the object-oriented Python wrapper? If so, is this alone a sufficiently unique contribution for this journal?*

As clearly stated in line 610, "We stress that SuperflexPy is not a wrapper of earlier Superflex code but offers a completely new implementation". We also state that SuperflexPy is purely based on Python (line 608).

*RC2.7: As part of this, they will necessarily have to discuss some of the strengths and weaknesses of existing modular hydrology tools and what role SuperflexPy takes in addressing perceived gaps. This is the most critical comment for the authors to address.*

We agree this is an important question – see detailed response as part of MP1 and MP2.

2. *RC2.8: The authors refer a number of things that might someday be done using the framework but have not yet been implemented, which I found problematic. Specifically, they discuss transport of contaminants and isotopes and use of a more complex numerical solver. However, none of these advances have actually been implemented in this model. This content needs to be removed, as it is not a current advantage of the software tool, it is a hypothetical future advantage.*

Please see MP5.

In particular, the paper highlights that the design of the software is such that these applications (transport processes and more complex numerical solvers) can be easily implemented. Regarding numerical solvers, this design enables the introduction of new solvers without changing the model components.

We believe that it is relevant to highlight such possibilities since they are part of the "ease of modification" principle of SuperflexPy.

3. **RC2.9:** *The evaluation of the computational efficiency lacks rigour. There are insufficient model details to evaluate the computational benefits of SuperflexPy or the specific Numba implementation, and the speedup is quantified as improving from "a couple of minutes" to "a few seconds" of runtime. A quantitative assessment is warranted here if the authors wish to make a defensible argument regarding computational efficiency.*

Please see MP3.

4. **RC2.10:** *The authors have made a unique choice of coupling all of the constitutive laws for fluxes into a single element, i.e., use of the UnsaturatedReservoir() element implies use of the relationships in Eqns 6-8. This is quite different from what is seen in models such as SUMMA, MARMMoT, or RAVEN where the swappable "element" is the constitutive law rather than the collection of constitutive laws applied to one storage element. Can you justify this selection and/or discuss the implications of this approach as compared to the flux-based components? It seems like just swapping one of the constitutive laws for a storage element will often necessitate creating a new component.*

We agree this is an interesting point that merits elaboration. In our experience, making entire elements as a single entity has the advantage of creating a clear distinction between different elements and makes possible to document (docstring) properly some methods (e.g. set_input, get_output, etc.). As shown in section 3.2.2, the creation of a new element that is slightly different from an existing one requires very little code effort.

The change proposed by the reviewer can be made relatively easily and can potentially become part of a future release of SuperflexPy.

**RC2.11:** *Likewise, I would like to see a defense of the use of a fixed number of layers which necessitates the use of "transparent elements" and a clarification of the role of these layers – why are they even necessary? What problem do they solve?*

Please see point RC1.33.

5. **RC2.12:** *I recommend including a UML depicting the inheritance structure and currently implemented elements in the SuperflexPy code (or a subset of these in the UML with a list/table of elements elsewhere). It is very unclear how the breadth and quantity of options compares to other modular frameworks including the original SUPERFLEX.*

Please see point RC1.46.

*I have included most of my minor comments in the attached PDF file.*

Thank you for these detailed comments, which we have copied below for convenience. Please refer to the supplement submitted by RC2 for more context.

*RC2.13: L 173: I'm not confident this is true - there are some other very flexible frameworks which accommodate this. I would amend the term "only" unless you are very confident in this statement.*

The "only" refers to the list cited previously (137-146). We will reword to avoid confusion.

*RC2.14: L 227: The function of the reservoir element?*

We will add a brief explanation.

*RC2.15: L 245: I find it odd that this language is so heavily abstracted, rather than using hydrological language - the unit here seems to be equivalent to a hydrologic response unit, and the node to a subcatchment. Why not use this language of hydrology?*

This is an important point. Indeed, SuperflexPy is intended to provide an intuitive platform for hydrologists to implement their catchment conceptualizations.

Although most of the applications will identify unit with HRU, node with subcatchment, and network with catchment, we decided to keep the usage of SuperflexPy open to a wider range of application. For example, units and nodes can potentially be used in a grid-based model where, depending on the configuration, they would no longer be tied to their current interpretations as HRUs and subcatchments.

We will clarify this in the revised paper and documentation.

*RC2.16: L 249: this is unclear. And the "overall model structure" is the structure of the unit connections?*

In this context, "overall model structure" coincides with the structure of the unit. We will clarify the sentence.

*RC2.17: L 249: This is not the only restriction, judging by the transparent elements discussed below. It seems like a fixed number of "layers" is required when implementing paralell fluxes? This could be better described here.*

As explained in point RC1.33, the unit structure could be defined in multiple ways. The definition as a "succession of layers" has been chosen as our favored option, with important advantages that we believe outweigh the limitations.

Transparent elements are used to skip a layer in the structure, and hence avoid some restrictions in the range of supported model structures. More details on this are provided in the documentation. We will add a cross-reference to this information in the revised paper.

**RC2.18:** *L 252: not sure "upstream" to "downstream" is valid here when you might be discussing (e.g.,) snow to topsoil.*

The terms "upstream" and "downstream" used here do not refer to physical entities but to topological entities in the DAG. We could have adopted terminology from graph theory (e.g. "parent" and "child") however, we believe that that would have made things more difficult to understand.

We will clarify this in the revised paper.

**RC2.19:** *L 265: routes*

Thanks. We will fix the typo.

**RC2.20:** *L 269: introduction of "level" definition prior to use would make this easier to read, i.e., "We here can consider problem complexity in terms of levels which require increasing numbers of units, elements, and nodes"*

The concept of "levels" has already been defined prior to this sentence. See line 221: "The SuperflexPy framework has a hierarchical organization with four nested **levels**: "element", "unit", "node", and "network", collectively referred as "components"". However, we will clarify the text to avoid confusion.

**RC2.21:** *L 295: solve the ODEs associated with the reservoir elements?*

Yes. We will apply this change in the paper

**RC2.22:** *L 315: shouldn't this be topology??*

Yes. We will change the term in the paper and in the code.

**RC2.23:** *L 318: assuming this is in days. How are units generally handled in SuperflexPy?*

Please see point RC1.34

**RC2.24:** *L. 351: units would be appreciated here. I assume S is in mm and the fluxes in mm/day?*

Please see point RC1.34

**RC2.25:** *L 358: likewise - please include units*

Please see point RC1.34

*RC2.26: L. 365: then why confound things by labeling this with a P, which implies precipitation. Just write eqn 9 with Q^(UR)*

Precipitation inputs are generally referred with the letter P, and we decided to keep this convention when describing elements. In this particular case, the input happens to be the output of another reservoir, and therefore $P^{(FR)}=Q^{(UR)}$.

*RC2.27: L. 376: The authors have made a unique choice of coupling all of the constitutive laws for fluxes into a single element, i.e., use of the UnsaturatedReservoir() element implies relationships 6-8. This is quite different from what is seen in models such as SUMMA, MARMMoT, or RAVEN where the swappable "element" is the constitutive law rather than the set of constitutive laws. Can you justify this selection and/or discuss the implications of this approach as compared to the flux-based components? It seems like just swapping one of the constitutive laws will often necessitate creating a new component?*

Please see point RC2.10

*RC2.28: L. 384: odd to label this a standard approach? Perhaps a low-inheritance or 'from scratch' approach (as opposed to the second approach which more effectively utilizes the benefits of inheritance)*

Good point. We will rename these sections to 'Building a new element "from scratch" (general approach)' and 'Building a new element by enhancing an existing element'.

Note that the approach proposed in 3.2.2 is valid only if the extent of required changes is minor, and can have certain drawbacks - e.g. changes to the PowerReservoir will influence the behavior of the NewFastReservoir.

*RC2.29: L. 391: undefined*

This detail is explained in section 4.3.

*RC2.30: L. 396: is the ODE solver home-grown or using an existing Python package?*

As described in section 4.3, the "ODE solver" is composed by the "numerical approximator" and the "root finder". The "ODE solver" can be implemented either from scratch (as we did with IE) or by wrapping existing Python libraries. In the proposed application, the "ODE solver" is "home-grown".

*RC2.31: L. 400: why? Could this be detailed?*

Chapter 8 of the Documentation details how to implement the fluxes methods.– We will include a cross-reference to guide the reader.

*RC2.32: L. 400: what is the numerical approximator? What is the approximation?*

This is explained in section 4.3.

**RC2.33:** *L. 442: is this checked by the software?*

Given the choice to define a unit as a succession of layers, it is impossible to define a structure that is not a DAG. See also point RC1.33.

**RC2.34:** *L. 443: why is this constraint required? It seems unnecessarily complicated...*

Please see point RC2.17.

**RC2.35:** *L 449: the WR reservoir*

We agree the current text is confusing. We will replace "WR" with "the WR element".

**RC2.36:** *L. 449: delete "in fact"*

Agreed.

**RC2.37:** *L. 460: This fixed number of layers in all units seems to be a bit of a liability of this means of flexible representation, and one not present in other flexible frameworks. What does the layering help you to accomplish?*

Please see detailed response in point RC2.17.

**RC2.38:** *L. 471: not much discussion of routing at this point - is there a mechanism for routing time delay between nodes, e.g., a muskingum or diffusive wave approach? Or is this all instantaneous routing?*

This aspect is detailed in the documentation. We will include a cross-reference to help the reader.

**RC2.39:** *L. 476: should cite documentation*

This is done in the appropriate section "Code availability" at the end of the paper.

**RC2.40:** *L. 479: unclear how this is storing history rather than instantaneous snapshot of states. Could you clarify which is being stored. If history, this could get very memory-intensive for large applications with long model durations.*

We are saving the full history of the states. This is a part of the trade-off between memory footprint vs runtime of the model: saving the states allows for a rapid computation of the fluxes whenever they are needed but requires extra memory.

In terms of memory usage, we use 64 bits float arrays, which means that a year of hourly data takes 70 kB per state. A large model like M2 (80 states) requires around 5 MB per year of hourly data. We believe that model M2 is broadly representative of the typical scale of application of

SuperflexPy. However, more complex model structures and/or high resolution data will increase memory usage. We will consider this issue in future releases of SuperflexPy by adding, for example, the possibility of disabling the saving of the history.

*RC2.41: L. 488: I think it would likely be useful to have a UML diagram showing the currently supported component classes in the software.*

Agreed - please see point RC1.46

*RC2.42: L. 498: which frameworks require this?*

This is done in Superflex (unpublished distributed setup), for example. Probably also other frameworks do the same. We will verify it and, in case, we will consider removing sentence.

*RC2.43: L. 501: there is a potentially big cost to this...is this dynamic memory?*

Please see point RC2.40

*RC2.44: L. 507: "The mass balance relationships for Reservoir elements are described..."*

Thanks. We will change the sentence as proposed.

*RC2.45: L. 516: has this been done? It may not be particularly straightforward, and I would be reluctant to note this as an advantage until this work is actually done. Also, I'm not sure it is appropriate to comment on work that hasn't been done except under a discussion of potential future efforts.*

Please see point RC2.8 and MP5.

*RC2.46: L. 523: hundreds or thousands*

Thanks. We will implement this change

*RC2.47: L. 537: these experiments must be discussed - you can't just say "everything is faster" without discussing the benchmarking approach and or providing quantified results.*

Please see MP3.

*RC2.48: L. 541: isn't this also conditional upon number of elements?*

The assessment of the performance of SuperflexPy refers to a model with a few reservoirs (line 539). The answer to the question here is not straightforward: having more elements means having more code that Numba needs to compile and, therefore, a similar trade-off between simulation runtime and compile time will remain.

***RC2.49:*** *L. 543: for which model configuration? Calibrating how many parameters? for what model duration? There are not enough details for this statement to carry weight.*

Please see MP3.

In this particular context, the number of calibrated parameters is essentially irrelevant, because we are interested only in the pure SuperflexPy runtime sans the overhead of the calibration algorithm itself (e.g. to generate a new set of parameters).

***RC2.50:*** *L. 555: If this work has not been done, I don't think it is appropriate here to suggest that transport is readily supported. This section should be removed.*

Please see MP5.

Specific to this comment, the full sentence is "The available examples in SuperflexPy do not include transport processes. However, the framework architecture foresees this need, and has been designed to be **readily** extended to accommodate such processes."

The term "readily" refers to the possibility of extending SuperflexPy with new elements to represent transport processes and not, as the reviewer implies, that "transport is readily supported".

***RC2.51:*** *L. 592: I don't believe this is true. What about refreezing of liquid water in a snowpack? Capillary rise? Saturation excess runoff?*

These examples are indeed cited a few lines below the sentence in question. Such interactions can be incorporated inside a single element and, in our experience with Superflex, are minor and can sometimes be removed with different parameterizations (although introducing approximations).

***RC2.52:*** *L. 613; not yet in press?*

The paper is currently under review. We will update this information as soon as we have updates.

***RC2.53:*** *L 618: but what does it offer relative to other flexible frameworks? This is the big missing component from my perspective.*

Please see detailed response to MP1. Desirable features covered by SuperflexPy have been listed in the introduction (lines 157-171) and showcased throughout the paper.

***RC2.54:*** *L. 626: why not report in this document?*

A primary aim of SuperflexPy is to enable researchers to build and use customizable hydrological models. Model mimicry is a currently seen as a (relatively) lower priority, and in the interest of keeping the paper short is not reported.

However, we report such model mimicry examples in the documentation to show the process of reproducing an existing model with SuperflexPy. We will ensure appropriate cross-referencing between the paper and the documentation to make it easy for a reader to find the relevant information.

**RC2.55:** *L. 635: components?*

"Levels" is here is the correct word, but we will reword the sentence for clarity.

**RC2.56:** *L654: I don't see much value in this figure. I recommend removal.*

Ok. We will remove figure 12

**4  Response to comments by RC3**

*The paper "SuperflexPy 1.2.0: An open source Python framework for building, testing and improving conceptual hydrological models" by Dal Molin et al. describes the development and implementation of a flexible hydrological modeling platform. SuperflexPy is based on the earlier SUPERFLEX model, but uses the Python language and adds new features.*

*The paper is well-written and generally well-structured, and I believe that SuperflexPy can provide a valuable and flexible tool for hydrologists.*

We thank the reviewer for their careful reading of the manuscript and their insightful suggestions.

*But I have some concerns with how the authors frame and present the model.*

1. ***RC3.1:*** *In section 1.3, the authors state that their 3rd aim is to provide a broad discussion of how SuperflexPy contributes to the hydrological community and how it compares to existing modeling tools. The included discussion does very little to situate SuperflexPy with respect to existing models. The authors should include a discussion about the similarities and differences, as well as perceived advantages and disadvantages, of SuperflexPy with respect to existing model frameworks. Why should someone use this model rather than previously available tools?*

   This is a pertinent point - please see detailed response to MP2.

2. ***RC3.2:*** *On line 160, the authors state that a flexible model framework should cover substance transport modeling, but the version of SuperflexPy presented does not have modules to handle substance transport. It would be possible to add substance transport, but that's presented as a hypothetical rather than an existing part of the model framework.*

   Please see response to MP5.

3. ***RC3.3:*** *Beginning on line 162, the authors outline three computational criteria that a model should meet: ease of use, including interoperability with external software, ease of modification, and computational efficiency. Of these three, I think that only ease of modification is adequately addressed in the manuscript. Installation and operation are discussed, as the authors outline the strengths of Python as an object-oriented and commonly used programming language. But they do not discuss operability with external software, model calibration, or uncertainty analysis. There are references to recently published manuscripts that seem to cover these topics in more detail, but I believe that more discussion is warranted.*

   We believe that the three criteria are actually addressed in the paper and in the documentation:

- Ease of use is demonstrated through the examples: Sections 2.2, 3.1, and 3.3 show how to create models of different degrees of complexity. Installation is outlined on line 609. Further technical details can be found in the documentation (which will be better cross-referenced in the revised paper). The coupling with external packages is mentioned in line 280 and a practical example (Spotpy) is given in the documentation.
- Ease of modification, as written by RC3, is described in Sections 2.3 and 3.2
- Computational efficiency is briefly discussed in Section 4.4 (and further details provided in the documentation).

Please see detailed response to MP3 for the discussion about the balance of content between paper and documentation.

*RC3.4: Section 4.4 covers computational efficiency, but does not provide much quantification or comparison with other models. The authors state that Numpy and Numba provide a speed up compared to native Python of "factors up to 30" and state a difference between "a few seconds with the Numba implementation compared to a couple of minutes with native Python execution" but do not provide sufficient details about the structure or complexity of the model used to calculate these runtimes. A comparison to other modeling frameworks would be especially valuable.*

Please see MP3.

4. *RC3.5: I think that Sections 4.1, 4.2, 4.3, and potentially 4.5 should come before Section 3. These detail the computational structure of the model and would follow the initial description in Section 2 quite well.*

Please see MP4

*RC3.6: An expanded version of section 4.4 could then be incorporated into an expanded discussion comparing SuperflexPy to other model frameworks.*

Please see MP2

*Specific line comments are included below:*

*RC3.7: 125-128: This paper doesn't explore model comparison, and it's not clear to me how the latter half of this paragraph informs the manuscript.*

The sentence explains that a reason for using an FMF is comparing in a systematic way different model structures. The objective of the sentence is to support the necessity of FMFs. We will reword the sentence for clarity.

*RC3.8: 178: The term granularity is unclear*

The reviewer might be referring to line 188. The term "granularity" is used in the Superflex papers and in Clark et al. (2011). We will provide some explanation.

*RC3.9: 224: The terms element and unit are somewhat unintuitive. It's clear how nodes connect in a network, but it's not clear why a unit should be made up of elements.*

We appreciate this concern. Please see point RC2.15.

*RC3.10: 250. The acyclic directional graph seems like a significant drawback – backwater effects, capillary action, hyporheic flow, etc. are common hydrologic occurrences. How do other modeling platforms handle these?*

This is an important design consideration, discussed in depth in Section 5.2.

One solution to overcome this limitation is to define new elements that incorporate these interactions internally (e.g. multi-states reservoirs).

*RC3.11: 270: The definition of Levels should be introduced prior to this*

Please see point RC2.20.

*RC3.12: 315: Topography should be topology*

We agree – this term will be changed in the paper and in the code.

*RC3.13: 351: Are specific units required? A small discussion of how the model handles units would be helpful.*

Please see point RC1.34.

*RC3.14: 371-373: Unclear whether importing data as a Numpy array is required*

As detailed in the documentation section 4.5.5, the fluxes must be Numpy arrays.

In addition to performance reasons (numpy arrays are faster than lists), operations such as a1+a2 have a different behavior when applied to Python lists.

*RC3.15: 436: If nodes share units, does that mean that the unit is duplicated in each node?*

Partially duplicated (states) and partially not (parameters). When units are put together to create a node, states are copied and parameters are shared (all units reference to the same dictionary). We have achieved this behavior by manipulating the copy behavior of the elements and of the units (`__copy__` and `__deepcopy__` methods). The documentation provides a detailed description and illustration of this behavior – please see Sections 4.3, which will be cross-references in the paper. Please also see response to MP3.

*RC3.16:* *443-445: It's not clear to me why connectivity gaps would occur and why a transparent element is the only way to solve this issue.*

This is a pertinent comment - Please see point RC1.33.

*RC3.17:* *500: Are there memory concerns with retaining history, particularly for complex systems or long runtimes?*

This is also highly pertinent - please see point RC2.40

*RC3.18:* *528: Numba was mentioned before but is detailed for the first time here.*

We will explain Numba is a Python package allowing just-in-time code compilation.

*RC3.19:* *615:SuperflexPy used to construct and compare*

Thanks for finding this typo.

**5 Response to comments by RC4 (Prof. Riccardo Rigon)**

*The paper illustrates the new software superflexPy, a system for doing hydrological modelling at catchment scale, which originates from the previous Superflex adding to it a more appealing implementation and offering a easier access and improved usability within the Jupyter/Python interface.*

We thank the reviewer for his careful reading of the manuscript and his insightful suggestions.

*RC4.1: I think it is built on solid scientific premises and well deployed, even if I believe that overall its engineering has not the quality of a System Product (Frederick and Frederick, 1995). However it will be useful for many researchers in Hydrology. Its relatively easiness of use, its being based on Open Source tools, its effort to be object oriented, and its use of Jupyter infrastructures for the documentation and dissemination will encounter the favor of many users and researchers. It comes as one Python infrastructures, in which Landlab (Hobley et al., 2017, and https://landlab.github.io/ #/) is a mature example. From its pros is the freshness of the approach and its usability and the very flat learning curve (if we do not include the learning of all the tools for developing illustrated in Figure 12 of the paper).*

Thank you for the references. Existing frameworks (including both in hydrology and outside hydrology) are a firm point to take inspiration from.

*RC4.2: The paper is well written, well organised and requires very minor modifications. The main concerns I have regard the Figures, in which many symbols are incorrect, as I list below.*

*Another concern has to do with the traditional representation that is given in the paper of the model M2 and M04. As proponent of a different way to represent the hydrological models I believe that some further effort can be made as I mention below, even if this is a side issue. Essentially I think that the representation used in the paper does not actually shows the mathematics behind the model. One part that does not work as it is, for instance is the "lag" item used in the paper. As shown in Bancheri et al., 2019, and Rigon and Bancheri (2020) the lag functions imply the existence of a reservoir which remains, in this case hidden because its functioning is assigned through a travel time distribution. Therefore, willing to preserve the same type of representation used in the paper, the lag item should be promoted at the same graphical level than the reservoirs of which is just a different expression. In our, Extended Petri Net representation, (Bancheri and Rigon, 2020) the M02 and M04 would be represented as shown in the attached pdf. Having awareness of this is indeed important in this modelling because the strategy underneath SuperflexPy is the comparison of model structures for a better representation of catchment processes (Clark et al., 2011) and getting, for what is possible in lumped models, the "right answer for the right reasons" (Kirchner, 2006).Though my point is marginal to the economy of this paper I want the Authors and the readers to consider that having a proper visualization of the models is deemed necessary in view of the selection tasks and of implementing those extensions to treat contaminants that the Authors envision in the final parts*

*of the manuscript. In synthesis, I believe that the paper can be published with very minor modifications, even if I would prefer, but I cannot require that much, that they change their model representation.*

We understand the reasoning and the benefits of the proposed model representation and we thank the reviewer for the effort made in producing the illustrations. A systematic and complete representation of the model structure, as the one proposed by the reviewer, is clearly important.

To this end, we will improve the graphical representation of our model, by making more obvious the correspondence between the figure and the implemented model elements. In particular, we will better characterize graphical elements such as the lag functions and the connections, which are unclear in the current figure setup, and we will improve the consistency in the naming of fluxes and elements.

However, completeness in terms of graphical representation may come at the expense of readability, especially when the model structure becomes more complex. In our experience, hydrologists are often accustomed to see graphical representations of conceptual model structures in terms of conceptual elements such as reservoirs or lag functions, as this is how models have traditionally been presented. Therefore, our improved graphical representation will attempt to preserve clarity and consistency inasmuch as possible.

In particular, in the current paper, we will adhere to the practice to only represent "water" fluxes (e.g. figure 8) that contribute to the water balance (e.g. precipitation P, streamflow Q, and actual evapotranspiration E). We found it difficult to represent other input variables that influence the behavior of the elements (e.g. temperature, potential evapotranspiration) without compromising the readability of the graphical representation. We believe considering such input variables, together with introducing specific abstract symbols (e.g. squares, rounds, etc.) for different elements and fluxes, would increase the complexity of the figure, and make it harder to comprehend without a detailed inspection.

We do agree with the reviewer that more thought/effort should be given to the graphical representation of models, and we will continue thinking in this direction on ways to provide an intuitive yet complete graphical representation of conceptual model structures.

**Detailed comments**

***RC4.3:*** *Line 235 - As I mentioned in the main text, the lag function is nothing different from a reservoir from the point of view of the model structure. It is just that the reservoir dynamics is given a different way. E.g. Rigon and Bancheri, 2020*

We agree with the reviewer that a mathematical correspondence between reservoirs and lag functions can be established. However, we also realize that (i) both types of elements are in use in conceptual models; e.g., the popular HBV or GR4J models are composed of both reservoirs and

transfer functions; (ii) these elements correspond to distinct mathematical operations: reservoirs are described using ODEs, and lag functions are characterized using convolution integrals, and (iii) these elements require distinct computational implementations; for instance, reservoirs are (typically) characterized by a single state variable, whereas lag functions are characterized by a vector of state variables, and (iv) even if a series of reservoirs and lag functions can be mapped to each other, this is not always convenient or possible. For example, the lag function corresponding to a power law reservoir would have a state dependent shape, and a lag function such as the gamma function would correspond to a series of reservoirs (not to an individual reservoir).

Given these arguments, we prefer to distinguish the two concepts in our presentation.

*RC4.4: Line 257 - Is the node here what elsewhere is called "Hydrologic Response Unit" ?*

No, this is not the case. A unit can represent either an entire lumped model or an HRU (when aggregated with other units inside a node). A node is used to represent a subcatchment that incorporates (potentially) more units.

The difference is that units inside a node operate in parallel, whereas nodes inside a network operate in a tree structure. Another difference is in the possibility of sharing parameters among the same unit belonging to different nodes.

We will clarify the explanation of these SuperflexPy concepts in the paper.

*RC4.5: Line 276 - Among the models that generate Level 4 predictions, I would cite Formetta et al., 2014*

Thank you for providing this reference, which will be included in the revision.

*RC4.6: Line 440 - acyclic directional graph or directed acyclic graph ?*

Correct. We will fix the error (also elsewhere).

*RC4.7: Figure 3 - The right reservoir should be renamed S_{FR}*

Thanks – we will implement this.

*RC4.8: Figure 4 - The Figure suggests me to ask if there is any method to have, inside SuperflexPy, the list of the available reservoirs.*

Not yet. However, chapter 7 of the documentation lists all implemented elements. A method that lists the available elements may be difficult to maintain and automate given the community development approach that SuperflexPy is designed to support (where developers can add new elements, etc).

*RC4.9: Figure 8 - The bottom reservoir should be renamed S_{SR}*

We agree – thanks for the suggestion.

**RC4.10:** *Figure 9 - Why transparent layers are necessary ? Are not they a weakness in the software design ?*

This is a pertinent comment - please see point RC1.33

**RC4.11:** *Figure 10 - A curiosity here: how can parameters of the consolidated and unconsolidated can be distinguished. Is there any problem with identifiability ?*

Joint calibration to multiple outlets enables these model parameters to be identified. For more details, please see Fenicia et al. (2016).

**RC4.12:** *Further comments on lumped models representation*

*As side issue, I show here below the representation of the Model M2 with the Extended Petri Net*

[Figure]

*This Figure is actually intended to clarify the fact that the delay introduced by the lag function assume the existence of a reservoir. It is the yellow circle in Figure and I stress that is important*

*to account for properly when discussing of the models structure. The black frame on the flux is used to indicate that the budget of this reservoir is assigned through a travel time distribution (see also Rigon and Bancheri, 2020).*

[Figure]

*M4 model is much simpler though and possibly the EPN representation does not have any particular added value with respect to representation used in the reviewed paper. For a short introduction to the EPN, please see http://abouthydrology.blogspot.com/2020/10/introducing-extended-petri-net-by.html*

We thank the reviewer for their effort in producing this schematic. We agree there are several opportunities to improve the graphical representation used in the SuperflexPy manuscript. Please see point RC4.2.

**6 References**

Ammann, L., Doppler, T., Stamm, C., Reichert, P., & Fenicia, F. (2020). Characterizing fast herbicide transport in a small agricultural catchment with conceptual models. *Journal of Hydrology, 586*, 124812.

Astagneau, P. C., Thirel, G., Delaigue, O., Guillaume, J. H. A., Parajka, J., Brauer, C. C., et al. (2020). Hydrology modelling R packages: a unified analysis of models and practicalities from a user perspective. *Hydrol. Earth Syst. Sci. Discuss., 2020*, 1-48.

Clark, M. P., Kavetski, D., & Fenicia, F. (2011). Pursuing the method of multiple working hypotheses for hydrological modeling. *Water Resources Research, 47*.

Fenicia, F., Kavetski, D., & Savenije, H. H. G. (2011). Elements of a flexible approach for conceptual hydrological modeling: 1. Motivation and theoretical development. *Water Resources Research, 47*.

Fenicia, F., Kavetski, D., Savenije, H. H. G., Clark, M. P., Schoups, G., Pfister, L., & Freer, J. (2014). Catchment properties, function, and conceptual model representation: is there a correspondence? *Hydrological Processes, 28*(4), 2451-2467.

Fenicia, F., Kavetski, D., Savenije, H. H. G., & Pfister, L. (2016). From spatially variable streamflow to distributed hydrological models: Analysis of key modeling decisions. *Water Resources Research, 52*(2), 954-989.

Fenicia, F., Wrede, S., Kavetski, D., Pfister, L., Hoffmann, L., Savenije, H. H. G., & McDonnell, J. J. (2010). Assessing the impact of mixing assumptions on the estimation of streamwater mean residence time. *Hydrological Processes, 24*(12), 1730-1741.

Kavetski, D., & Kuczera, G. (2007). Model smoothing strategies to remove microscale discontinuities and spurious secondary optima in objective functions in hydrological calibration. *Water Resources Research, 43*(3).

---

## Author Comment (AC2) · 19 Mar 2021

Dear Reviewer,

Our replies are collectively reported as reply to EC1.

Thank you

---

## Author Comment (AC3) · 19 Mar 2021

Dear Reviewer,

Our replies are collectively reported as reply to EC1.

Thank you

---

## Author Comment (AC4) · 19 Mar 2021

Dear Reviewer,

Our replies are collectively reported as reply to EC1.

Thank you

---

## Author Comment (AC5) · 19 Mar 2021

Dear Reviewer,

Our replies are collectively reported as reply to EC1.

Thank you

---

## Referee Report (RR1)

The manuscript has been improved in several areas, but especially the mathematical description needs improvement.

MP1 and MP2:

The discussion issue (MP1, PK1.1) has been solved in the revision and the manuscript has been much improved, both by toning down the introduction and by expanding section 5. The same applies to MP2.

MP3:

This issue is not solved sufficiently and needs improvement prior to publication.

RC1.4: Still unclear why the term numerical approximator and not integrator or solver is used (as in the math-lit), but an improvement is available.

RC1.5: This was not meant as an implementation question: In cases of rapid, non-linear changes (eg Power-Law-Equation with an exponent > 4), implicit solvers often fail to converge for a specific time step – even A-stable solvers like the implicit Euler. Complex solvers (eg. RKF 45, CVODE and many others) use an adaptive time stepping scheme, which is, as the authors explain in their answer to RC1.8, not suitable for SuperFlexPy. This is important information and should be mentioned in the section 4.3

RC1.6: Reference is provided now, but the properties of the algorithm should be stated in the supplemental material (limits and speed of convergance). The algorithm is not explained or just described as a mixture of regula falsi with the secant method.

RC1.7: After careful reading of Supl-Section 5.1, I cannot find the information from this answer. The need for smoothing when using the implicit solver must be mentioned as a one-liner in the main text.

RC1.8: My concerns about numerical errors by the operator split are explained in the answer to the reviewers (RC1.8), but have not made it in the manuscript – neither in the main text nor in the supplemental material. In fact, both m/s and supplement are plainly wrong: m/s l. 514 suggest a free choice for the selected numerical solver and the supplement mat 5.1 suggests RK-solvers as an additional (not yet used) choice. However, RC1.8 explains me (but not the readers), that only single step Euler solvers are suitable to solve the system as other solvers would introduce the need for a formal integration of the fluxes over the (outer) timestep:

"When fixed-step solvers are used, this "one-element-at-a-time" strategy is equivalent to applying the same (fixed-step) solver to the entire ODE system simultaneously (i.e., no additional approximation error is introduced). " (from answer to RC1.8)

This section needs to make it in the main text of the manuscript, together with a reference for the claim.

RK-solvers of nth order use n-1 (or more) substeps to predict the final state at the output timestep by fitting an nth-order polynom into these substeps. Using the flux at Y(t, S(t)) or Y(t+1, S(t+1)) is not the correct number, as the solver calculates the ODE between these timesteps and introduces an uncaught numerical error into the system. While the user is free to use any Jacobian-free root finder, the choice of the ODE-solver is (obviously) limited to implicit and explicit Euler methods (PECE methods might also an alternative). There is no interface to calculate the Jacobian matrix of an Element, hence Newton-like root finding algorithms are not suitable. The freedom of the solver choice is quite limited by the use of the sequential solution of the DAG approach – this is of course valid, but should be made explicit.

MP4:

The classical structure of scientific writing is of course not directly fitting with a model description paper. However, I would see the choice of math, programming language and design principles rather as the methods of a model implementation and the resulting code and use examples as the results section. The new, frequent links to other sections, are a poor surrogate for a cleaner structure but are an improvement over the original manuscript.

Additional issues:

Section 4.2:

The m/s mentions 8 times the object oriented design of the implementation and but does not feature the object oriented design choices at a prominent place. The UML-diagram is now hidden in the last section of the supplemental material. The UML-like diagram should be moved to the main text in section 4.2, as it is essential for the understanding of the object oriented design. Now section 4.2 lists, how the OO-design helps to accomplish certain goals, but we, as the readers, can only guess what that OO-design is.

Fig 12: I am familiar with most services and software mentioned in Fig 12 (except binder), however, I had a hard time to understand it. Mixing cloud services like github, binder, zenodo, and read the docs with a file format (Jupyter-Notebooks) on an equal level does not help to understand any of these services. Having the developer and the user as the same person (symbol) complicates the understanding with the blue and black lines. The authors state, that explaining the ecosystem around SuperflexPy is important – while I do not follow the premise, explaining the services is possibly better done with text. But if the ecosystem is important enough for a large figure in the main text (while omitting the UML-Diagram), then the importance should be highlighted throughout the paper, as the object oriented design is. I still recommend to delete this figure. If the authors are absolutely sure, this figure is needed, they need to a) redraw the figure using two persons and kicking out Jupyter-Notebooks (as they are not a web service themself) and test the figure with friends from the intended audience, b) explain the figure in much more detail and the role of every mentioned service therein, and c) introduce throughout the paper the importance of a webservice ecosystem for modern model development (eg. mention in section 1.2, practical criteria). However, as of now, this figure is hardly explained, and someone who is not familiar with these services will not profit from it. Even worse, the figure in its current state is prone to misunderstandings and does a disservice to the paper. Moving the figure as is to the supplement material does not solve the issues mentioned above.

Jansen et al (2020) reference: This is unpublished work, and I as a reviewer am unable to check the content of this reference and review its role in the paper. The author's claim about its content might be wrong. As such, the reference needs to be removed. If a public preprint had been cited, this problem would not exist.

---

## Author Response (AR2)

**Response to the reviews of "SuperflexPy 1.2.1: An open source Python framework for building, testing and improving conceptual hydrological models."**

We thank the two reviewers and the Editor for their careful reading of the manuscript. Their additional insightful feedback and suggestions have helped us further improve the manuscript and address remaining issues.

In the remainder of this document, the original comments by the reviewers are in typeset in *blue and italics font* and our replies are typeset in black font. The two reviewers are referred to as PK (Dr. Philipp Kraft) and AR2 (Anonymous Referee #2).

**Response to comments by the Editor (Dr. Andrew Wickert)**

*Dear authors,*

*Thank you for your diligent revisions. After carefully reviewing both referee reports, as well as your response to the referees, I agree with them about some significant additional improvements that the manuscript will require prior to consideration for publication in GMD. Referee 1 has several substantial concerns, and Referee 2 suggests minor revisions but does have one major concern regarding the focus on transport vs. its apparent non-implementation in the current version of your code base.*

*I look forward to seeing a revised manuscript following your work to address these comments.*

We thank the Editor and the reviewers for acknowledging our effort in improving the manuscript addressing most of their earlier comments, which in particular helped us clarify the contribution of SuperflexPy and place it in a better context in relation to other hydrological models.

We appreciate the additional reviewers' comments, which we address in this document.

**Response to comments by PK (Dr. Philip Kraft)**

*The manuscript has been improved in several areas, but especially the mathematical description needs improvement.*

We thank the reviewer for a very careful review of the manuscript and for multiple detailed comments, all of them with clear technical merit. Motivated by the reviewer's comments on numerical aspects, we have made several enhancements to increase the functionality of SuperflexPy and better demonstrate its capabilities. In particular, we have implemented: (i) a new numerical approximator implementing the Runge-Kutta 4 method (within the constraints of the "constant-within-timestep" approximation of fluxes explained in point PK.2), (ii) a new root finder implementing a Newton-bisection method that uses the analytical derivatives of the fluxes.

We appreciate the interest of the reviewer in numerical aspects of hydrological modelling and, consequently, of SuperflexPy. As authors, we of course also share these interests, and have historically pursued some of them in previous publications. We have responded to all technical questions below, explaining the organization of SuperflexPy numerical implementation, how it is intended to operate and have provided a clearer description of its assumptions and limitations. This is certainly an important improvement to the presentation.

With that in mind, we now also note, both here and in the manuscript itself, that SuperflexPy is "primarily" intended for experimentation with the conceptual model structure, with "secondary" options to experiment with the numerical implementation. This choice is in line with the target audience of the paper (general hydrological/environmental modelers), as explained in point PK.11. For these reasons, in order to keep the manuscript focused, we have opted to preserve the overall balance of the presentation which focuses on these conceptual modelling aspects and their software implementation, and have avoided adding a large amount of numerical detail, which as we note below generally follows the recommendations from previous publications.

Further, some questions raised by the reviewer have prompted us to reflect more deeply on the assumptions made in "practical" numerical approximations. That is a large research topic in its own right – and while related to it is nonetheless distinct from the current paper, which focuses on the software implementation of a flexible structure model. As such, several of these questions deserve a separate study where they can be investigated – and reviewed – in appropriate depth. It would be an injustice to these questions to be tacked in somewhere in this paper, and presenting them without a suitably detailed (and therefore length) context could cause confusion to readers without the technical background of the reviewer. We hope the reviewer appreciates this perspective. In any case we once again thank the reviewer for eliciting these clarifications and reflection, which we agree and hope will reduce the potential for reader confusion. We also hope they can lead to follow up studies focused more specifically on numerical implementations.

We should also add here that we agree with the reviewer on the need for a better description and illustration (in Figure 12, now 13 in the re-submitted manuscript) of how SuperflexPy is integrated into the ecosystem of modern online software management tools, and on the need for
the UML diagram to be included in the main text (Figure 12 in the re-submitted manuscript). Our
responses to all these issues are provided below.

*MP1 and MP2:*

**PK.1:** *The discussion issue (MP1, PK1.1) has been solved in the revision and the manuscript has*
*been much improved, both by toning down the introduction and by expanding section 5. The same*
*applies to MP2.*

We thank the reviewer for acknowledging our effort in improving the paper with respect to these
points.

**PK.2:** *MP3: This issue is not solved sufficiently and needs improvement prior to publication.*

We agree that numerical aspects are very important in a hydrological modelling software. For this
reason, SuperflexPy is designed to provide a balance of efficiency and flexibility in the selection
of numerical solvers within the constraints imposed by the assumed model architecture (DAG).
We agree with the reviewer that some of these ideas were not so clear in the previous submission
- therefore we take the opportunity to clarify them here and in the revised manuscript.

Before moving to the specific comments, we describe in a consolidated way the numerics of
SuperflexPy and their relationship to the DAG assumption as well as additional approximations.

Our design of SuperflexPy is oriented towards facilitating experimentation with the conceptual
model structure, including the number and connectivity of storage elements, the shape and
parameterization of constitutive functions, spatial discretization, and so forth. For pragmatic
reasons, we have made some assumptions in the numerical approximation that, while robust in
their own right, do limit to some extent the numerical flexibility of the framework and have some
implications when techniques such as adaptive time stepping are used. We thank the reviewer for
identifying several of these limitations. Note also that these choices are based on our previous
research publications and general experience with flexible modelling frameworks, which
included detailed testing of multiple numerical algorithms (notably Clark & Kavetski, 2010;
Kavetski & Clark, 2010). These earlier studies have indicated that the implicit Euler scheme with
fixed time step is a robust choice for general hydrological modelling, which is the application that
SuperflexPy is designed for. Nevertheless, SuperflexPy does offer some flexibility in the choice
of numerical solvers, within the restrictions explained below.

SuperflexPy requires the conceptual model architecture to be defined as a directional acyclic
graph (DAG), which implies some restrictions in the coupling of equations. If the model structure
is not a DAG, then the model structure must be transformed in a DAG, e.g., by encapsulating the
part of the structure that contains feedbacks into a self-contained new element. This is already
discussed in section 5.2 of the paper and section 5.2 of the documentation.

Model structures without feedbacks (i.e., DAGs) offer several practical advantages, as already
elaborated in Section 5.1.1 of the paper. From the numerical perspective, such models lend
themselves to the simple "one-element-at-a-time" numerical solution approach, which reduces the
solution of an ODE system to the solution of a sequence of multiple scalar ODEs. Note that, if
the model structure is a DAG, the "one-element-at-a-time" approach per se does *not* introduce
additional numerical errors.

However, within the "one-element-at-a-time" approach, we make the (additional) numerical
approximation that the input fluxes into each element are constant within the model time step $\Delta t$.
This approximation is consistent with the typical format of hydrological data, such as rainfall,
PET, etc, which are tabulated in discrete steps (e.g., daily, hourly, etc). However, in our case we
also apply this approximation to internal fluxes. This pragmatic approximation enables a further
simplification of the solution procedure, because the output flux from each element becomes a
scalar value - however it comes at the cost of introducing additional first-order discretization
error, because the variation of internal fluxes within the time step $\Delta t$ is ignored.

These first order approximations do not impact on time stepping schemes that are first order
anyway (e.g., explicit/implicit Euler) and, which, at a given time step, use a single value of input
fluxes to estimate a single value of output fluxes. The lack of impact on first order schemes is an
appealing practical point because these time stepping schemes methods are commonly used in
hydrological models and indeed are recommended for their general robustness (e.g., Kavetski &
Clark, 2010). However, second and higher order time stepping schemes, as well as (adaptive)
substepping schemes are impacted – because these approaches require input flux values at
intermediate points within the time step. The impact of additional errors will reduce the overall
accuracy back to first order (with respect to the full exact solution). However, they would not
introduce any instabilities and indeed would still permit the advantages of adaptive time stepping
in terms of facilitating convergence of the nonlinear root finder (connecting to reviewer's specific
comment in point **PK.6**).

For completeness, we should also note that the "constant-within-timestep" approximation of
fluxes is not per se a direct requirement for the "one-element-at-a-time" strategy (nor of the DAG
assumption). Potentially, each element could output fluxes that vary within the time step $\Delta t$,
allowing for a reduction (or even elimination) of these additional flux averaging errors. A more
general implementation of SuperflexPy could adopt a different format for the fluxes – for
example, using (instead of a single number) a look-up array of values, a function, or another data
structure that allows for "time queries", etc. This approach would (potentially) re-enable higher
order schemes and adaptive time stepping schemes to reach their formal asymptotic order of
accuracy. However, we have not pursued these options in the current version of SuperflexPy,
because the asymptotic order of accuracy (i.e., the order of accuracy as $\Delta t \to 0$) is far from the
main concern when hydrological models are applied with input data resolution as course as daily.
Moreover, for say a Runge Kutta 4 solver to achieve genuine 4th order accuracy would require a
4th order approximation of the rainfall and PET time series, which as such as impossible with practical data. For these and other reasons we favour the current numerical implementation based
on the fixed step implicit Euler scheme, - indeed this implementation was used in all previous
SUPERFLEX-F90 case studies.

In summary, the numerical implementation within SuperflexPy has the following characteristics:

• If each element is solved using a non-adaptive first order method (e.g., implicit Euler
without substepping), then no additional approximation error is introduced; an
explanation is given in point **PK.7**.
• If an element is solved using a higher order method and/or an adaptive-step method, then
its outputs are averaged over the time step before they are used as the inputs to a
downstream element, which does introduce additional numerical approximation error. As
noted by the reviewer, this error will not be "seen" by the adaptive time stepping. This
limitation does not affect stability but impacts on the overall accuracy (truncation error).

Therefore, the SuperflexPy user can still employ adaptive time stepping and higher order
methods, albeit within the stated limitations. We agree these points are pertinent and were not
sufficiently clear in the previous response RC1.8 and in section 5.2 ("Sequential solution of the
elements") of the documentation. We have now added a brief description of these issues in
section 4.3 of the paper and section 5.2 of the documentation.

We now respond to the specific points raised.

*PK.3: RC1.4: Still unclear why the term numerical approximator and not integrator or solver is*
*used (as in the math-lit), but an improvement is available.*

Our implementation of SuperflexPy proposes a specific architecture for the (numerical) solution
of the ODEs, which considers two functionally distinct procedures, named the "numerical
approximator" and the "root finder".

The "numerical approximator" routine, which is an instance of the abstract class
`NumericalApproximator`, is responsible for creating a discrete **approximation** of the
differential equation. For example, when using implicit Euler, the numerical approximator
routine `ImplicitEulerPython` transforms the differential equation

$$\frac{\mathrm{d}S}{\mathrm{d}t} = P - kS^{\alpha}$$

into the algebraic function

$$f(S_{t+1}) = \frac{S_{t+1} - S_t}{\Delta t} - P + kS_{t+1}^{\alpha}$$

where $S_{t+1}$ and $S_t$ is the state of the reservoir at the end and beginning of the time step,
respectively; $P$ is the precipitation over the time step and $k$ and $\alpha$ are parameters.

The "root finder" routine, which is an instance of the class `RootFinder`, is then responsible for
finding the value of $S_{t+1}$ such as $f(S_{t+1}) = 0$. For example, SuperflexPy offers the root solver
`PegasusPython`, which implements the Pegasus algorithm.

The separation of the overall ODE solution into these two components simplifies the
implementation of new ODE solvers by allowing cleaner re-use of existing procedures. For
example, a numerical approximator can be used with a different root finders with no changes to
its code. In addition, the same root finder could be used with different numerical approximators.

Section 5.1.1 and 5.1.2 of the documentation indicate how to implement new numerical
approximators and root finders by extending the abstract classes `NumericalApproximator`
and `RootFinder`. When such architecture is adopted, the new code needed reduces to the
definition of the algebraic approximation of the ODE (for the numerical approximator) and to the
implementation of the algorithm for finding its solution (for the root finder) – i.e., avoiding the
need to implement all the (considerable) auxiliary code that is needed to actually solve the ODE
(e.g., looping in time, interfacing for the `ODEsElement`, etc.).

In terms of the choice of specific names "numerical approximator" and "root finder", we agree
that many potential alternatives could be possible. However, note that the numerical
approximator on its own does not actually **solve** or **integrate** the differential equation. For this
reason we prefer to not use the terms "solver" or "integrator", which in our experience have a
different meaning in the literature. Potentially, the name "integrator" could be assigned to the
combined usage of "numerical approximator" and "root finder" to solve the differential equation,
but in our opinion this is not really necessary and would just complicate the nomenclature.

As part of the manuscript revisions, we have enhanced section 4.3 of the paper and have
restructured section 5.1 of the documentation. In particular the new section 5.1.3 explains how to
implement a numerical solver for the ODEs from scratch, i.e. bypassing the numerical
approximator and the root finder architecture and interfacing directly with the `ODEsElement`.

*PK.4: RC1.5: This was not meant as an implementation question: In cases of rapid, non-linear*
*changes (eg Power-Law-Equation with an exponent > 4), implicit solvers often fail to converge*
*for a specific time step – even A-stable solvers like the implicit Euler. Complex solvers (eg. RKF*
*45, CVODE and many others) use an adaptive time stepping scheme, which is, as the authors*
*explain in their answer to RC1.8, not suitable for SuperFlexPy. This is important information and*
*should be mentioned in the section 4.3*

For clarity, the original question (RC1.5) was:

"What happens if the root finding procedure does not converge? Flexible time stepping or does the implementation stop with an exception? Typically happens with fast snowmelt or power law equations with a large exponent."

And our reply was:

"We agree this is an important point. In the Python implementation, we raise an exception; when using Numba, we return None because Numba does not support exceptions. In both cases user notices the problem (either the simulation crashes or the result is plenty of None values)."

We apologize for having mis-understood this question. The wording "does the implementation stop with an exception?" suggested to us it was a question about the behavior of the implementation.

We are aware that, in some situations, numerical solvers may fail to converge for a given time step size. As noted by the reviewer, adaptive time stepping in such cases would reduce the step size and attempt the step again.

In SuperflexPy, if the implemented fixed-step solvers (implicit or explicit Euler) fail to converge they do not fall back on other solvers (e.g., reducing the time step or changing the solver algorithm) but simply fail (i.e., raise an exception in the Python implementation or return `None` in the Numba implementation).

However we should add that the "one-element-at-a-time" strategy employed in SuperflexPy (see **PK.2**) enables the use of robust solvers that operate on a single ODE at a time. In such cases, the root finder also operates on a single algebraic equation at a time. Moreover, SuperflexPy proposes root finders that implement bracketing methods, which are guaranteed to converge (to a tolerance within the common constraints of floating point arithmetic) as long as the initial solution bounds are known. The bounds of the solution can be constructed from the reservoir equations and are provided by the flux methods. For example, the storage cannot be negative and cannot exceed the current storage plus all the input. In our experience with the earlier SUPERFLEX-F90, this setup achieves a robust numerical behavior.

Furthermore, as now clarified in **PK.2**, SuperflexPy users can develop adaptive time stepping schemes for the single elements with the "constant-flux-within-a-timestep" limitations already discussed earlier in point **PK.2**. For this reason, it is also possible to overcome convergence problems by employing adaptive time stepping to the solution of the single equations.

We have reflected these points in the updated manuscript (section 5.1.5) and documentation (section 5.1).

***PK.5:*** *RC1.6: Reference is provided now, but the properties of the algorithm should be stated in*
*the supplemental material (limits and speed of convergance). The algorithm is not explained or*
*just described as a mixture of regula falsi with the secant method.*

We have now added the following content to the Documentation (Section 5.1):

The Pegasus algorithm is a bracket-based nonlinear solver similar to the well-known
Regula Falsi algorithm. It employs a re-scaling of function values at the bracket endpoints
to accelerate convergence for strongly curved functions. The authors of the paper (Dowell
& Jarratt, 1972) claim that the algorithm exhibit superior asymptotic convergence
properties to other modified linear methods.

The reference (Dowell & Jarratt, 1972) provides a complete algorithmic description of the
Pegasus root finder. The algorithm is implemented exactly as described in the reference; hence,
we prefer to avoid duplication of this content.

***PK.6:*** *RC1.7: After careful reading of Supl-Section 5.1, I cannot find the information from this*
*answer. The need for smoothing when using the implicit solver must be mentioned as a one-liner*
*in the main text.*

For clarity, the original question (RC1.7) was:

How do the solvers deal with discontinuous or not continuously differentiable flux
equations? The problem is described by Knoben et al 2019's MARRMoT Paper, Ch. 2.4
(https://doi.org/10.5194/gmd-12-2463-2019) - it is the reason why I gave up mimicking
exisiting models with CMF.

And our reply was:

This is a pertinent point. Generally speaking the SuperflexPy philosophy is to use smooth
flux functions. This may include applying smoothing to otherwise discontinuous
formulations – please see previous publications such as Kavetski and Kuczera (2007).

That said, if a user wanted to perform modelling experiments with discontinuous flux
functions, the framework enables to do so. The EE solver can work with non-smooth RHS
of the differential equations, whereas the IE solver requires smooth equations. Users
could also integrate in SuperflexPy their own solvers with more specialized techniques for
non-smooth problems.

These points will be noted briefly in the revised paper and documentation.

This aspect is now clarified in the supplementary material, section 5.1

The suggestion to use smooth methods is indeed mentioned in section 5.1

264   "SuperflexPy provides two built-in numerical approximators (implicit and explicit Euler)
265   and a root finder (Pegasus method). **These methods are best suited when dealing with**
266   **smooth flux functions. If a user wants to experiment with discontinuous flux**
267   **functions, other ODE solution algorithms should be considered.**"

268 However, note that the use of non-smooth flux functions could cause convergence problems only
269 if the root finder does not maintain brackets on the solution– e.g., in the classic Newton-Raphson
270 root finder. Technically speaking non-smooth flux functions can also be used when the root
271 finder is implemented using a bracketing algorithm such as bisection or Pegasus (e.g., Press et al.,
272 1992). Indeed, this is another robustness benefit of the "one-element-at-a-time" strategy.

273 On the other hand, we still recommend smoothing the flux functions because jump discontinuities
274 in these functions can cause mass balance discrepancies (essentially depending on which side of
275 the jump discontinuity is used to calculate the fluxes). We have cited the work of Kavetski and
276 Kuczera (2007) which provide a broader motivation for smoothing the constitutive functions.

277 We have now mentioned the preference for the usage of smooth flux functions also in the paper,
278 section 4.3 and elaborated more in the documentation, section 5.1.

279 *PK.7: RC1.8: My concerns about numerical errors by the operator split are explained in the*
280 *answer to the reviewers (RC1.8), but have not made it in the manuscript – neither in the main text*
281 *nor in the supplemental material. In fact, both m/s and supplement are plainly wrong: m/s l. 514*
282 *suggest a free choice for the selected numerical solver and the supplement mat 5.1 suggests RK-*
283 *solvers as an additional (not yet used) choice. However, RC1.8 explains me (but not the readers),*
284 *that only single step Euler solvers are suitable to solve the system as other solvers would*
285 *introduce the need for a formal integration of the fluxes over the (outer) timestep:*

286 *"When fixed-step solvers are used, this "one-element-at-a-time" strategy is equivalent to*
287 *applying the same (fixed-step) solver to the entire ODE system simultaneously (i.e., no additional*
288 *approximation error is introduced). " (from answer to RC1.8)*

289 *This section needs to make it in the main text of the manuscript, together with a reference for the*
290 *claim.*

291 We agree that information on these numerical issues is pertinent, and have added it to the main
292 text. The new content includes material from all points listed thus far.

293 The information provided in the reply to RC1.8 is already present in the documentation. For
294 clarity, we report here our reply, highlighting in bold the parts that have already been copied in
295 the documentation in section 5.2, titled "Sequential solution of the elements":

296   "**The SuperflexPy framework is built on a model representation that maps to a**
297   **directional acyclic graph. Model elements are solved sequentially from upstream to**

**downstream, with the output from each element being used as input to its downstream elements.**

**When fixed-step solvers are used, this "one-element-at-a-time" strategy is equivalent to applying the same (fixed-step) solver to the entire ODE system simultaneously** (i.e., no additional approximation error is introduced). This is one of the pragmatic reasons we favor the fixed-step implicit Euler scheme.

**When the solvers use internal substepping, then the "one-element-at-a-time" strategy does introduce additional approximation error. This additional approximation error is due to treating the fluxes as constant over the time step, whereas the exact solution would have varying fluxes within the time step. However, in most practical applications, this "uniform flux" approximation is already applied to the meteorological inputs (rainfall and PET), hence applying it to internal fluxes does not represent a large additional approximation.**

The option to solve the system of equation jointly would avoid the "constant flux" approximation for the internal fluxes (but not for the meteorological one). However, the gain in accuracy is expected to be small and come at the expense of a considerable computational effort and additional code complexity.

We agree these details are pertinent – they will be explained in the Documentation and a cross-reference will be added to the Paper.

If individual elements have multiple outgoing fluxes (e.g., streamflow and evapotranspiration), these are calculated simultaneously by solvers such as IE, and there is no need to specify an order for how such outgoing fluxes are calculated (it is however necessary if EE is used)."

Next, we elaborate on the following statement:

"When fixed-step solvers are used, this "one-element-at-a-time" strategy is equivalent to applying the same (fixed-step) solver to the entire ODE system simultaneously (i.e., no additional approximation error is introduced)"

In this case, the solution of upstream elements does not require the solution of the downstream elements. Technically speaking, the Jacobian matrix associated with the system of equations is lower triangular. Hence, the solution can proceed from upstream to downstream elements with no further approximation or iteration needed.

As a quick example, consider the system of ODEs for model M4, which has 2 reservoir elements, UR and FR (section 3.1 of the paper). When discretized using the implicit Euler (IE) scheme, the following system of nonlinear algebraic equations is obtained:

$$\begin{cases} \dfrac{S_{t+1}^{(UR)} - S_t^{(UR)}}{\Delta t} = P - E_P \dfrac{\overline{S_{t+1}^{(UR)}}\left(1+m^{(UR)}\right)}{S_{t+1}^{(UR)}+m^{(UR)}} - P\left(\overline{S_{t+1}^{(UR)}}\right)^{\beta^{(UR)}} & \text{..... equation 1 (element 1, UR)} \\[4mm] \dfrac{S_{t+1}^{(FR)} - S_t^{(FR)}}{\Delta t} = P\left(\overline{S_{t+1}^{(UR)}}\right)^{\beta^{(UR)}} - k^{(FR)}\left(S_{t+1}^{(FR)}\right)^{\alpha^{(FR)}} & \text{..... equation 2 (element 2, FR)} \end{cases}$$

where the unknowns are $S_{t+1}^{(UR)}$ and $S_{t+1}^{(FR)}$, i.e., the storages in element 1 (UR) and element 2 (FR)

respectively (note that $\overline{S_{t+1}^{(UR)}}$ is a function of $S_{t+1}^{(UR)}$).

Equation 1 contains unknown 1, and equation 2 contains both unknowns 1 and 2. Hence the
system of equations (more precisely, its Jacobian matrix) is lower triangular, and can be solved
using forward elimination: solve equation 1 for unknown 1, and then solve equation 2 for
unknown 2 (keeping unknown 1 fixed).

These arguments generalize quite trivially when more than two reservoirs are present.

$$\begin{cases} f_1(S_1) = 0 \\ f_2(S_1, S_2) = 0 \\ f_3(S_1, S_2, S_3) = 0 \\ \quad\vdots \\ f_N(S_1, S_2, S_3, \dots S_N) = 0 \end{cases}$$

It can be seen that no additional approximations are introduced when solving equations one at a
time starting from unknown 1 and finishing with unknown $N$.

Note also that this analysis is distinct from the assumption that the fluxes are constant over the
time step (see point **PK.2**).

Section 5.2 of the documentation is now titled "Sequential solution of the elements and numerical
approximations". This aspect is also mentioned in section 4.3 of the paper.

*PK.8: RK-solvers of nth order use n-1 (or more) substeps to predict the final state at the output*
*timestep by fitting an nth-order polynom into these substeps. Using the flux at Y(t, S(t)) or Y(t+1,*
*S(t+1)) is not the correct number, as the solver calculates the ODE between these timesteps and*
*introduces an uncaught numerical error into the system.*

We now provide a new numerical approximator that implements the Runge Kutta 4 (RK4)
algorithm. Note that, however, due to the "constant-within-timestep" approximation (refer to
**PK.2**), input fluxes to the element are treated as constant; output fluxes, on the other hand, can be
calculated with intermediate states, when solving the differential equation of the element.

*PK.9: While the user is free to use any Jacobian-free root finder, the choice of the ODE-solver is (obviously) limited to implicit and explicit Euler methods (PECE methods might also an alternative). There is no interface to calculate the Jacobian matrix of an Element, hence Newton-like root finding algorithms are not suitable.*

The lack of facility to communicate the Jacobian of an element was indeed a limitation of the previous version of SuperflexPy. As part of the revision, and motivated by the reviewer comment, we have generalized the implementation of the flux function methods to accommodate the analytical calculation of the derivatives of the fluxes with respect to the state. These values are then propagated by the numerical approximators and are provided to the root finder. In turn, this enables the root finder to use algorithms that employ analytical derivatives, such as the classic Newton-Raphson. We have provided a new root finder `NewtonPython` that uses derivatives unless the resulting root jumps out of the brackets, in which case a bisection step is employed (see Press et al., 1992 for the principles of this algorithm).

For generality, this new functionality is implemented as "optional": if the user implements a new flux function but does not wish to derive and implement analytical derivatives, they can specify `None` as the value and then use a derivative-free root finder such as Pegasus.

Note also that the derivatives can be calculated numerically by the root finder itself as part of its internal approximations – this option is trivially available to any root finder but can be computationally expensive and according to the numerical literature is seldom beneficial when solving scalar equations.

We have updated the documentation (chapters 5, 8 and 10) to reflect this enhancement in the SuperflexPy framework.

*PK.10: The freedom of the solver choice is quite limited by the use of the sequential solution of the DAG approach – this is of course valid, but should be made explicit.*

The new section 5.1.3 of the documentation shows how to implement new solvers "from scratch" within the limitations stated in **PK.2**.

*MP4:*

*PK.11: The classical structure of scientific writing is of course not directly fitting with a model description paper. However, I would see the choice of math, programming language and design principles rather as the methods of a model implementation and the resulting code and use examples as the results section. The new, frequent links to other sections, are a poor surrogate for a cleaner structure but are an improvement over the original manuscript.*

As noted in the previous round of reviews, the paper has been organized to cater to two distinct audiences:

• general hydrological/environmental modelers with interest in the capabilities and usage
patterns of the software;
• specialist researchers with interest in technical implementation details.

Meeting the expectations of these two audiences requires some compromises. Our choice has
been to progress from simple aspects accessible to the broader audience to more specialized
aspects requiring a stronger technical background in numerical computation and software design.

We appreciate that a specialist reader may prefer a different presentation structure, but putting
highly technical details first could easily confuse readers without a specialist background. With
that in mind, we do appreciate the reviewer feedback that the revision has been an improvement
over the original manuscript.

*Additional issues:*

*Section 4.2:*

**PK.12:** *The m/s mentions 8 times the object oriented design of the implementation and but does*
*not feature the object oriented design choices at a prominent place. The UML-diagram is now*
*hidden in the last section of the supplemental material. The UML-like diagram should be moved*
*to the main text in section 4.2, as it is essential for the understanding of the object oriented*
*design. Now section 4.2 lists, how the OO-design helps to accomplish certain goals, but we, as*
*the readers, can only guess what that OO-design is.*

We fully agree with this comment. The UML diagram has been integrated in section 4.2.

**PK.13 [the comment has been re-formatted to facilitate its reading]:** *Fig 12: I am familiar with*
*most services and software mentioned in Fig 12 (except binder), however, I had a hard time to*
*understand it.*

1. *Mixing cloud services like github, binder, zenodo, and read the docs with a file format*
*(Jupyter-Notebooks) on an equal level does not help to understand any of these services.*

2. *Having the developer and the user as the same person (symbol) complicates the*
*understanding with the blue and black lines.*

3. *The authors state, that explaining the ecosystem around SuperflexPy is important – while*
*I do not follow the premise, explaining the services is possibly better done with text. But if*
*the ecosystem is important enough for a large figure in the main text (while omitting the*
*UML-Diagram), then the importance should be highlighted throughout the paper, as the*
*object oriented design is. I still recommend to delete this figure.*

4. *If the authors are absolutely sure, this figure is needed, they need to*

*a. redraw the figure using two persons and kicking out Jupyter-Notebooks (as they are not a web service themself) and test the figure with friends from the intended audience*

*b. explain the figure in much more detail and the role of every mentioned service therein, and*

*c. introduce throughout the paper the importance of a webservice ecosystem for modern model development (eg. mention in section 1.2, practical criteria).*

*5. However, as of now, this figure is hardly explained, and someone who is not familiar with these services will not profit from it. Even worse, the figure in its current state is prone to misunderstandings and does a disservice to the paper. Moving the figure as is to the supplement material does not solve the issues mentioned above.*

We believe that a brief description of the "ecosystem" of web services is important for general users in the hydrological community, who in our experience are often not up-to-date with many of these web services/tools. Figure 12 is intended to help readers navigate the way SuperflexPy is integrated into this broader ecosystem. That said, we agree with several points made by the reviewer regarding some technical inaccuracies/confusions in the way Figure 12 was presented, and have made the following changes to the figure:

- Removed Jupyter as it is indeed a file format not a web service
- Distinguished the "user" from the "developer"
- Distinguished automated steps (dashed lines) from "manual" steps (continuous lines)

Moreover, as suggested by the reviewer, we have enhanced the main text to motivate the importance of a deployment pipeline and of using web services in the introduction of the paper (lines 179-181).

A succinct explanation of the tools and their roles depicted in Figure 12 can be found in the main text Section 5.1.3

Figure 12 shows the online software management tools that are used to develop and deploy SuperflexPy. The framework itself, including source code, documentation, examples, etc., is hosted on **GitHub**. Automated workflows are then used to create new releases (**PyPI**), get DOIs for the software releases (**Zenodo**), host the documentation (**ReadTheDocs**), and run the examples (Jupyter and **Binder**).

A more detailed explanation is provided in the documentation Chapter 2, which shows already the same picture and explains, with greater detail, the role of the services (Binder was missing and has been now added)

The source code, documentation, and examples are part of the official repository of
SuperflexPy hosted on **GitHub**. A user who wishes to read the source code and/or modify
any aspect of SuperflexPy (source code, documentation, and examples) can do it using
GitHub.

New releases of the software are available from the official Python Package Index (**PyPI**),
where SuperflexPy has a dedicated page. [link to the PyPI page]

The documentation builds automatically from the source folder on GitHub and is
published online in **Read the Docs**. [link to the documentation]

Examples are available on GitHub as Jupyter notebooks. These examples can be
visualized statically or run in a sandbox environment (see Examples for further details).
[Link to a page in the documentation that lists the examples and links to GitHub and
Binder]

We thank the reviewer for their feedback on this important usability aspect, which is now
presented in a clearer and technically more sound way.

*PK.14: Jansen et al (2020) reference: This is unpublished work, and I as a reviewer am unable to*
*check the content of this reference and review its role in the paper. The author's claim about its*
*content might be wrong. As such, the reference needs to be removed. If a public preprint had*
*been cited, this problem would not exist.*

The reference in question is the following one:

Jansen, K. F., Teuling, A. J., Craig, J. R., Dal Molin, M., Knoben, W. J. M., Parajka, J., Vis,
M., and Melsen, L. A.: Mimicry of a conceptual hydrological model (HBV): What's in a
name?, Water Resources Research, n/a, e2020WR029143,
https://doi.org/10.1029/2020WR029143, 2021.

That paper is not unpublished work - it was accepted before the previous revision of the
SuperflexPy manuscript was re-submitted, and the reference given was (and still is) for the
accepted paper.

We checked that the doi is working properly, so that the Reviewer can certainly access it if they
wish. We understand that the "n/a" may have caused some confusion, and have corrected it.

**Response to comments by AR2**

*AR2.1: The authors have done a commendable job addressing the detailed comments of the reviews. While there was quite a bit of "pushback" with respect to reviewer suggestions for including more rigorous comparisons with existing frameworks and inclusion of more implementation details, I found their arguments for resisting these recommendations for the most part convincing, and they have done an effective job addressing the spirit of these comments without (for instance) over-duplicating the contents of the software documentation or getting pulled into the details of an exhaustive model intercomparison. As such, I recommend acceptance subject to (very) minor revision.*

We thank the reviewer for their recognition of our effort in improving the paper and for their appreciation of our reasoning against some earlier proposed changes (where those would have been impractical within the scope of the current paper).

*Major comment:*

*AR2.2: I still have a bit of concern with the undue apparent stress on transport simulation capabilities which are \*not present in the existing model\*. This is highlighted as a key "realm" in the application scope (line 193 of marked up revision), at line 203, and elsewhere.*

*This concern could be mitigated by revising line 193 to "Extendibility for future applications, e.g., isotope or pesticide transport modelling". Any hydrological model can technically be extended to support transport, and it is by no means clear that SuperFlexPy is more extendible than others (without explicitly demonstrating it).*

We appreciate the concern of the reviewer. As part of the revisions, to avoid an inadvertent "undue stress" on this concept, we have checked every mention of "transport simulation" in the context of SuperflexPy to ensure it clearly refers to extendibility for future applications (thus addressing the reviewer concerns) rather than a currently available feature. References to transport simulation in general hydrological modelling contexts (rather than in SuperflexPy-specific contexts) were kept as is, as they indeed provide the motivation to support future extendibility.

We list below all the sentences in the paper where modelling of transport processes or chemistry is mentioned (only here, in order to maintain a correspondence between question and answer, line numbers refer to the marked up version that the reviewers refers to) to clarify our actions:

1. Line 89

    "However, their application extends to the simulation of other environmental variables such as groundwater levels (e.g., Seibert and McDonnell, 2002) and soil moisture (e.g., Matgen et al., 2012), as well as water chemistry (e.g., Bertuzzo et al., 2013; Ammann et al., 2020)."

This sentence is about the general areas of application of hydrological models, not about SuperflexPy. Therefore, the sentence was kept as is.

2. Line 189

> "In terms of application scope of a flexible framework for conceptual hydrological modeling, we focus on the following "realms": [...] Substance transport modelling, including water isotopes, pesticides, etc".

We have changed to "Support or extendibility for future applications, e.g. substance transport modelling, including water isotopes, pesticides, etc.", as proposed by the reviewer. Note that this is a general statement for flexible frameworks.

3. Line 195

> "In terms of software implementation, we consider the following practical criteria: [...] 2. Ease of modification and extension. Even a comprehensive software implementation will eventually require extension. For example, a modeling framework intended to simulate streamflow may require extension to simulate water chemistry."

This sentence refers to the desired property that a flexible framework should be easy to modify and extend – and mentions the simulation of transport processes as an example of possible future extension. SuperflexPy is designed with this requirement in mind (i.e., of being easy to modify and extend). Therefore, no change has been made – indeed by definition an example of future extension should be something not implemented in the current code.

4. Line 231

> "The original Fortran implementation of SUPERFLEX, hereafter referred to as SUPERFLEX-F90, has been used in a series of case studies over the last decade, [...] inclusion of pesticide/substance transport (e.g. Ammann et al., 2020)."

This sentence refers to past applications of SUPERFLEX-F90, which does in fact include a substance transport module. Note that SUPERFLEX-F90 is a different implementation that as such is unrelated to SuperflexPy. Therefore, the sentence is factually correct and was kept as is.

5. Line 722

> "The capability to simulate multiple fluxes and states is intended to support the extension of SuperflexPy to new modelling scenarios. Several such scenarios may be of interest, including the transport of chemical substances (e.g., Fenicia et al., 2010; Ammann et al., 2020) [...]".

The sentence lists possible applications where "the capability to simulate multiple fluxes and states" may be useful. The general ability to simulate multiple fluxes and states does not imply that specific modelling contexts where such one of the applications is simulating transport processes does not implies that this is readily available.

We already have remarked this concept also in the following paragraph (line 730)

> "While the current examples in SuperflexPy do not include all the cases listed above, […]"

We have changed the sentence to "support the **future** extension" (i.e., adding the word "future") to put emphasis, on the fact that this feature is not yet implemented.

These changes address the remaining confusion regarding "what is" vs. "what is not" supported, and clearly state that transport simulation is currently not supported.

*Minor comments: (line numbers refers to marked up manuscript)*

*AR2.3: line 200- "modifications and extensions"-->"modification and extension"*

Thank you – change implemented.

*AR2.4: line 217- remove "or even impossible"*

Thank you – change implemented.

*AR2.5: line 244- "highlighted implementation choices" - such as? This is very vague. If you are going to note that SuperFlexPy will address these limitations, you have to state what they are.*

We agree this was vague. We have clarified on line 207 that this mainly refers to the use of a "master template" from which specific model structures are derived. Note that subsequent Tables 1 and 2 provide a detailed summary of differences, which are moreover discussed in the text in section 5.1.

*AR2.6: line 383- The value of the stand alone statement "All SuperFlexPy componets are..." is unclear (as is the connection to the previous paragraph). What does it mean to be "characterized by" a state or parameter?*

This sentence introduces that SuperflexPy components have states and p*arameters*. We have changed "characterized by" to "have" for clarity.

*AR2.7: line 409- "More specifically" ->"Specifically"*

Thank you – change implemented.

*AR2.8: references- some cleanup of the references is needed w.r.t. inconsistent capitalization, etc.*

Thank you for noticing this - we have now fixed all issues we could spot.

**References**

Clark, M. P., & Kavetski, D. (2010). Ancient numerical daemons of conceptual hydrological modeling: 1. Fidelity and efficiency of time stepping schemes. *Water Resources Research, 46*(10). https://agupubs.onlinelibrary.wiley.com/doi/abs/10.1029/2009WR008894

Dowell, M., & Jarratt, P. (1972). The "Pegasus" method for computing the root of an equation. *BIT Numerical Mathematics, 12*(4), 503-508. https://doi.org/10.1007/BF01932959

Kavetski, D., & Clark, M. P. (2010). Ancient numerical daemons of conceptual hydrological modeling: 2. Impact of time stepping schemes on model analysis and prediction. *Water Resources Research, 46*(10). https://agupubs.onlinelibrary.wiley.com/doi/abs/10.1029/2009WR008896

Press, W. H., Teukolsky, S. A., Flannery, B. P., & Vetterling, W. T. (1992). *Numerical recipes in Fortran 77: volume 1, volume 1 of Fortran numerical recipes: the art of scientific computing*: Cambridge university press.